# Pericyte-derived fibrotic scarring is conserved across diverse central nervous system lesions

David O. Dias [1,8], Jannis Kalkitsas[1,8], Yildiz Kelahmetoglu [1], Cynthia P. Estrada[2], Jemal Tatarishvili[3], Daniel Holl[1], Linda Jansson [3], Shervin Banitalebi[4], Mahmood Amiry-Moghaddam[4], Aurélie Ernst [1,5], Hagen B. Huttner[6], Zaal Kokaia [3], Olle Lindvall[3], Lou Brundin [2], Jonas Frisén[1] & Christian Göritz [1,7 ✉]

Fibrotic scar tissue limits central nervous system regeneration in adult mammals. The extent of fibrotic tissue generation and distribution of stromal cells across different lesions in the brain and spinal cord has not been systematically investigated in mice and humans. Furthermore, it is unknown whether scar-forming stromal cells have the same origin throughout the central nervous system and in different types of lesions. In the current study, we compared fibrotic scarring in human pathological tissue and corresponding mouse models of penetrating and non-penetrating spinal cord injury, traumatic brain injury, ischemic stroke, multiple sclerosis and glioblastoma. We show that the extent and distribution of stromal cells are specific to the type of lesion and, in most cases, similar between mice and humans. Employing in vivo lineage tracing, we report that in all mouse models that develop fibrotic tissue, the primary source of scar-forming fibroblasts is a discrete subset of perivascular cells, termed type A pericytes. Perivascular cells with a type A pericyte marker profile also exist in the human brain and spinal cord. We uncover type A pericyte-derived fibrosis as a conserved mechanism that may be explored as a therapeutic target to improve recovery after central nervous system lesions.

[1] Department of Cell and Molecular Biology, Karolinska Institutet, Stockholm, Sweden. [2] Department of Clinical Neuroscience, Karolinska University Hospital, Solna, Sweden. [3] Lund Stem Cell Center, Lund University, Lund, Sweden. [4] Division of Anatomy, Department of Molecular Medicine, Institute of Basic Medical Sciences, University of Oslo, Oslo, Norway. [5] Group Genome Instability in Tumors, German Cancer Research Center, Heidelberg, Germany. [6] Department of Neurology, University Hospital Erlangen, Erlangen, Germany. [7] Ming Wai Lau Centre for Reparative Medicine, Stockholm Node, Karolinska Institutet, Stockholm, Sweden. [8] These authors contributed equally: David O. Dias, Jannis Kalkitsas. ✉email: christian.goeritz@ki.se

Regeneration of the central nervous system (CNS) in adult mammals is limited. Scar tissue that forms after CNS lesions is essential for sealing off the injured tissue and containing the damage[1–4]. However, it also functions as a permanent barrier to regenerating axons, contributing to the long-lasting functional deficits observed after injury[5,6].

The mature CNS scar is a compartmentalized, multicellular structure. Non-neural cells, including extracellular matrix (ECM)-producing stromal fibroblasts and peripherally derived macrophages, constitute the fibrotic, or stromal, component of the scar[2]. Most research has focused on reactive astrocytes and NG2-expressing glia constituting the glial component of the scar[5,7], and much less is known about fibrotic scar tissue.

Using in vivo lineage tracing, we previously identified a small subset of perivascular cells, named type A pericytes, as the cellular origin of stromal fibroblasts present in fibrotic scar tissue after penetrating spinal cord injury. Type A pericytes are initially recruited to mediate wound closure, but consequently form fibrotic scar tissue, which constitutes a barrier for axon regeneration[6]. Type A pericytes are located in the vascular wall encased by basal lamina and represent about 10% of all platelet-derived growth factor receptor-beta (PDGFRβ)-expressing perivascular cells on the microvasculature in the uninjured adult mouse spinal cord. They can be distinguished from other non-scar-forming perivascular cells by the expression of the glutamate aspartate transporter GLAST (gene name Slc1a3), which can be used for selective targeting of type A pericytes. Following spinal cord injury, type A pericytes enter the lesion site with sprouting vessels during a transient phase of vascular remodeling. Between 3 and 9 days post-injury, type A pericytes proliferate intensely, break through the surrounding vascular basal lamina and migrate into the lesion, where they cluster and contribute to the formation of the fibrotic scar[3].

Although different cell types were proposed to contribute to fibrotic scarring[8–11], the cellular origin of scar-forming fibroblasts following CNS lesions in the brain remains elusive, due to the lack of in vivo genetic fate mapping studies required to unequivocally trace cells and establish lineage relationships. It is also unclear to what extent fibrotic scar tissue enriched in stromal fibroblasts is formed in humans after CNS lesions, such as spinal cord injury, stroke, multiple sclerosis (MS), and glioblastoma multiforme (GBM).

Here, we show that fibrotic scar tissue formation by type A pericytes is preserved throughout the CNS, while the extent and distribution of stromal cells varies depending on the lesion. Type A pericytes are the main source of PDGFRβ-expressing stromal cells in mouse models of penetrating and non-penetrating spinal cord injuries, traumatic brain injury, ischemic stroke and MS, but contribute less extensively to tumor stroma. Furthermore, as in the mouse, we find a subset of perivascular cells with a type A pericyte marker profile in the healthy human brain and spinal cord. After spinal cord injury, humans develop non-neural, fibrotic scar tissue with large numbers of non-vessel-associated stromal fibroblasts. In spinal cords of individuals with active MS, PDGFRβ-expressing stromal cells mostly accumulate in the perivascular space. In comparison, PDGFRβ-expressing stromal cells increase in density but remain mostly associated with the vasculature in stroke patients with subcortical ischemic lesions and in the stroma of aggressive grade IV human GBM tumors.

## Results

### Genetic labeling of type A pericytes in the adult mouse CNS.
We employed GLAST-CreER$^{T2}$ transgenic mice[12] carrying Rosa26-enhanced yellow fluorescent protein (R26R-EYFP) or Rosa26-tdTomato (R26R-tdTom) reporter alleles (hereafter referred to as GLAST-CreER$^{T2}$;R26R-EYFP or GLAST-

CreER$^{T2}$;R26R-tdTom) to genetically fate map type A pericytes, as previously established in the spinal cord[3,6] (Supplementary Fig. 1a). Upon tamoxifen-mediated genetic recombination, GLAST-expressing type A pericytes are inheritably labeled by EYFP or tdTom expression. Under homeostatic conditions, type A pericytes are associated with blood vessels throughout the gray and white matter spinal cord parenchyma and distribute along the capillary bed, and upstream venous and arterial vasculature[3] (Supplementary Fig. 1b–d). Type A pericytes express the widely used pericyte markers PDGFRβ and CD13 (also known as aminopeptidase N), and T-box transcription factor 18 (Tbx18), but are not marked by desmin and alpha smooth muscle actin (αSMA), found in other mural cells (i.e., type B pericytes and vascular smooth muscle cells)[3] (Supplementary Fig. 1e–i). GLAST-expressing type A pericytes had typical ultrastructural features of pericytes including being juxtaposed to the capillary endothelial cells and being embedded within the basal lamina (Supplementary Fig. 2a, b). As previously described[3], at positions where processes intersect, type A pericytes were commonly located abluminal to the other, non-recombined, pericytes (i.e. type B pericytes) (Supplementary Fig. 2c).

Likewise, we detected recombined perivascular cells with similar distribution along the vasculature, sharing the same marker expression and comprising about 10% of all PDGFRβ-expressing perivascular cells in the uninjured adult cerebral cortex and striatum (Supplementary Fig. 3a–h), identifying them as type A pericytes. Apart from type A pericytes, we observed occasional recombination in a small subset of ependymal cells and white matter radial astrocytes in the spinal cord[3] and associated with the vasculature in the meninges surrounding the brain and spinal cord (Supplementary Figs. 1b, 3a). Additionally, the GLAST-CreER$^{T2}$ transgene allows recombination in a subset of parenchymal astrocytes throughout the brain (i.e. striatal and cortical astrocytes, Supplementary Fig. 4a–e), as well as in subventricular zone astrocyte-like neural stem cells[12,13]. Astrocytes, ependymal cells, and their respective progeny do not participate in fibrotic scar tissue formation[14] and do not express the stromal marker PDGFRβ, which allows us to distinguish them from type A pericyte-derived scar-forming fibroblasts.

### Type A pericytes contribute to fibrotic scar formation following penetrating and non-penetrating spinal cord injury.
To compare the contribution of type A pericytes to fibrotic scarring after penetrating and non-penetrating spinal cord injury, we lineage traced type A pericytes after a dorsal funiculus incision[3,14,15] and after a complete crush injury[16], respectively (Fig. 1 and Supplementary Fig. 5). Injuries were performed following a clearing period of 7 days after tamoxifen-mediated recombination, ensuring that all recombination occurs prior to lesioning[3,6].

We compared type A pericyte-derived scar formation during the subacute phase at 5 days post-injury (dpi), at the time of glial-stromal border establishment at 14 dpi, and at a chronic time point at 6–7 weeks post-injury (wpi). While the uninjured spinal cord only contained a small number of recombined (EYFP$^+$) type A pericytes (Fig. 1a, b and Supplementary Figs. 1, 5a, b), we found many EYFP$^+$ type A pericyte-derived cells within the lesioned parenchyma at all studied time points after complete spinal crush and dorsal funiculus incision (Fig. 1c–h and Supplementary Fig. 5c–h).

At 5 dpi, the total number of EYFP$^+$ cells expressing the stromal marker PDGFRβ$^+$ had drastically increased in the lesion core compared to uninjured conditions, and most recombined cells were no longer associated with the blood vessel wall (Fig. 1i–k and Supplementary Figs. 5i–k, 6a). EYFP$^+$ cells proliferated (Ki67$^+$) both when associated with and away from

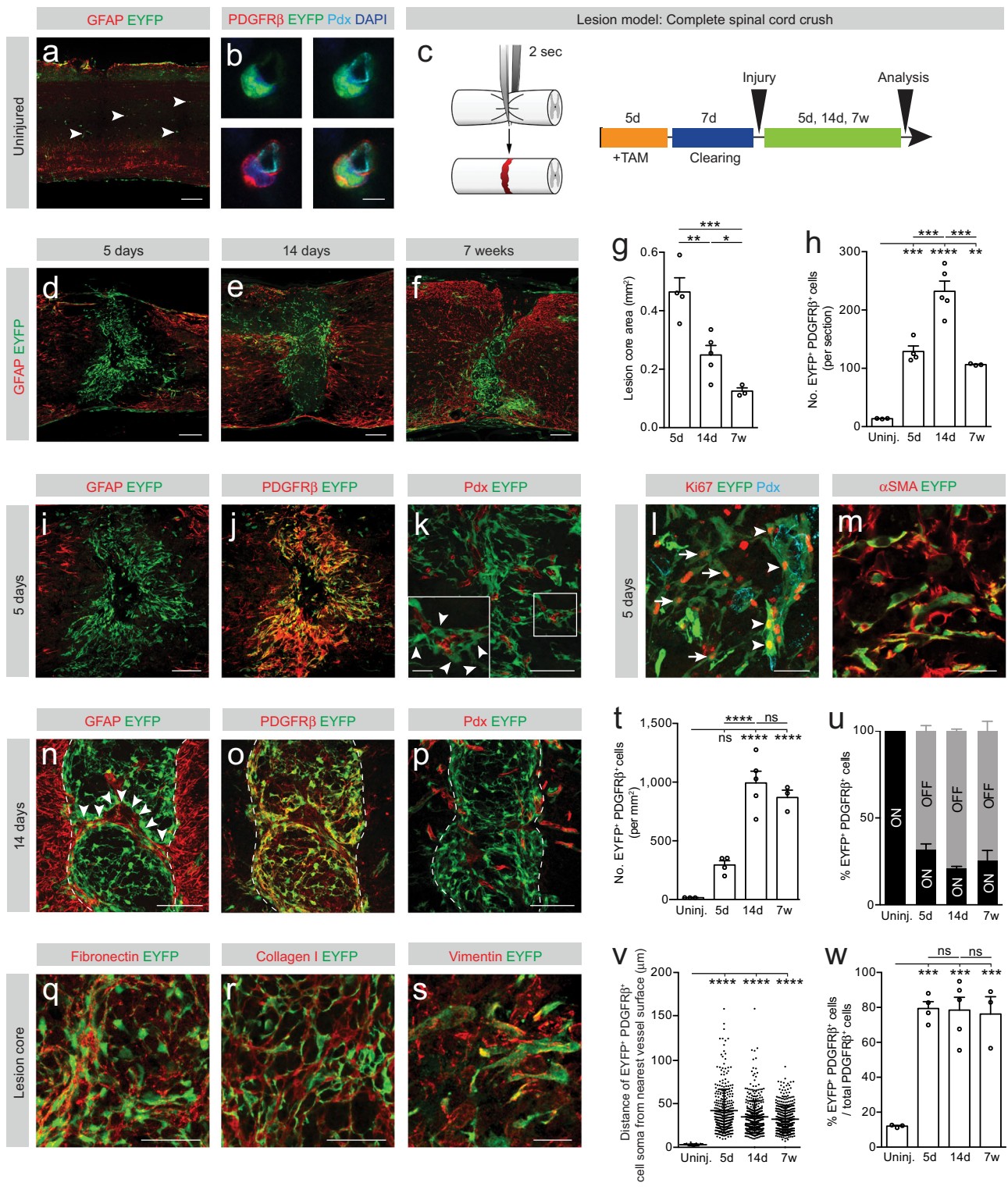

the vessel wall and expressed the (myo)fibroblast markers αSMA and vimentin[17] (Fig. 1l, m, s and Supplementary Fig. 5l, m, r). While we found type A pericytes and progeny expressing the NG2 proteoglycan, the majority of EYFP⁺ cells did not share this marker after injury (Supplementary Fig. 6b), suggesting that type A pericyte progeny is distinct from NG2-expressing cells involved in scar formation after spinal cord injury[18]. In addition, we observed that type A pericytes and progeny were embedded in collagen I-positive ECM in the lesion core and surrounding tissue (Supplementary Fig. 6c).

At 14 dpi the fibrotic lesion core was flanked by glial fibrillary acidic protein (GFAP)⁺ astrocytes, establishing a sharp border between the glial and fibrotic scar compartments (Figs. 1n, o and Supplementary Fig. 7a–c). As a result of lesion border formation, type A pericyte-derived cells apposed to reactive glia at the glial-fibrotic interface exhibited a distinct morphology and arranged differently compared to EYFP⁺ cells in the inner core of the fibrotic scar (Supplementary Fig. 7b, c). In regions of the scar devoid of type A pericyte-derived cells, we could observe GFAP⁺ astrocyte processes crossing the lesion site, forming glial bridges

**Fig. 1 Type A pericytes are the main source of stromal fibroblasts that form fibrotic scar tissue after non-penetrating spinal crush injury. a** A subset of perivascular cells, named type A pericytes, is recombined (EYFP$^+$, arrowheads) in the uninjured spinal cord of *GLAST-CreER^T2;R26R-EYFP* mice. **b** Type A pericyte (EYFP$^+$) lining the endothelial wall (marked with podocalyxin, Pdx) and expressing the pericyte marker PDGFRβ. **c** Lesion model and experimental timeline. Red represents the lesion. Distribution of EYFP$^+$ cells and GFAP$^+$ glial cells at 5 dpi (**d**), 14 dpi (**e**) and 7 wpi (**f**). Lesion core area (**g**) and number of EYFP$^+$PDGFRβ$^+$ cells per section (**h**). EYFP$^+$PDGFRβ$^+$ cells are bordered by GFAP$^+$ reactive astrocytes (**i, j**) and locate outside the vascular wall (Pdx$^+$; arrowheads) at 5 dpi (**k**). Inset in (**k**) shows magnified boxed region. EYFP$^+$ cells proliferate (Ki67$^+$) while associated with (arrowheads) or away from (arrows) the vascular wall (**l**), and express the (myo)fibroblast marker αSMA (**m**) at 5 dpi. **n** A sharp lesion border forms at 14 dpi, segregating the glial (GFAP$^+$) and fibrotic (EYFP$^+$) compartments of the scar. Arrowheads point at a glial bridge. EYFP$^+$PDGFRβ$^+$stromal cells populate the lesion core (**o**) and a fraction is located away from the vascular wall (**p**) at 14 dpi. EYFP$^+$ cells are embedded in fibronectin- (**q**) and collagen I- (**r**) rich ECM (14 dpi) and express vimentin (**s**) (5 dpi). **t** Density of EYFP$^+$PDGFRβ$^+$ cells in the fibrotic core. **u** Percentage of EYFP$^+$ cells that express PDGFRβ (EYFP$^+$PDGFRβ$^+$ cells) associated with (ON vessel) or located away from (OFF vessel) the vascular wall. **v** Distance of EYFP$^+$PDGFRβ$^+$ cells ON vessel (uninjured spinal cord) or OFF vessel (after injury) from the nearest vessel surface. Each dot represents one cell. **w** Percentage of PDGFRβ$^+$ cells that express EYFP out of total PDGFRβ$^+$cells. TAM tamoxifen; Scale bars: 200 μm (**a, d–f**), 100 μm (**i–k, n–p**), 50 μm (**l, q, r**), 25 μm (inset **k, m, s**) and 5 μm (**b**). Data shown as mean ± s.e.m. $n = 3$ (Uninjured), $n = 4$ (5d), $n = 5$ (14d) and $n = 3$ (7w) animals in (**g, h, t, u, w**); $n = 69$ (Uninjured), $n = 334$ (5d), $n = 423$ (14d) and $n = 408$ (7w) cells examined over 3 animals in (**v**). ns non-significant; *$p < 0.05$, **$p < 0.01$, ***$p < 0.001$, ****$p < 0.0001$ by One-Way ANOVA followed by Holm–Sidak post-hoc test in (**g, h, t, w**) and Kruskal–Wallis test followed by Dunn's post-hoc test in (**v**). Dashed lines in (**n–p**) outline the lesion core. **i, j, n, o** denote paired images. Cell nuclei are labeled with DAPI. All images show sagittal sections. Images are representative of two independent experiments. Source data and statistical details are provided as a Source Data file.

(Fig. 1n, o). EYFP$^+$ cells retained expression of PDGFRβ and *Tbx18* after dissociation from the vascular wall, and were surrounded by fibronectin- and collagen I-positive ECM deposits (Fig. 1o-r and Supplementary Figs. 5n–q, 6c, d). The number of EYFP$^+$PDGFRβ$^+$ cells peaked at 14 dpi and then dropped by 6–7 wpi, as the scar matured (Fig. 1h and Supplementary Fig. 5h). Over time, the lesion site condensed (Fig. 1g and Supplementary Fig. 5g), leading to increased EYFP$^+$PDGFRβ$^+$ cell density in the lesion core (Fig. 1t and Supplementary Fig. 5s).

Fibrotic scar tissue contains PDGFRβ$^+$ stromal cells associated with the blood vessel wall (ON the vessel), representing perivascular cells, and extravascular (OFF the vessel) PDGFRβ$^+$cells, representing stromal fibroblasts. We noticed that spinal cord injury triggered robust detachment of recombined cells from the blood vessel wall. Approximately 56 ± 4% and 74 ± 6% of EYFP$^+$ cells that expressed PDGFRβ were no longer associated with the vascular wall (i.e. no contact of processes or soma with the blood vessel, and soma >8 μm from the blood vessel surface) at chronic stages after dorsal funiculus incision and complete spinal cord crush, respectively (Fig. 1u, v and Supplementary Figs. 5t, u, 8a–e).

After complete spinal cord crush injury, the overall contribution of type A pericytes to all PDGFRβ$^+$ stromal cells in the fibrotic lesion core was on average 80% at all time points investigated. This value is likely to represent an underestimate as incomplete recombination is expected[3,6] and the highest animal showed up to 95% contribution (Fig. 1w). These results were further corroborated using the dorsal funiculus incision model (Supplementary Fig. 5v).

Type A pericytes reacted in a similar manner and showed comparable dynamics in response to penetrating and non-penetrating lesions to the spinal cord (Fig. 1 and Supplementary Fig. 5). Although astrocyte- and ependymal cell-derived progeny participate in glial scar formation, they do not contribute to the fibrotic compartment of the scar[14] and do not express the stromal marker PDGFRβ. Virtually all recombined cells found within the fibrotic scar expressed the stromal marker PDGFRβ (Fig. 1j, o and Supplementary Fig. 5i, n), ruling out significant contribution from sparsely recombined ependymal cells and radial astrocytes to fibrotic scar-forming cells, as previously established[3].

We conclude that nearly all scar-forming PDGFRβ$^+$ stromal cells derive from type A pericytes after penetrating and non-penetrating injuries to the spinal cord.

**Type A pericytes contribute to fibrotic scar formation following traumatic brain injury.** To investigate whether type A pericyte-derived scarring is conserved throughout the CNS, we lineage traced type A pericytes after a stab wound to the brain (Fig. 2 and Supplementary Fig. 9a).

Small stab lesions restricted to the cerebral cortex mainly induce gliosis[19] and do not generate extensive fibrotic tissue, as seen by no substantial increase in PDGFRβ$^+$ stromal fibroblasts at 14 dpi (Supplementary Fig. 9b–f). As expected, we observed recombined GFAP$^+$ astrocytes (Supplementary Fig. 9b) but little to no contribution of lineage traced type A pericytes after small cortical stab lesions (Supplementary Fig. 9c–f). In contrast, larger cortico-striatal stab lesions triggered robust fibrotic and glial scarring and allowed to investigate the contribution of type A pericytes to fibrotic scarring in the brain (Fig. 2a–h). At 5 dpi, we found that type A pericyte-derived EYFP$^+$ cells expressing the stromal marker PDGFRβ dispersed over the lesion core and no longer associated with the blood vessel wall (Fig. 2d, i–k). EYFP$^+$ cells proliferated both associated with and away from the vascular wall and expressed the (myo)fibroblast markers αSMA and vimentin (Fig. 2l, m, s). The number of recombined PDGFRβ$^+$ stromal cells peaked at 14 dpi and then dropped by 7 wpi (Fig. 2h). At 14 dpi a compact fibrotic scar core filled with type A pericyte-derived stromal cells was walled off by partially recombined GFAP-expressing astrocytes (Fig. 2e, n–p) and remained up to 7 wpi (Fig. 2f, Supplementary Fig. 8f, g). As observed after spinal cord injury, EYFP$^+$ cells were embedded in fibronectin- and collagen I- rich ECM (Fig. 2q, r). Overtime, the scar matured and condensed, leading to an increase in EYFP$^+$PDGFRβ$^+$ cell density in the lesion core (Fig. 2g, t). About half of the EYFP$^+$ cells that expressed PDGFRβ were no longer associated with the blood vessel wall in the lesion core (Fig. 2u, v). After cortico-striatal stab wounds around 80% of all PDGFRβ$^+$ stromal cells associated with the fibrotic scar were recombined at all time points investigated (Fig. 2w). This number may underrepresent the contribution of type A pericytes as recombination is expected to be incomplete. As for spinal cord injury, we conclude that almost all scar-forming PDGFRβ$^+$ stromal cells derive from type A pericytes after cortico-striatal stab lesions.

Fibrotic scar tissue only formed following larger cortico-striatal stab lesions as opposed to smaller injuries restricted to the cerebral cortex. To investigate whether the anatomical location or lesion size influence fibrotic scar tissue generation, we next lineage traced type A pericytes following a stab wound restricted to the cerebral cortex with the same depth and size as the cortico-striatal stab lesion. We refer to this lesion

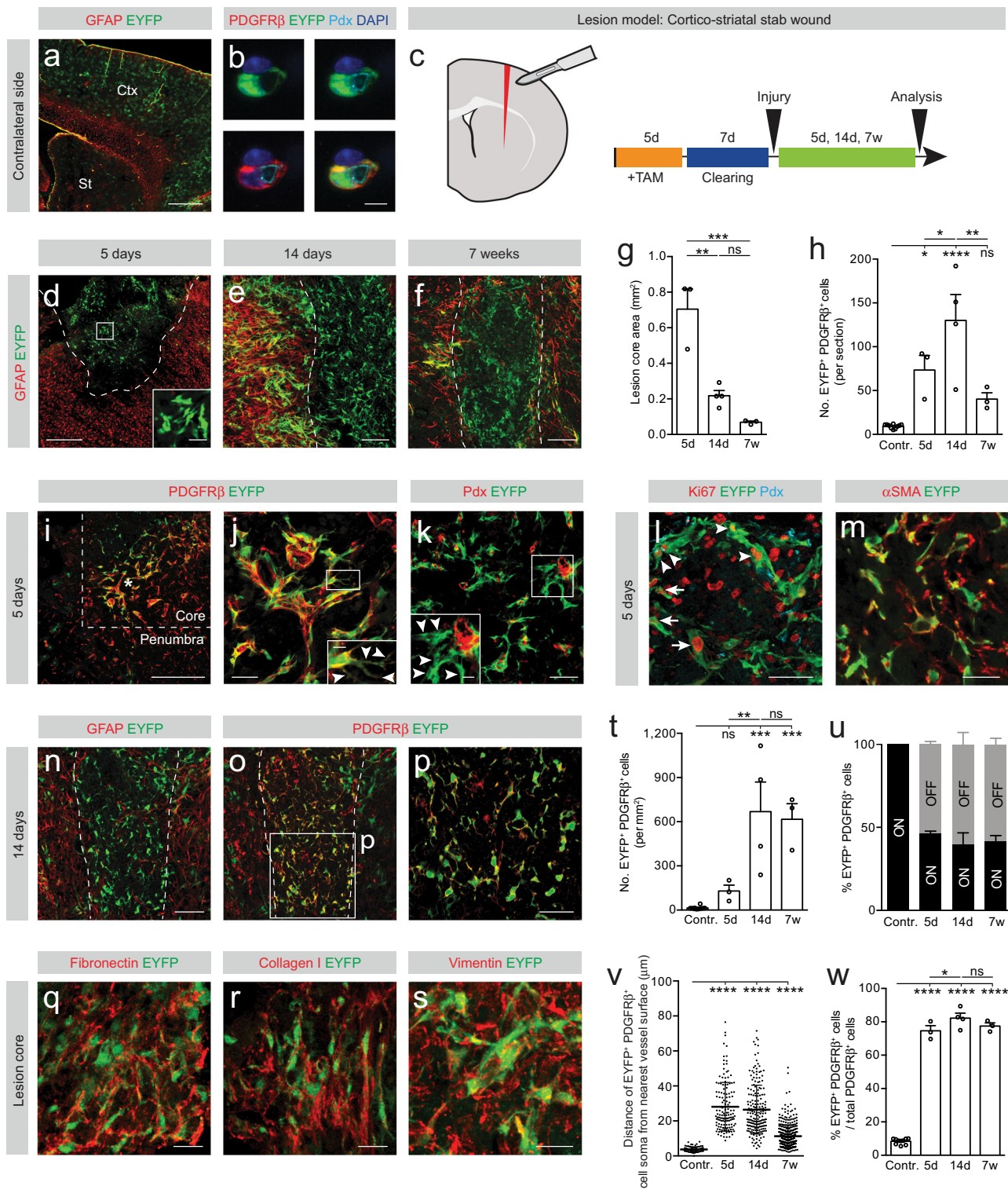

model as large cortical stab wound (Fig. 3a). As opposed to small cortical lesions (Supplementary Fig. 9), large cortical stab wounds triggered widespread glial and fibrotic scarring and enabled us to examine the behavior and contribution of type A pericytes to fibrotic scarring in the cerebral cortex (Fig. 3b–e).

At 5 dpi we found that type A pericytes and progeny co-expressed *Tbx18* and the stromal marker PDGFRβ and filled up the fibrotic lesion core, immediately flanked by the glial scar (Fig. 3b, d, f–h, Supplementary Fig. 10). Many recombined cells

no longer associated with the blood vessel wall, and expressed the mitotic marker Ki67 (Fig. 3i, j) and the (myo)fibroblast markers αSMA and vimentin (Fig. 3k, r).

At 14 dpi recombined cells in the fibrotic lesion core retained expression of PDGFRβ and most remained dissociated from the vascular wall (Fig. 3m–o). Type A pericyte-derived cells were delimited by dense deposits of fibronectin[+] and collagen I[+] ECM (Fig. 3p, q). As observed after cortico-striatal stab lesions, GFAP-expressing astrocytes making up the glial scar are partly recombined (Fig. 3b, c, f, l).

**Fig. 2 Type A pericytes are the main source of stromal fibroblasts that form fibrotic scar tissue after cortico-striatal stab lesions.** Type A pericytes and parenchymal astrocytes are recombined (EYFP[+]) throughout the cerebral cortex (Ctx) and striatum (St) of GLAST-*CreER^T2;R26R-EYFP* mice. **b** Type A pericyte (EYFP[+]) encircling the endothelial tube (podocalyxin[+], Pdx) and expressing the pericyte marker PDGFRβ in the cortex. **c** Lesion model and experimental timeline. Red represents the lesion. Distribution of EYFP[+] cells at 5 dpi (**d**), 14 dpi (**e**), and 7 wpi (**f**). The GFAP[+] glial scar is partly recombined. Type A pericyte-derived cells (EYFP[+]GFAP[−] cells) cluster in the lesion core. Inset in (**d**) shows magnified boxed region. Lesion core area (**g**) and number of EYFP[+]PDGFRβ[+] cells per section (**h**). EYFP[+] cells express PDGFRβ (**i**, **j**) and a fraction is located outside the vascular wall (Pdx[+]; arrowheads in **k**) at 5 dpi. Arrowheads point at protrusion-like structures in (**j**). Asterisk marks the lesion core; dashed line separates the lesion core from penumbral tissue (**i**). Insets show magnified boxed regions. EYFP[+]cells proliferate (Ki67[+]) while attached to the vascular wall (arrowheads) and away from it (arrows) (**l**) and express the (myo)fibroblast marker αSMA (**m**) at 5 dpi. **n–p** EYFP[+]PDGFRβ[+] cells occupy the lesion core and are bordered by GFAP[+] reactive glia at 14 dpi. **q–s** EYFP[+] cells are surrounded by fibronectin[+] (**q**) and collagen I[+] (**r**) ECM and express vimentin (**s**) at 14 dpi and 5 dpi, respectively. **t** Density of EYFP[+]PDGFRβ[+] cells in the fibrotic core. **u** Percentage of EYFP[+] cells that express PDGFRβ (EYFP[+]PDGFRβ[+] cells) associated with (ON vessel) or located away from (OFF vessel) the vascular wall. **v** Distance of EYFP[+]PDGFRβ[+] cells ON vessel (contralateral side to the lesion) or OFF vessel (after injury) from the nearest vessel surface. Each dot represents one cell. **w** Percentage of PDGFRβ[+] cells that express EYFP out of total PDGFRβ[+] cells. TAM tamoxifen; Scale bars: 400 μm (**a**, **d**, **i**), 100 μm (**e**, **f**, **k**, **n–p**), 50 μm (inset **d**, **j**, **l**), 25 μm (inset **k**, **m**, **q–s**), 10 μm (inset **j**) and 5 μm (**b**). Data shown as mean ± s.e.m. $n = 9$ (Contralateral), $n = 3$ (5d), $n = 3$ (14d) and $n = 3$ (7w) animals in (**g**, **h**, **t**, **u**, **w**); $n = 94$ (Contralateral), $n = 142$ (5d), $n = 194$ (14d) and $n = 397$ (7w) cells examined over 3 animals in (**v**). ns non-significant; *$p < 0.05$, **$p < 0.01$, ***$p < 0.001$, ****$p < 0.0001$ by One-Way ANOVA followed by Holm–Sidak post-hoc test in (**g**, **h**, **t**, **w**) and Kruskal–Wallis test followed by Dunn's post-hoc test in (**v**). Dashed lines in (**d–f**, **n**, **o**) outline the lesion core. **n**, **o** denote paired images. Cell nuclei are labeled DAPI. All images show sagittal sections. Images are representative of two independent experiments. Source data and statistical test results are provided as a Source Data file.

From 5 dpi to 14 dpi the number of recombined PDGFRβ[+] stromal cells decreased while the scar dramatically compacted, resulting in a higher cell density in the lesion core (Fig. 3s–u). At both time points investigated, around 60% of the recombined cells that expressed PDGFRβ were found away from the blood vessel wall in the lesion core (Fig. 3v, w). In addition, we observed that type A pericyte-derived cells contributed to ~70% of all scar-forming PDGFRβ[+] stromal cells after a large cortical stab wound (Fig. 3x).

We conclude that larger mechanical injuries generate more robust fibrotic scarring when compared to smaller injuries. Fibrotic scar tissue formation with substantial numbers of PDGFRβ[+] stromal cells is only triggered by large mechanical injuries to the cerebral cortex. In line with this, we show that the majority of all scar-forming PDGFRβ[+] stromal cells generated after large cortical stab lesions descend from type A pericytes.

Similar sized, large cortical and cortical-striatal lesions resulted in a comparable stromal cell density after scar condensation at 14 dpi (Figs. 2t and 3u). However, the initial recruitment of type A pericytes was stronger after cortical stab lesions, which developed larger fibrotic lesion cores when compared to cortico-striatal stab wounds, as observed at 5 dpi. This initial difference was compensated by a more efficient resolution of fibrotic cells during scar maturation after cortical wounds compared to cortico-striatal stab lesions (Fig. 2g, h and 3s, t).

**Type A pericytes contribute to fibrotic scar formation following experimental autoimmune encephalomyelitis.** Next, we explored the contribution of type A pericytes to fibrotic scar tissue formation after experimental autoimmune encephalomyelitis (EAE), a widely accepted model of demyelinating diseases, such as MS (Fig. 4). Because tamoxifen diminishes the clinical severity of EAE[20], we extended the clearing time after tamoxifen-induced recombination to 50 days, before inducing EAE in adult GLAST-*CreER^T2;R26R-EYFP* transgenic mice (Fig. 4a). For EAE induction, animals were immunized with myelin oligodendrocyte glycoprotein (MOG) peptide emulsified in Complete Freund's Adjuvant (CFA) and received pertussis toxin. Animals receiving CFA and/or pertussis toxin alone but not immunized with the MOG peptide were used as control. Daily evaluation of clinical symptoms and weight confirmed that the animals manifested signs of chronic disease-related neurological deficits, starting 10 days after immunization (Fig. 4b, c).

All animals were sacrificed at 30 days post EAE induction, during the chronic phase of the disease. EAE led to widespread demyelination and scar formation throughout the spinal cord, particularly evident in the ventral and ventrolateral white matter of thoracic and lumbar spinal segments (Fig. 4d–g). We did not observe weight loss, disease symptoms or myelin lesions in control animals (Fig. 4b–d, h). In all demyelinated lesions we found densely packed recombined cells intermingling with CD45[+] immune cells, including infiltrating CD3[+] T-lymphocytes (Fig. 4i–k). Recombined cells expressed the stromal marker PDGFRβ and a fraction no longer associated with the vascular wall (Fig. 4l–o). In addition, we found that recombined cells proliferated both in association with the blood vessels and at a distance to them, expressed the (myo)fibroblast markers αSMA and vimentin and were surrounded by ECM enriched in fibronectin and collagen I (Fig. 4p–t). EAE led to a robust increase in the number and density of type A pericyte-derived (EYFP[+]) cells (Fig. 4u, v). Of all EYFP[+] cells that expressed PDGFRβ, around 30 % dissociated from the vascular wall (Fig. 4w, x). Type A pericyte-derived cells accounted for more than 90% of all the PDGFRβ[+] stromal cells found in demyelinated lesions (Fig. 4y).

In contrast to the spinal cord crush and stab lesions described before, reactive astrocytes did not sharply wall off the entire lesion, but were instead intermingled with recombined stromal cells within the lesion (Fig. 4e, f). We conclude that fibrotic scar tissue is formed after EAE and that nearly all PDGFRβ[+] stromal cells are derived from type A pericytes.

**Type A pericytes contribute to stromal cells following ischemic lesions to the brain.** We continued to explore whether fibrotic scar tissue formation is preserved in other types of CNS lesions, and investigated the contribution of type A pericytes to an ischemic lesion to the brain (Fig. 5). We induced experimental stroke by a 35 min occlusion of the middle cerebral artery (MCA), a procedure that generated an ischemic lesion with neuronal loss primarily in the striatum, sparing the cerebral cortex[13,21] (Fig. 5c).

Type A pericytes were fate mapped as described above for spinal cord injury and traumatic brain injury. At all time points investigated, we observed an increase in the number of lineage traced type A pericytes co-expressing EYFP and PDGFRβ, compared to the contralateral stroke side (Fig. 5a, b, d–q, Supplementary Fig. 8h–j). EYFP[+] stromal cells were interspersed

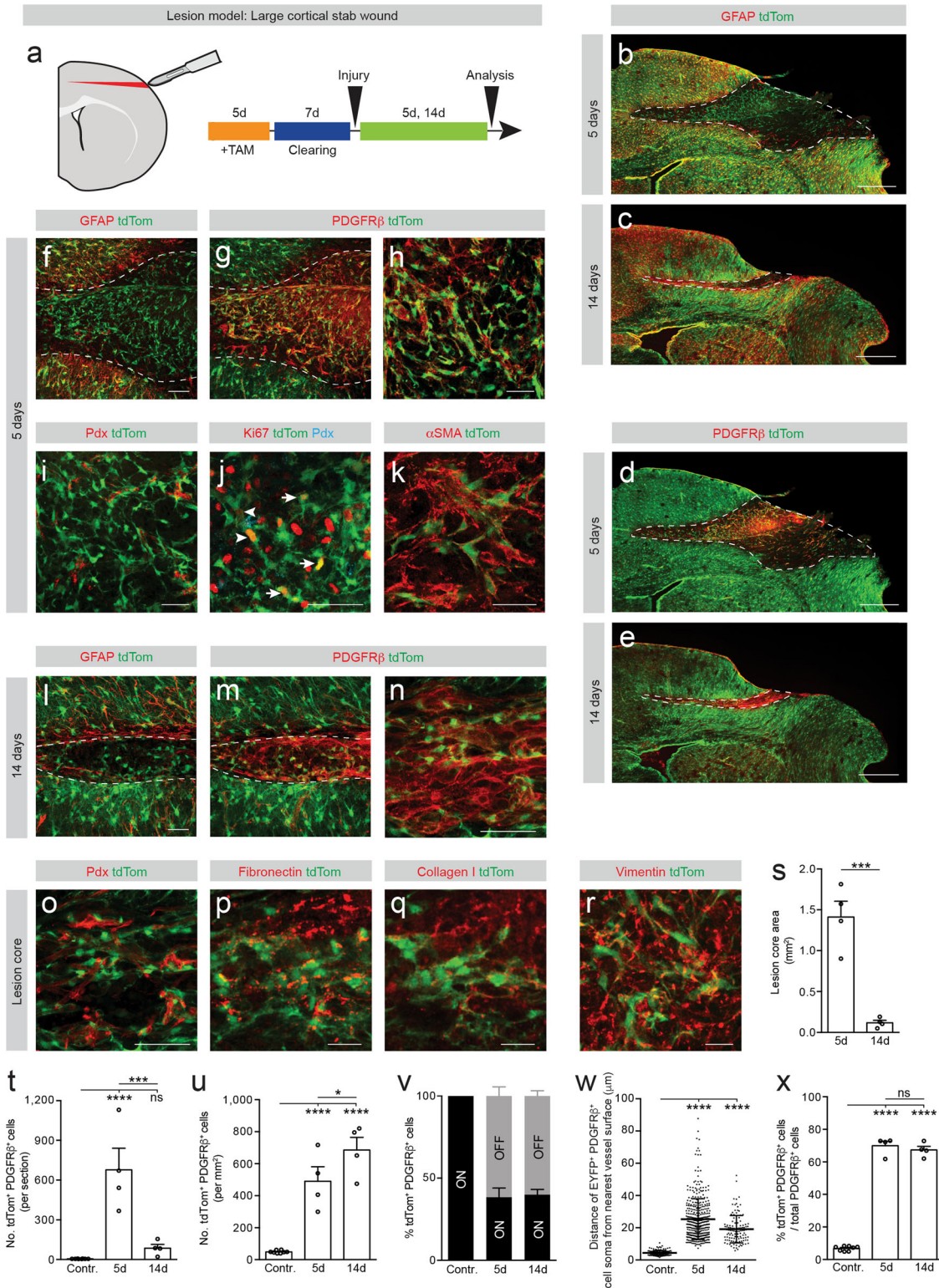

among CD45+ immune cells in the lesion (Fig. 5k, l, p, q). In sharp contrast to spinal cord injury, large brain stab lesions and EAE, the vast majority of PDGFRβ+ stromal cells, including nearly all EYFP+ cells that expressed PDGFRβ, remained in contact with the blood vessel wall at all time points investigated (Fig. 5i–w, Supplementary Fig. 8h–j). Nonetheless, we detected a small, but significant increase in the mean distance of the EYFP+ cells' soma to the nearest vessel surface, reflecting an altered

morphology and looser attachment of type A pericytes and progeny to the blood vessel wall in response to stroke, when compared to the contralateral control side (Fig. 5x). Recombined cells proliferated, were encased by fibronectin and collagen I ECM, but did not express αSMA (Fig. 5r–u), indicating these cells did not transition into (myo)fibroblasts.

The number of recombined PDGFRβ+ stromal cells that had drastically increased in the ischemic lesion core at 5 days after

**Fig. 3 Type A pericytes are the main source of stromal fibroblasts that form fibrotic scar tissue after large stab lesions restricted to the cerebral cortex. a** Lesion model and experimental timeline. Red represents the lesion. Distribution of tdTom$^+$ cells at 5 dpi (**b, d**) and 14 dpi (**c, e**). Type A pericyte-derived cells (tdTom$^+$PDGFRβ$^+$ cells) cluster in the lesion core (dashed lines), bordered by a partially recombined GFAP$^+$glial scar. **f–i** The lesion core (GFAP$^-$ area; outlined by dashed lines) is filled with tdTom$^+$PDGFRβ$^+$ cells (**f–h**) located outside the vascular wall (podocalyxin$^+$, Pdx) (**i**) at 5 dpi. TdTom$^+$ cells proliferate (Ki67$^+$) while attached to the blood vessel wall (arrowheads) and away from it (arrows) (**j**) and express the (myo)fibroblast marker αSMA (**k**) at 5 dpi. **l–o** TdTom$^+$ cells retain PDGFRβ$^+$ expression and occupy the lesion core (outlined by dashed lines) surrounded by GFAP$^+$ glial cells at 14 dpi (**l–n**). A fraction of tdTom$^+$ cells no longer associates with the vascular wall (**o**). **p–r** TdTom$^+$ cells are surrounded by fibronectin$^+$ (**p**) and collagen I$^+$ (**q**) ECM and express vimentin (**r**) at 14 dpi and 5 dpi, respectively. Quantification of the lesion core area (**s**), number of tdTom$^+$PDGFRβ$^+$ cells per section (**t**), and density of tdTom$^+$PDGFRβ$^+$ cells (**u**) overtime. **v** Percentage of tdTom$^+$ cells that express PDGFRβ (tdTom$^+$PDGFRβ$^+$ cells) associated with (ON vessel) or located outside (OFF vessel) the vascular wall. **w** Distance of tdTom$^+$PDGFRβ$^+$ cells ON vessel (contralateral cortex) or OFF vessel (after injury) from the nearest vessel surface. Each dot represents one cell. **x** Percentage of PDGFRβ$^+$ cells that express tdTom out of total PDGFRβ$^+$ cells. TAM tamoxifen; Scale bars: 500 μm (**b–e**), 100 μm (**f, g**), 50 μm (**h–o**) and 20 μm (**p–r**). Data shown as mean ± s.e.m. $n = 8$ (Contralateral), $n = 4$ (5d), and $n = 4$ (14d) animals in (**s–v, x**). $n = 131$ (Contralateral), $n = 401$ (5d), and $n = 117$ (14d) cells examined over 4 animals in (**w**). ns non-significant; *$p < 0.05$, ***$p < 0.001$, ****$p < 0.0001$ by two-sided, unpaired Student's $t$ test in (**s**), One-Way ANOVA followed by Holm–Sidak post-hoc test in (**t, u, x**) and Kruskal–Wallis test followed by Dunn's post-hoc test in (**w**). **b, d, c, e, f, g, l, m, n, o** denote paired images. Images show coronal sections. Images are representative of two independent experiments. Source data and statistical test results are provided as a Source Data file.

stroke declined over time along with condensation of the lesion, keeping the cell density stable (Fig. 5g, h, v). While in the contralateral, non-ischemic side, type A pericytes accounted for ~10% of all PDGFRβ$^+$ cells, 40–60% (depending on the time point) of the PDGFRβ$^+$ cells covering blood vessels in the ischemic lesion core were recombined and, therefore, originated from type A pericytes (Fig. 5y).

Interestingly, in cases in which the MCA was occluded for 45 min, resulting in an ischemic lesion core that extended from the striatum into the cortex (cortico-striatal stroke)[21] (Supplementary Fig. 11a), many PDGFRβ$^+$ cells could be observed at a distance from the blood vessel wall, primarily in the cortex but also in the striatum. The vast majority of these PDGFRβ$^+$ stromal cells were recombined and, therefore, originated from type A pericytes (Supplementary Fig. 11b–o).

We then asked whether the anatomical location of the stroke and environmental differences between the cortex and the striatum influence the generation of fibrotic scar tissue and type A pericyte detachment from the blood vessel wall. For this, we modeled ischemic lesions restricted to the cerebral cortex by a permanent occlusion of the distal portion of the MCA[22] (Fig. 6a) and lineage traced type A pericytes as aforementioned.

At both 5 days and 14 days after stroke, we found an increased number and density of recombined PDGFRβ$^+$ stromal cells compared to the contralateral stroke side (Fig. 6b–u). In contrast to striatal ischemic strokes, 60–70% of the recombined cells that expressed PDGFRβ were no longer associated to the vascular wall in the lesion core (Fig. 6i, o, v, w). Recombined cells proliferated while associated with vascular wall and away from it, were bordered by dense fibronectin$^+$ and collagen I$^+$ ECM and expressed the (myo)fibroblast markers αSMA and vimentin (Fig. 6j, k, p–r). After cortical ischemic stroke, type A pericyte-derived cells contributed ~80% to all scar-forming PDGFRβ$^+$ stromal cells at all time points investigated (Fig. 6x). Recombined reactive astrocytes, co-expressing EYFP or tdTom and GFAP (but not PDGFRβ), could be observed in the lesion rim as part of the glial scar across all ischemic stroke models (Figs. 5d–f, 6b, c, f, l and Supplementary Figs. 7g, h and 11b, i).

In cortical lesions, type A pericyte-derived fibroblasts aligned with GFAP$^+$ astrocytes along the glial-fibrotic lesion border (Fig. 6l, m and Supplementary Fig. 7d–f), as observed after spinal cord injury (Fig. 1n, o and Supplementary Fig. 7a–c). In contrast, in striatal lesions, where type A pericyte-derived cells remained at the vessel wall, we observed no formation of an aligned lesion border (Fig. 5m, n and Supplementary Fig. 7g–i), indicating that stromal fibroblast-reactive astrocyte contact is necessary for compact border formation.

Collectively, we observed that in ischemic lesions confined to the striatum most of the PDGFRβ$^+$ stromal cells are associated with the vascular wall, while after cortical ischemic lesions a large number of PDGFRβ$^+$ stromal cells can also be found in distance to the vessel wall. Type A pericytes contributed substantially to both PDGFRβ$^+$ stromal populations. In contrast to striatal ischemic lesions, type A pericytes and progeny could be observed away from the blood vessel wall in the striatum after cortico-striatal ischemic lesions, hinting that environmental factors associated with the cerebral cortex and/or adjacent brain structures may impact on type A pericyte recruitment from the vessel wall in the striatum.

**Type A pericytes contribute to brain tumor stroma.** Next, we explored type A pericyte contribution to tumor stroma in the murine glioma 261 (GL261) orthotopic glioma model[23], which mimics human GBM, the most frequent and aggressive malignant brain tumor in adults[24]. For this, we injected mouse GL261 cells in the striatum of adult GLAST-*CreER$^{T2}$*;*R26R-EYFP* mice after recombination and tamoxifen clearing (Fig. 7a). Animals developed visible tumors with abnormal and enlarged vasculature, 3 weeks after inoculation (Fig. 7b, c). The tumor mass covered most of the striatal area and was surrounded by GFAP$^+$ astrocytes that were partially recombined (Fig. 7d, e).

When compared to the contralateral side to the tumor, the total number and density of type A pericytes in the tumor stroma was reduced, in line with an overall reduction of pericyte coverage in tumor-associated vessels[25,26]. Nonetheless, we observed that about 30 ± 5% of all type A pericyte-derived EYFP$^+$ cells were located outside the blood vessel wall and dispersed among tumor cells (Fig. 7f–h, o–r). In addition, we found that EYFP$^+$ cells proliferated in contact and away from the vessel wall, expressed αSMA and vimentin and were surrounded by fibronectin and collagen I- enriched ECM (Fig. 7 i–n).

Overall, type A pericyte-derived cells accounted for 13 ± 2% of all PDGFRβ$^+$ cells in the tumor mass (Fig. 7s). In summary, we found that type A pericytes contribute to tumor-associated PDGFRβ$^+$ stromal cells, however not as the main source.

**Type A pericytes are required for extracellular matrix deposition after CNS lesions.** The difference between striatal and cortical ischemic lesions allowed us to investigate whether the variable distribution of type A pericyte-derived stromal cells in relation to the blood vessel wall influences the pathological outcome of tissue fibrosis, such as deposition of fibrotic ECM proteins.

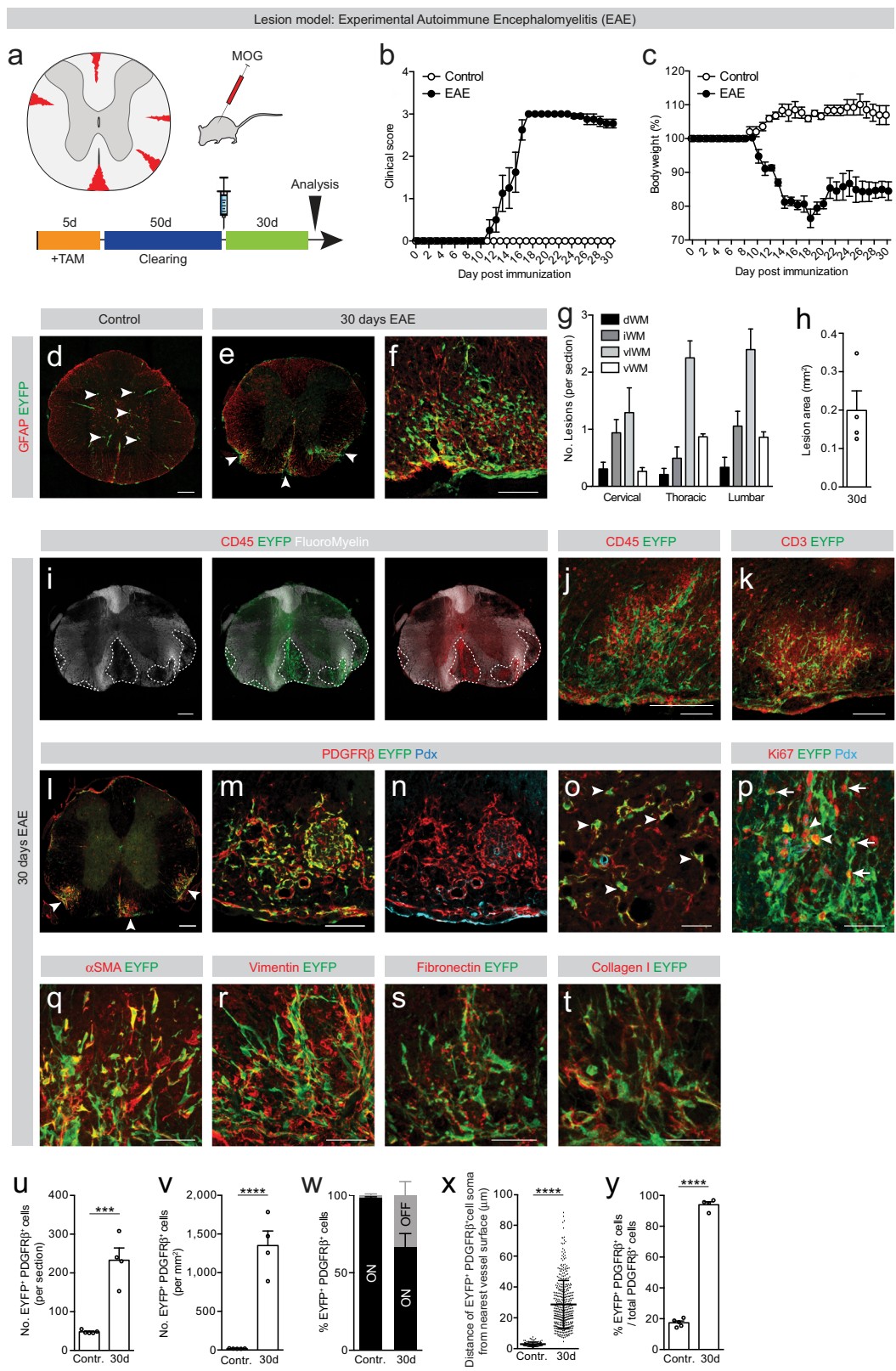

In striatal lesions, in which type A pericytes remained attached to the vascular wall (Fig. 8a), we observed a slight but not significant increase in the deposition of fibronectin and collagen I in the ischemic lesion core, compared to the contralateral stroke side. On the contrary, cortical ischemic lesions, with widespread, non-vessel-associated type A pericyte-derived fibroblasts, led to a robust increase in deposition of fibronectin and collagen I in comparison to striatal strokes and to the contralateral stroke side. These results indicate that detachment of type A pericyte-derived cells from the vascular wall, which leads to the generation of stromal fibroblasts, is required for the deposition of fibronectin and collagen I ECM proteins (Fig. 8b–e).

To validate this finding in an independent lesion model in the brain, we experimentally limited the generation of type A pericyte

**Fig. 4 Type A pericytes are the main source of stromal fibroblasts that form fibrotic scar tissue after experimental autoimmune encephalomyelitis.**
**a** MOG$_{35-55}$-induced EAE model and experimental timeline. Red represents the lesions. Clinical scores (**b**) and body weight curves relative to day 0 (pre-immunization) (**c**) for EAE and control animals. Distribution of EYFP$^+$cells (arrowheads) in the thoracic spinal cord of control (**d**) and EAE animals (**e**).
**f** Reactive astrocytes (GFAP$^+$) intermingle with EYFP$^+$ cells. **g** Distribution and number of lesions across the dorsal (dWM), intermediate (iWM), ventrolateral (vlWM), and ventral (vWM) white matter. **h** Area occupied by EAE lesions. EYFP$^+$ cells intermingle with CD45$^+$ immune cells (**i**, **j**), including infiltrating CD3$^+$ T-lymphocytes (**k**) in demyelinated EAE lesions, identified as regions lacking fluoromyelin signal (encircled by dashed lines). EYFP$^+$PDGFRβ$^+$ cells accumulate in EAE scars (**l**, **m**; arrowheads) and a fraction is located in distance to the vascular wall (podocalyxin$^+$, Pdx; arrowheads) (**n**, **o**). EYFP$^+$ stromal cells proliferate (Ki67$^+$) in association with the vascular wall (arrowheads) and away from it (arrows) (**p**), express the (myo) fibroblast markers αSMA (**q**) and vimentin (**r**) and are surrounded by ECM enriched in fibronectin (**s**) and collagen I (**t**). Number (**u**) and density (**v**) of EYFP$^+$PDGFRβ$^+$ cells in control and EAE animals. **w** Percentage of EYFP$^+$PDGFRβ$^+$ cells associated with (ON vessel) or located outside (OFF vessel) the vascular wall. **x** Distance of EYFP$^+$PDGFRβ$^+$ cells ON vessel (EAE control) or OFF vessel (EAE) from the nearest vessel surface. Each dot represents one cell. **y** Percentage of PDGFRβ$^+$ cells that express EYFP out of total PDGFRβ$^+$ cells. TAM tamoxifen, MOG myelin oligodendrocyte glycoprotein. Scale bars: 200 μm (**d**, **e**, **i**, **l**), 100 μm (**f**, **j**, **k**, **m**, **n**), and 50 μm (**o–t**). Data shown as mean ± s.e.m. $n = 5$ (Control) and $n = 4$ (30d) animals in (**b**, **c**, **g**, **h**, **u–w**, **y**); $n = 118$ (Control) and $n = 367$ (30d) cells examined over 4 animals in (**x**). ***$p < 0.001$, ****$p < 0.0001$ by two-sided, unpaired Student's $t$ test in (**u**, **v**, **y**) and two-sided Mann–Whitney test in (**x**). **i**, **m**, **n** denote paired images. All images show coronal sections. Images are representative of two independent experiments. Source data and statistical test results are provided as a Source Data file.

progeny in the context of cortico-striatal stab lesions. Cortico-striatal stab lesions induced proliferation in type A pericytes and led to a significant generation of progeny (Fig. 2). We therefore employed a cell-specific and inducible genetic strategy to inhibit the generation of progeny by type A pericytes, via specifically blocking injury-induced proliferation. This was achieved by generating new transgenic mice that, in addition to carrying the GLAST-*CreER$^{T2}$* and *R26R-EYFP* alleles, were homozygous for *H-Ras* and *N-Ras* null alleles and for floxed *K-Ras* alleles[27], hereafter referred to as GLAST-Rasless-EYFP mice (Fig. 8f). Both Cre$^+$ and Cre$^{WT}$ animals were treated with tamoxifen prior to injury, but Cre$^{WT}$ animals did not undergo genetic recombination and served as control.

As previously established after spinal cord injury[3,6], tamoxifen-induced genetic recombination in Cre$^+$ GLAST-Rasless-EYFP animals prior to stab injury resulted in the loss of all 3 *Ras* genes, important players in mitogenic signaling and cell cycle progression, specifically in type A pericytes and conferred less proliferative capacity to this population of cells upon injury when compared to control animals, as assessed 5 dpi. Interestingly, reduced proliferation of type A pericytes was paralleled by a higher proportion of type A pericyte-derived cells remaining attached to the vasculature (Supplementary Fig. 12a–f), suggesting that proliferation is required for pericyte dissociation from the vascular wall. Although we detected a decrease in the percentage of proliferating recombined GFAP$^+$ astrocytes, it did not lead to a significant change in the bulk of reactive GFAP$^+$ astrocytes participating in glial scar formation in comparison to control animals (Supplementary Fig. 12g–j).

We next compared deposition of fibronectin and collagen I ECM proteins in Cre$^{WT}$ versus Cre$^+$ GLAST-Rasless-EYFP animals at 14 days after cortico-striatal stab lesions. As expected, Cre$^{WT}$ control animals, where type A pericytes proliferate and leave the blood vessel wall extensively, developed a dense fibrotic lesion core enriched in PDGFRβ-expressing stromal cells and widespread deposition of fibronectin and collagen I-rich ECM. On the contrary, Cre$^+$ animals, in which type A pericyte proliferation was reduced and a higher percentage of cells remained attached to the blood vessel wall, presented significant attenuation of fibrotic scar tissue formation, revealed as smaller lesions with decreased PDGFRβ$^+$ scarring compared to control animals. In these animals, we observed reduced deposition of fibronectin and collagen I compared to control animals (Fig. 8g–h and Supplementary Fig. 12k). Interestingly, in Cre$^{WT}$ control animals fibronectin and collagen I signals localized to the brain surface (meninges) and within the brain parenchyma. Conversely, in Cre$^+$ animals, fibronectin and collagen I signals remained

predominantly at the brain surface, while their deposition was significantly reduced within the brain parenchyma (Fig. 8g–h). This suggests that type A pericytes contribute to fibrotic ECM deposition within the brain parenchyma and reduction of type A pericyte-derived scarring does not substantially impact on meningeal ECM deposition. Taken together, these results demonstrate that the extent of type A pericyte progeny located outside the vascular wall determines the magnitude of fibrotic ECM deposition after brain lesions, supporting previous observations after spinal cord injury[3,6].

**Fibrotic tissue forms after diverse types of CNS lesions in humans.** The experiments above show that GLAST$^+$ type A pericytes are the main source of PDGFRβ-expressing stromal cells that constitute fibrotic scar tissue following CNS lesions in the mouse. We next sought to determine whether similar GLAST-expressing perivascular cells exist in the human CNS (Fig. 9).

PDGFRβ-expressing perivascular cells, including pericytes with characteristic protruding ovoid cell bodies and processes encircling the endothelial wall of small caliber vessels, could be readily identified associated with the vasculature in post-mortem healthy tissue from human spinal cord (Fig. 9a–d) and occipital cortex (Fig. 9e–h). Interestingly, we found numerous GLAST$^+$PDGFRβ$^+$ perivascular cells, defined as blood vessel-associated cells positive for *PDGFRB* and *SLC1A3* mRNA signals, suggesting that the human vasculature may contain cells that are similar to type A pericytes in the mouse[3]. In line with this, we could also identify GLAST$^-$PDGFRβ$^+$ perivascular cells and parenchymal (non-blood vessel-associated) GLAST$^+$PDGFRβ$^-$ cells, likely representing astrocytes (Fig. 9c, g).

We next investigated the composition of scar tissue that forms following traumatic spinal cord injury in post-mortem spinal cord tissue from six patients with massive compression or contusion/cyst type injuries in the cervical region, who survived for intervals up to 61 days after injury (Supplementary Table 1). In all cases we found regions of non-neural scar tissue enriched in PDGFRβ$^+$ cells, often organized in clusters, and bordered by reactive glia (Fig. 10a, b). The areas of perivascular fibrosis were much larger compared to the surrounding astroglial scar. Significant numbers of PDGFRβ$^+$ cells within the lesion core were found at a distance from a blood vessel wall (Fig. 10b inset).

Likewise, we detected increased perivascular aggregates of PDGFRβ$^+$ cells accompanied by gliosis in post-mortem spinal cord samples from six patients diagnosed with chronic MS (secondary progressive MS type), who survived for 17–56 years after the disease onset (Supplementary Table 2, Fig. 10c–f). In active and chronic active MS lesions PDGFRβ$^+$ stromal cells were

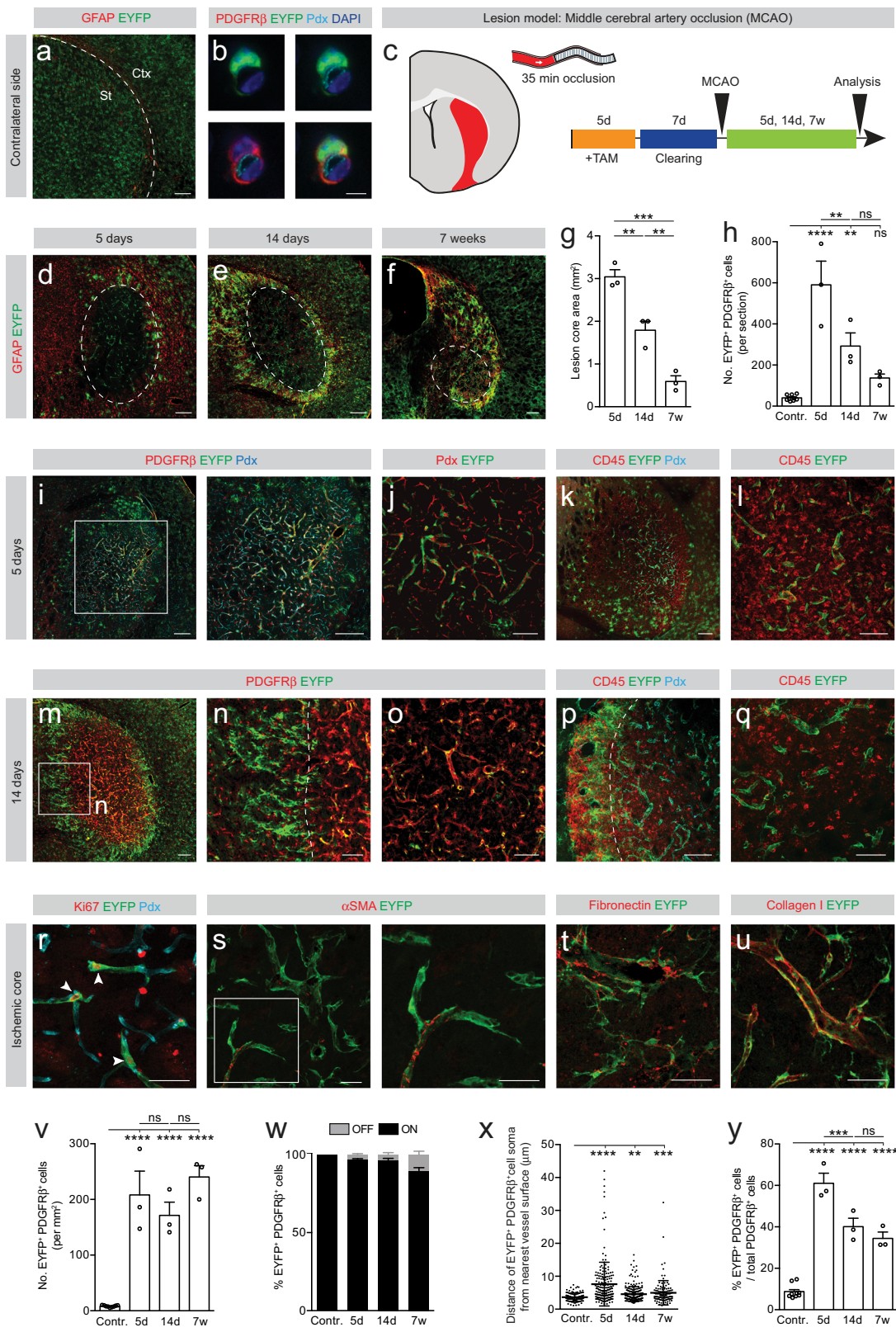

organized in multiple perivascular rings, and a substantial number of cells no longer associated with the blood vessel wall (Fig. 10e). Perivascular fibrosis was closely associated with demyelinated white matter regions.

Next, we investigated human post-mortem brain tissue obtained from 4 individuals that had a history of a supratentorial territorial or lacunar stroke, involving cerebral cortical

and subcortical areas, and who died weeks to months after infarction (Supplementary Table 3). We found that the density of PDGFRβ⁺ cells was elevated in most of the subacute/chronic lesions when compared to surrounding healthy tissue. PDGFRβ⁺ stromal cells within the ischemic lesion core mainly associated with the vasculature after subcortical infarcts (Fig. 10g, h), as observed following ischemic damage confined to the striatum in

**Fig. 5 Type A pericytes remain associated with the vascular wall following ischemic stroke confined to the striatum. a** Type A pericytes and parenchymal astrocytes are recombined (EYFP+) throughout the cortex (Ctx) and striatum (St) of GLAST-*CreER*T2;*R26R-EYFP* mice. Dashed line separates cortex from striatum. **b** Type A pericyte (EYFP+) encircling the endothelial tube (podocalyxin+, Pdx) and expressing the pericyte marker PDGFRβ in the striatum. **c** Lesion model and experimental timeline. Red represents the lesion. Distribution of EYFP+ cells at 5 days (**d**), 14 days (**e**), and 7 weeks (**f**) following ischemia. Type A pericyte-derived cells (EYFP+GFAP– cells) occupy the ischemic lesion core (outlined by dashed lines), surrounded by a partially recombined GFAP+ glial scar. Lesion core area (**g**) and number of EYFP+PDGFRβ+ cells per section (**h**). EYFP+PDGFRβ+ cells in the ischemic core (**i**) do not leave the vascular wall (Pdx+) (**j**). Magnified boxed region in (**i**) shown on the right. EYFP+ cells disperse among CD45+ immune cells at 5 days (**k, l**) and 14 days (**p, q**) after stroke. EYFP+PDGFRβ+ cells occupy the lesion core, bordered by partly recombined EYFP+PDGFRβ− glial cells at 14 days after stroke (**m–o**). **k, p, m** show overviews of the lesion and **l, n, o, q** close-ups of the ischemic core. Dashed lines in (**n, p**) demarcate the glial-fibrotic lesion border. EYFP+ cells proliferate (arrowheads in (**r**) point at Ki67+EYFP+ cells) and do not express αSMA (**s**) at 5 days after stroke. Magnified boxed region in (**s**) shown on the right. EYFP+ cells are encased by fibronectin (**t**) and collagen I (**u**) ECM at 14 days after stroke. **v** Density of EYFP+PDGFRβ+ cells overtime. **w** Most EYFP+PDGFRβ+ cells remain associated with the vascular wall (ON vessel) after striatal stroke. **x** Distance of EYFP+PDGFRβ+ cells from the nearest vessel surface in the contralateral striatum and after stroke. Each dot represents one cell. **y** Percentage of PDGFRβ+ cells that express EYFP out of total PDGFRβ+ cells. TAM tamoxifen; Scale bars: 200 μm (**a, d–f, i, k, m**), 100 μm (close up **i, j, l, n–p**), 50 μm (**q–u**) and 5 μm (**b**). Data shown as mean ± s.e.m. $n = 9$ (Contralateral), $n = 3$ (5d), $n = 3$ (14d) and $n = 3$ (7w) animals in (**g, h, v, w, y**); $n = 127$ (Contralateral), $n = 195$ (5d), $n = 248$ (14d) and $n = 122$ (7w) cells examined over 3 animals in (**x**). ns non-significant; **$p < 0.01$, ***$p < 0.001$, ****$p < 0.0001$ by One-Way ANOVA followed by Holm–Sidak post-hoc test in (**g, h, v, y**) and Kruskal–Wallis test followed by Dunn's post-hoc test in (**x**). Cell nuclei are labeled with DAPI. All images show coronal sections. Images are representative of two independent experiments. Source data and statistical test results are provided as a Source Data file.

the mouse (Fig. 5). The lesion core was flanked by a GFAP+ glial scar.

Finally, we investigated brain tissue samples from three patients with aggressive grade IV GBM tumors (Supplementary Table 4). All samples presented morphologically abnormal and disorganized vasculature, as opposed to the thin-wall blood vessels of non-malignant adjacent brain tissue. We observed general aberrant hypervascularization within the tumor mass and increased density of PDGFRβ+ stromal cells across all three patient samples analyzed. Nonetheless, even within the same tumor sample, we detected heterogeneity in the density of blood vessels (Supplementary Table 4 and Fig. 10i). Although malformed blood vessels often presented thick-wall and irregular shape, we found that most of the PDGFRβ+ cells within the tumor stroma remained associated with the vasculature (Fig. 10j).

In summary, we observed accumulation of PDGFRβ+ stromal cells across all pathological tissues investigated, but their distribution varied depending on the type of lesion. After spinal cord injury and MS, we found accumulation of PDGFRβ+ stromal cells in extravascular positions, while we registered an increased number of cells associated with the vasculature after subcortical ischemic stroke and in GBM tumors.

## Discussion

We identified a small subset of perivascular cells, termed type A pericytes, as the main source of scar-forming PDGFRβ+ stromal cells, contributing to fibrotic tissue generation across a vast number of lesions in the brain and spinal cord. We found that the magnitude of fibrotic scar tissue generation and the distribution of PDGFRβ+ stromal cells depend on the extent and type of lesion (Supplementary Fig. 13).

Importantly, perivascular cells with an analogous marker profile as type A pericytes are present along the human brain and spinal cord vasculature and fibrotic scarring is similar in human pathological tissue and corresponding mouse models. Our findings suggest that fibrotic scarring by a small subset of perivascular cells, defined as type A pericytes, is an evolutionary conserved mechanism for scar formation in the CNS.

Non-penetrating spinal cord injury, in which the dura mater remains relatively intact, limits the invasion of dura-derived meningeal fibroblasts into the lesion. Penetrating, and non-penetrating spinal cord injury led to similar recruitment of PDGFRβ+ stromal cells from the vascular wall, and consequent fibrotic scar tissue formation (Fig. 1 and Supplementary Fig. 5). Nearly all PDGFRβ+ stromal cells derived from type A pericytes, indicating that dura-derived fibroblasts

are not the major source of stromal cells. In humans, we observed extensive fibrotic scar tissue formation enriched in PDGFRβ+ stromal cells at subacute stages after massive compression, as well as contusive injuries. These results, in conjunction with previous reports acknowledging the accumulation of inflammatory cells and ECM deposits[1,28–31], show cumulative evidence of fibrotic scar tissue generation after spinal cord injury in humans. In all 6 cases studied, regions of non-neural tissue were much larger compared to the astrocyte scar border (Fig. 10a, b).

EAE lesions generated substantial fibrotic scar tissue enriched in PDGFRβ+ stromal cells accumulating outside the blood vessel wall (Fig. 4). Again, we found that nearly all PDGFRβ+ stromal cells derived from type A pericytes. In spinal cords of individuals with active MS lesions, we found PDGFRβ+ stromal cells accumulating outside, but near the vascular wall (Fig. 10c–f), as reported in MS brain lesions[32].

Previous studies employed a Col1α1-GFP reporter line, which labels perivascular fibroblasts primarily located around large diameter blood vessels, and described an increase in Col1α1-producing cells following contusive spinal cord injury[33] and EAE[34] in the mouse. Since Col1α1-GFP only reflects Col1α1 transcriptional activity, and no fate mapping of Col1α1-expressing cells had been employed, the contribution of Col1α1+ cells to scar-forming stromal fibroblasts remained speculative. A recent study lineage traced Col1α2-expressing cells and showed that CNS perivascular fibroblasts contributed to fibrotic scar formation following EAE[35]. Single-cell sequencing analyses of vascular cells in the mouse brain revealed that in addition to pericytes, GLAST (*Slc1a3*) is expressed by a subset of perivascular fibroblasts[36], suggesting that type A pericytes could be more heterogeneous than previously anticipated. Future studies exploring type A pericyte heterogeneity at the single-cell level will be required to clarify this matter.

In comparison to spinal cord injury and EAE, stab lesions to the mouse brain showed similar fibrotic scar tissue formation only after large cortical and cortico-striatal injuries, but not after small injuries, restricted to the cortex (Figs. 2 and 3 and Supplementary Fig. 9), indicating that larger mechanical injuries to the brain are required for the generation of extensive fibrotic scar tissue. Nearly all PDGFRβ+ stromal cells derived from type A pericytes after cortical and cortico-striatal stab lesions.

A recent study using a *Tbx18-CreER*T2 line to fate map pericytes after discrete cerebral cortex stab lesions led to the generalized conclusion that pericytes do not contribute to fibrotic scar tissue after brain injury[37]. Since no substantial fibrotic tissue

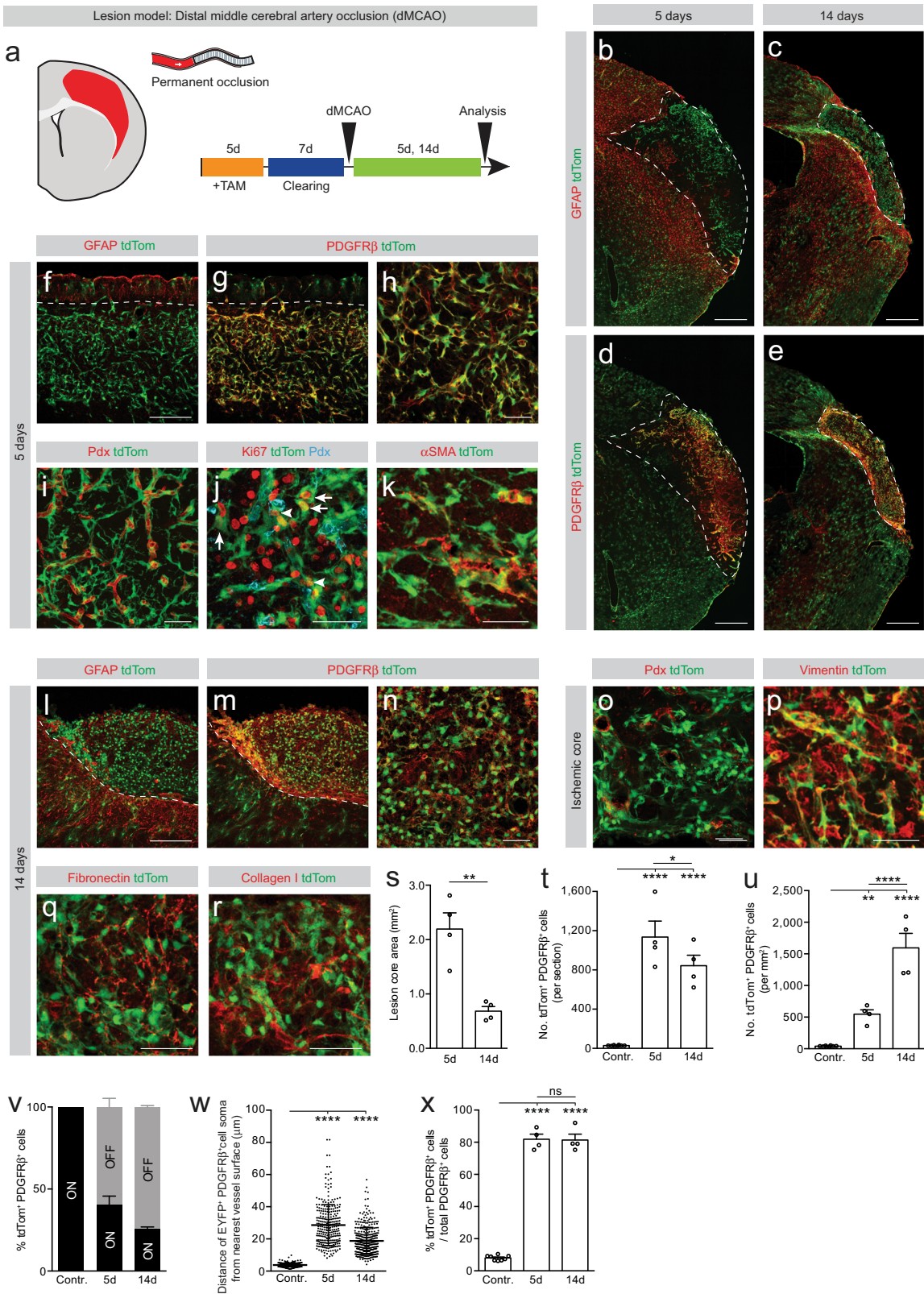

is generated after small cortical lesions it is expected to observe no recruitment of Tbx18-lineage cells and therefore our results employing a similar injury paradigm are in line with their findings. We show that lineage-traced type A pericytes as well as their progeny express *Tbx18*, suggesting that Tbx18-expressing perivascular cells contribute to CNS fibrosis in lesion models that generate substantial fibrotic scar tissue.

MCAO-induced hypoxic lesions restricted to the striatum showed an increase of type A pericyte-derived PDGFRβ[+] stromal cells in association with the vascular wall, but little to no stromal cells outside the vessel wall (Fig. 5). In sharp contrast, many type A pericyte-derived PDGFRβ[+] stromal cells could be found outside the vessel wall after cortical and cortico-striatal ischemic lesions (Supplementary Figs. 6 and 11). Interestingly, a large

**Fig. 6 Type A pericytes are the main source of stromal fibroblasts that form fibrotic scar tissue following cortical ischemic stroke. a** Lesion model and experimental timeline. Red represents the lesion. Distribution of recombined cells (tdTom+) at 5 days (**b**, **d**) and 14 days (**c**, **e**) after stroke. Type A pericyte-derived cells (TdTom+PDGFRβ+) occupy the ischemic lesion core (outlined by dashed lines), flanked by a partly recombined GFAP+ glial scar. **f**–**o** The ischemic core is filled with tdTom+ cells that express PDGFRβ and delimited by GFAP+ glial cells at 5 days (**f**–**h**) and 14 days (**l**–**n**) after stroke. Dashed lines separate the glial (GFAP+) and fibrotic (tdTom+) scar compartments. A fraction of tdTom+ cells no longer associates with the vascular wall (labeled by podocalyxin, Pdx) (**i**, **o**). TdTom+ cells proliferate (Ki67+) while attached to the vascular wall (arrowheads) and away from it (arrows) (**j**) and express the (myo)fibroblast marker αSMA (**k**). TdTom+ cells express vimentin (**p**) at 5 days after stroke and are surrounded by fibronectin (**q**) and collagen I (**r**) at 14 days after stroke. Quantification of the lesion core area (**s**), number of tdTom+PDGFRβ+ cells per section (**t**) and density of tdTom+PDGFRβ+ cells (**u**). **v** Percentage of tdTom+PDGFRβ+ cells associated with (ON vessel) or located outside (OFF vessel) the vascular wall. **w** Distance of tdTom+PDGFRβ+ cells ON vessel (contralateral cortex) or OFF vessel (after stroke) from the nearest vessel surface. Each dot represents one cell. **x** Percentage of PDGFRβ+ cells that express tdTom out of total PDGFRβ+ cells. TAM tamoxifen; Scale bars: 500 μm (**b**–**e**), 200 μm (**f**, **g**, **l**, **m**) and 50 μm (**h**–**k**, **n**–**r**). Data shown as mean ± s.e.m. $n = 8$ (Contralateral), $n = 4$ (5d) and $n = 4$ (14d) animals in (**s**–**v**, **x**); $n = 252$ (Contralateral), $n = 286$ (5d) and $n = 360$ (14d) cells examined over 4 animals in (**w**). ns non-significant; *$p < 0.05$, **$p < 0.01$, ****$p < 0.0001$ by two-sided, unpaired Student's $t$ test in (**s**), One-Way ANOVA followed by Holm–Sidak post-hoc test in (**t**, **u**, **x**) and Kruskal–Wallis test followed by Dunn's post-hoc test in (**w**). **b**, **d**, **c**, **e**, **f**, **g**, **l**, **m** denote paired images. All images show coronal sections. Images are representative of two independent experiments. Source data and statistical test results are provided as a Source Data file.

number of type A pericytes and progeny could be observed away from the blood vessel wall in the striatum only when the ischemic lesion expanded from the striatum into the cerebral cortex, suggesting that environmental cues associated with the cerebral cortex and/or underlying corpus callosum white matter may influence type A pericyte recruitment from the vessel wall. These observations may explain previous results regarding pericyte detachment from the vascular wall[38,39] and suggest that the location and magnitude of the ischemic insult dictate the distribution of type A pericyte progeny and generation of stromal fibroblasts. In the lesion core of subcortical ischemic stroke lesions in humans, we found nearly all PDGFRβ+ stromal cells associated with the vascular wall (Fig. 10g, h).

In the GL261 murine glioma model, the vasculature is known to present low coverage by pericytes[26], including type A pericytes. However, despite their reduced number, we observed PDGFRβ+ stromal cells outside tumor vessels interspaced with glioma cells, partially derived from type A pericytes (Fig. 7). In human GBM tumors, most PDGFRβ+ cells remained in close association with the stroma vasculature (Fig. 10i, j).

A recent study investigates ECM deposition by pericytes after ischemic stroke. The authors use conventional, non-conditional, knock out (KO) mice for *Rgs5*, to alter the pericyte response to injury. They propose that reducing the number of pericytes that leave the blood vessel wall after ischemic stroke does not significantly impact on the deposition of collagen I and fibronectin[40]. Their interpretation is contrary to our observations, showing a minimal increase in ECM deposition after striatal ischemic lesions, where type A pericytes remain attached to the vascular wall, in contrast to abundant deposition of ECM in conjunction with type A pericyte recruitment after cortical ischemic lesions (Fig. 8a–e). Additionally, attenuation of type A pericyte-derived scarring results in decreased deposition of fibronectin and collagen I after brain stab wounds, showing that type A pericytes are required for ECM deposition (Fig. 8f–h), as observed previously after spinal cord injury[3,6]. These observations reveal that the number and location of type A pericytes and progeny in relation to the vascular wall influence the deposition of collagen I and fibronectin ECM proteins after injury. The different results by Roth et al. may be explained by the fact that deletion of *Rgs5* impacts not only on pericytes, but also astrocytes, as indicated by the reduced thickening of the vascular basal membrane and alteration of the glial scar in RGS5-KO compared to WT mice in response to stroke. Since the glial scar intimately borders the lesion core and there is crosstalk between reactive astrocytes and fibrotic cells (Supplementary Fig. 7), a less pronounced glial scar can influence fibrotic scar tissue generation and ECM deposition.

Pericytes have been shown to be the main contributors to ECM-producing (myo)fibroblasts observed in dermal, renal, hepatic and pulmonary fibrotic tissue[41–44], suggesting functional similarities among pericytes in the CNS and peripheral organs. Nonetheless, while pericyte heterogeneity has not been extensively addressed in the context of peripheral organ fibrosis, we define the origin of PDGFRβ-expressing stromal cells to a specific perivascular subset, only accounting for 10% of all CNS pericytes. This distinction is of special importance for the design of potential therapeutic strategies to improve axonal regeneration and functional recovery following CNS injury[6], as it may allow specific targeting of fibrotic pericytes without compromising the vast majority of pericytes and their functions (i.e. maintenance of the blood-brain/spinal cord barrier and blood flow regulation)[45,46].

## Methods

**Transgenic mice**. GLAST-*CreER*[T2] transgenic mice[12] were crossed to the *Rosa26-enhanced yellow fluorescent protein* (*EYFP*)[47] or *Rosa26-tdTomato*[48] Cre-reporter lines (obtained from the Jackson Laboratory, B6.129×1-Gt(Rosa)26Sor[tm1(EYFP)Cos]/J, JAX stock: 006148; B6.Cg-Gt(Rosa)26Sor[tm14(CAG-tdTomato)Hze]/J, JAX stock: 007914) to generate GLAST-*CreER*[T2];*R26R-EYFP* or GLAST-*CreER*[T2];*R26R-tdTomato* mice in which CreER[T2] is hemizygous and Rosa26-EYFP or Rosa26-tdTomato is either heterozygous or homozygous. For a set of experiments, GLAST-*CreER*[T2];*R26R-EYFP* were further crossed to Rasless mice[27] to generate GLAST-*CreER*[T2];Rasless;*R26R-EYFP* mice, hereafter referred to as GLAST-Rasless-EYFP mice. All Rasless mice used were homozygous for *H-Ras* and *N-Ras* null alleles and homozygous for floxed *K-Ras* alleles.

All animals were in a C57BL/6J genetic background and ≥8 weeks old at the onset of experiments. Animals were housed in group in a pathogen-free facility with controlled humidity (50 ± 15%) and temperature (22 ± 2 °C), with 12:12-h light:dark cycles and free access to food and water. All experimental procedures were performed in accordance to the Swedish and European Union guidelines and approved by the institutional ethical committees (Stockholms Norra Djurförsöksetiska Nämnd or Malmö/Lunds Djurförsöksetiska Nämnd).

**Genetic labeling of transgenic mice**. Genetic recombination in GLAST-*CreER*[T2];*R26R-EYFP* and GLAST-*CreER*[T2];*R26R-tdTomato* mice was induced by a daily intraperitoneal injection of 2 mg of tamoxifen (Sigma, 20 mg/ml in 1:9 ethanol:corn oil) for five consecutive days, as previously[3,6].

For experiments involving GLAST-Rasless-EYFP mice, recombination was induced using the same protocol aforementioned. Both Cre+ and Cre[WT] mice, homozygous for *H-Ras* and *N-Ras* null alleles and homozygous for floxed *K-Ras* alleles, were treated with tamoxifen prior to injury, but Cre[WT] animals did not undergo genetic recombination and served as control.

**Clearing period following tamoxifen-induced genetic recombination**. A potential confound with tamoxifen-inducible mouse studies is the effect of residual tamoxifen in the CNS at the time of injury. Tamoxifen and its active metabolite 4-hydroxytamoxifen have a half-life of 6–12 h in the mouse[49]. Analysis of CreER[T2] distribution in the adult mouse spinal cord 6 days after the last tamoxifen administration has revealed no CreER[T2] in the nucleus of cells at this time, directly

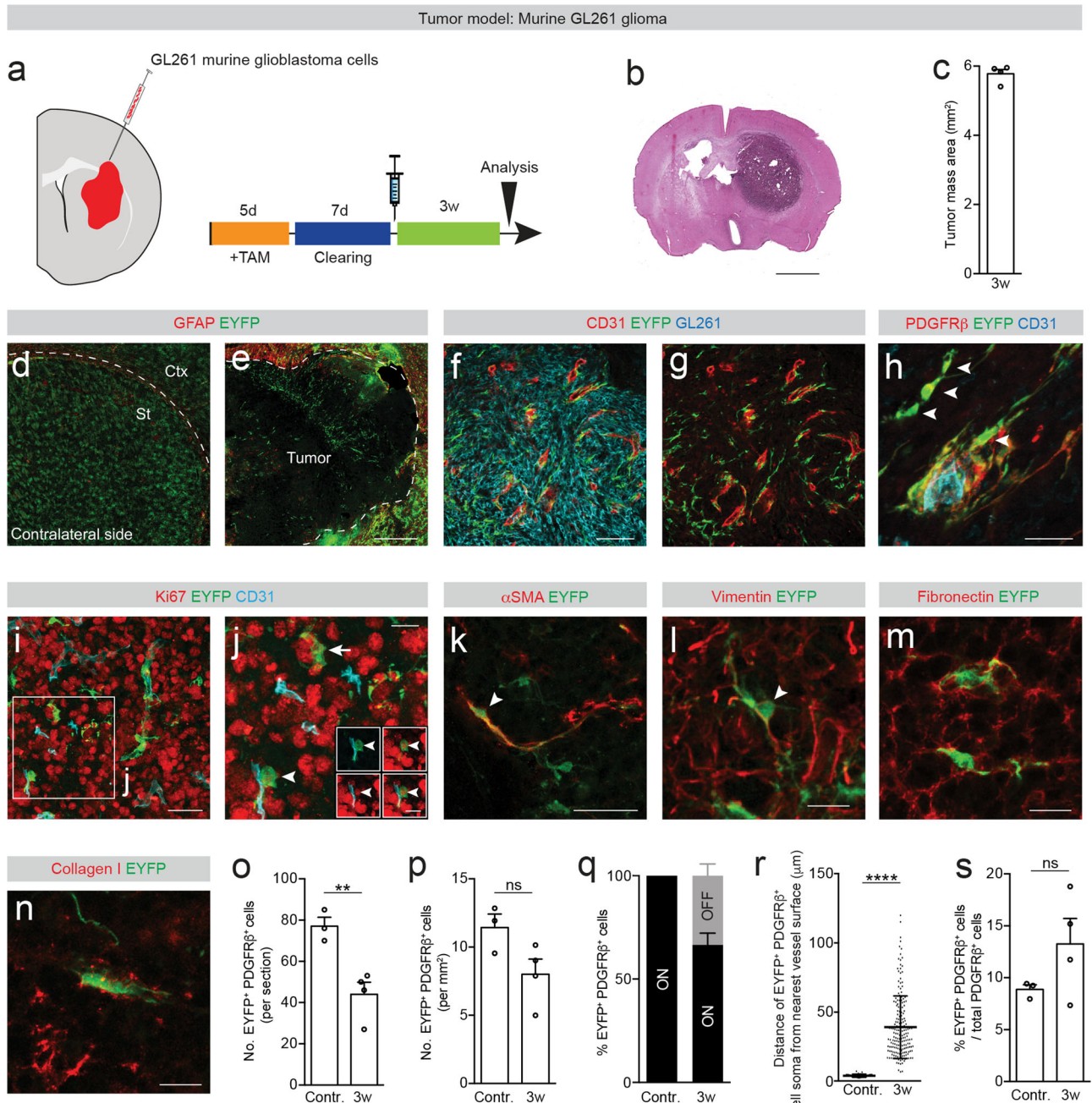

**Fig. 7 Type A pericyte-derived cells contribute to the tumor stroma. a** Glioblastoma model and experimental timeline. Red represents the tumor. **b** Tumor cell mass is confined to the striatum (hematoxylin and eosin stain). **c** Area occupied by the tumor cell mass. Distribution of EYFP$^+$ cells in the contralateral (**d**) and ipsilateral (**e**) side to the tumor. Type A pericytes and progeny (EYFP$^+$GFAP$^-$) are found within the tumor core, whereas partially recombined GFAP$^+$ reactive astrocytes border the tumor. Dashed line separates the cortex (Ctx) and striatum (St) in (**d**) and outlines the tumor core in (**e**). **f**, **g** EYFP$^+$ cells intermingle with GL261 tumor cells (labeled with an antibody against NG2[58]). **h** A fraction of EYFP$^+$PDGFRβ$^+$ cells locates outside the vascular wall (arrowheads; endothelial cells marked by CD31), **i**, **j** EYFP$^+$ cells proliferate in contact (arrowhead) and away (arrow) from the vascular wall. Insets in (**j**) show a EYFP$^+$ cell associated with the vascular wall (CD31$^+$) expressing the mitotic marker Ki67. EYFP$^+$ cells express αSMA (**k**; arrowhead) and vimentin (**l**), and are surrounded by fibronectin (**m**) and collagen I (**n**) ECM. Number (**o**) and density (**p**) of EYFP$^+$PDGFRβ$^+$ cells in the tumor core and contralateral side. **q** Percentage of EYFP$^+$PDGFRβ$^+$ cells associated with (ON vessel) or located outside (OFF vessel) the vascular wall. **r** Distance of EYFP$^+$PDGFRβ$^+$ cells ON vessel (contralateral side to the tumor) or OFF vessel (tumor) from the nearest vessel surface. Each dot represents one cell. **s** Percentage of PDGFRβ$^+$ cells that express EYFP out of total PDGFRβ$^+$ cells. TAM tamoxifen; Scale bars: 1.5 mm (**b**), 200 μm (**d**, **e**), 100 μm (**f**, **g**), 50 μm (**i**, **k**) and 20 μm (**h**, **l**, **m**, **n**, **j** and insets in **j**). Data shown as mean ± s.e.m. $n = 3$ (Contralateral) and $n = 4$ (3w) animals in (**c**, **o**, **p**, **s**); $n = 3$ (Contralateral) and $n = 5$ (3w) animals in (**q**); $n = 36$ (Contralateral) and $n = 189$ (3w) cells examined over 3 animals in (**r**). ns non-significant; **$p < 0.01$, ****$p < 0.0001$ by two-sided, unpaired Student's $t$ test in (**o**, **p**, **s**) and two-sided Mann–Whitney test in (**r**). **f**, **g** denote paired images. All images show coronal sections. Images are representative of two independent experiments. Source data and statistical test results are provided as a Source Data file.

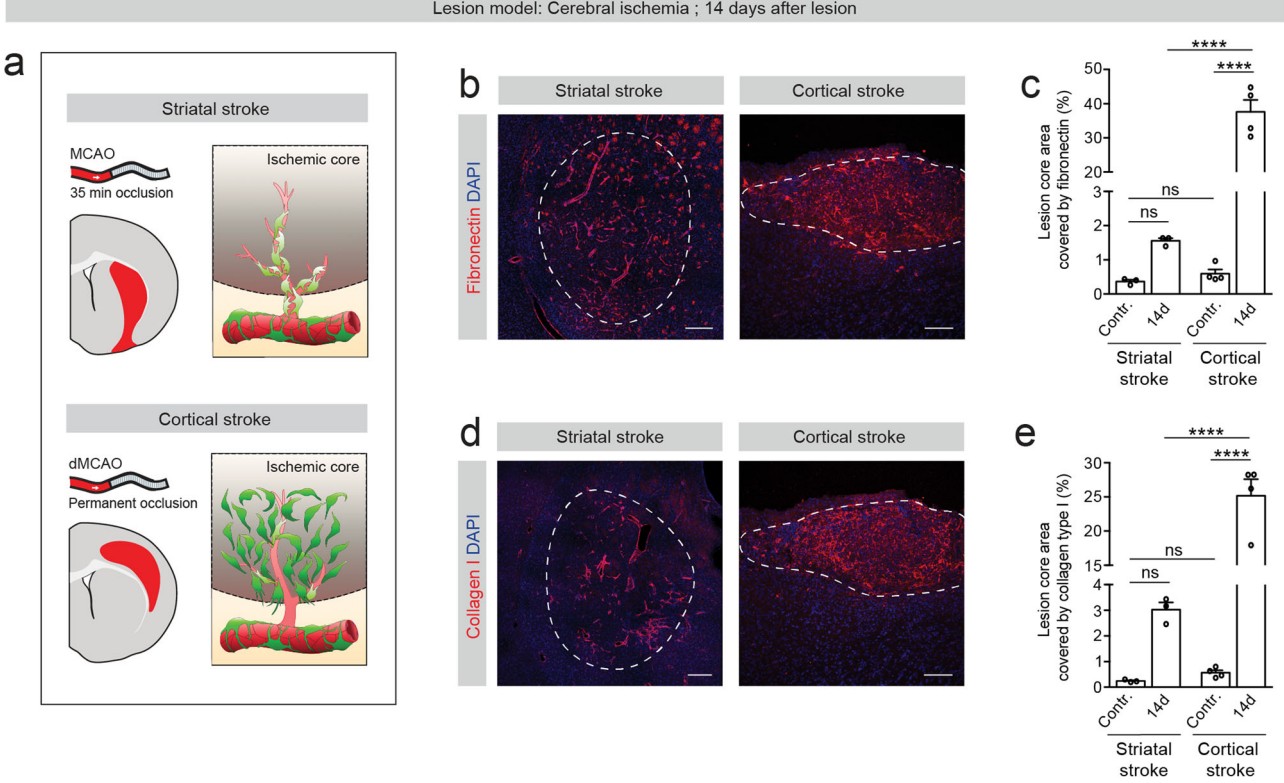

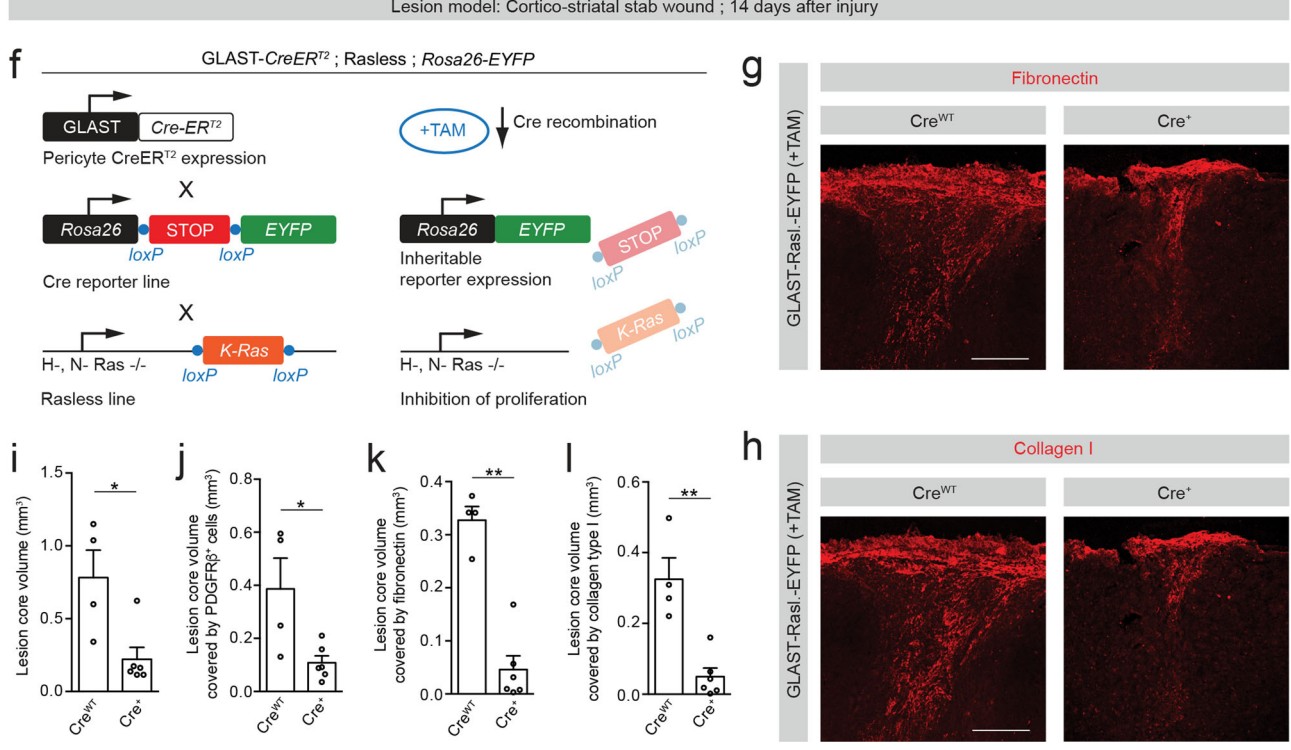

demonstrating that tamoxifen has been cleared at this time point[15]. Moreover, recent studies in the mouse spinal cord and brain have reinforced that 7 days is enough time for tamoxifen to be fully metabolized to ineffective concentrations that are insufficient to result in further recombination[50,51].

Therefore, in this study, injuries (spinal cord injuries, stab wound, and ischemic stroke lesions) and tumor induction (GL261 murine glioma model) were performed after a 7-days clearing period following the last tamoxifen injection. Experimental autoimmune EAE was induced following a 50-day clearing period after the last tamoxifen injection, because the clinical severity of the disease was

diminished if MOG-peptide immunization took place within 10 days after tamoxifen treatment[20].

**Surgical procedures**. For all surgical procedures, animals were anesthetized with 4% isoflurane until unconscious followed by 2% isoflurane during surgery. All animals received analgesia (Temgesic/Buprenorphine, Schering-Plough, 0.1 mg/kg body weight and Rimadyl/Carprofen, Pfizer, 5 mg/kg body weight; subcutaneous injection) for post-operative pain relief and a uniform layer of eye gel (Viscotears/

**Fig. 8 Type A pericytes are required for the generation of fibrotic scar tissue and deposition of ECM after injury. a** Schematic illustrations depicting the contribution of type A pericytes to ischemic lesions confined to the striatum or cortex (lesions represented in red). Upper row: Following striatal stroke, type A pericytes and progeny increase in number but remain associated with the vascular wall (cells colored light green). Lower row: After cortical stroke, type A pericyte-derived cells increase in number and dissociate from the vascular wall (cells colored dark green). Light colored regions represent peri-lesion tissue and darker regions the ischemic stroke core. Representative overview images and quantification of fibronectin (**b**, **c**) and collagen I (**d**, **e**) deposition in the lesion core. Dashed lines encircle the lesion core. **f** Genetic strategy to block the generation of progeny by type A pericytes. Tamoxifen (TAM)-induced genetic recombination in Cre$^+$ GLAST-$CreER^{T2}$;Rasless;$Rosa26$-EYFP (GLAST-Rasless-EYFP) animals prior to injury results in inhibition of injury-induced type A pericyte proliferation through cell-specific deletion of floxed $K$-$Ras$ in mice with $H$-$Ras$ and $N$-$Ras$ null alleles. Both Cre$^+$ and Cre$^{WT}$ animals are treated with tamoxifen prior to injury, but Cre$^{WT}$ animals do not undergo genetic recombination and serve as control. Representative overview images of fibronectin (**g**) and collagen I (**h**) deposition in lesion core. Quantification of the total lesion core volume (**i**) and lesion core volume covered by PDGFRβ$^+$stromal cells (**j**), fibronectin (**k**) or collagen I (**l**) in Cre$^{WT}$ and Cre$^+$ GLAST-Rasless-EYFP animals, 14 days after a cortico-striatal stab wound. TAM tamoxifen; Scale bars: 200 μm (**g**, **h**, striatal stroke in **b**, **d**) and 100 μm (cortical stroke in **b**, **d**). Data shown as mean ± s.e.m. $n = 3$ (Contralateral side striatal stroke), $n = 3$ (Striatal stroke), $n = 4$ (Contralateral side cortical stroke), $n = 4$ (Cortical stroke), $n = 4$ (Stab wound Cre$^{WT}$) and $n = 6$ (Stab wound Cre$^+$) animals. ns non-significant; *$p < 0.05$, **$p < 0.01$, ****$p < 0.0001$ by One-Way ANOVA followed by Tukey's multiple comparisons post-hoc test in (**c**, **e**) and two-sided Mann–Whitney test in (**i-l**). **g**, **h** denote paired images. Cell nuclei are labeled with DAPI. Images show coronal (**a**, **b**) and sagittal (**g**, **h**) sections. Images are representative of two independent experiments. Source data and statistical test results are provided as a Source Data file.

Carbomer, 2 mg/g, Novartis) was applied onto the eye ball to prevent drying. All animals received local anesthesia (Xylocaine/Lidocaine, AstraZeneca, 10 mg/ml and Marcain/Bupivacaine, AstraZeneca, 2 mg/kg body weight) 2–3 min prior to craniotomies and spinal or brain lesions. For recovery from surgery, animals were placed on a heating pad and only returned to their home cage once they were fully awake.

*Penetrating spinal cord injury: dorsal funiculus incision model.* A laminectomy was performed at the mid-thoracic level to expose the dorsal portion of the spinal cord and the dorsal funiculus and adjacent gray matter were cut transversely to a depth of ~0.6 mm with microsurgical scissors. This incision was extended rostrally with microsurgical scissors to span one spinal segment[3,6,15]. Both male and female mice were used.

GLAST-$CreER^{T2}$;R26R-EYFP animals were sacrificed at 5 days ($n = 3$), 14 days ($n = 6$), and 6 weeks ($n = 4$) after spinal cord injury. A separate cohort of animals ($n = 3$) mice received no injury and served as uninjured control.

*Non-penetrating spinal cord injury: complete spinal cord crush model.* A laminectomy was performed to expose the dorsal portion of the spinal cord at the thoracic level 10 (T10). The spinal cord was then fully crushed for 2 s with Dumont No.5 forceps (11295-00, Fine Science Tools) without spacers and that had been filed to a width of 0.1 mm for the last 5 mm of the tips[16,52]. Since this injury paradigm minimizes mechanical disruption of the dura mater, it limits confounding factors such as invasion of dura-derived meningeal fibroblasts into the lesion.

To replace blood loss and prevent/treat bladder infections and post-operative pain, antibiotics (Trimetoprimsulfametoxazol/Tribrissen vet, 400 mg/ml Sulfadizin, 80 mg/ml Trimetoprim, MSD Animal Health, 100 mg/kg body weight/24 h; subcutaneous injection), warm sterile Ringer's solution (500 μl; subcutaneous injection) and analgesics (Rimadyl/Carprofen, Pfizer, 5 mg/kg body weight, once a day and Temgesic/Buprenorphine, Schering-Plough, 0.1 mg/kg body weight, twice per day; subcutaneous injection), were administered in the first 3 days following surgery. Bladders were manually expressed 2–3 times per day until the end of the experiment. As practical refinements to improve animal welfare, mice are housed in group and provided a grid on the cage floor in conjunction with soft nesting material and litter to help mobility. To facilitate animal's hydration water bottles with longer sprouts, hydration gel (HydroGel, Clear H$_2$O) and moistened food pellets are provided in the bottom of the cage. Locomotor recovery was assessed using the Basso Mouse Scale[53] open field test on day 1 after injury and animals showing incomplete paralysis of the hindlimbs were excluded from the experiment. As pre-established, animals that lost more than 15% of their pre-operative bodyweight were euthanized. Only female mice were used for this severe spinal cord injury model.

GLAST-$CreER^{T2}$;R26R-EYFP animals were sacrificed at 5 days ($n = 4$), 14 days ($n = 5$), and 7 weeks ($n = 3$) after spinal cord injury. A separate cohort of animals ($n = 3$) mice received no injury and served as uninjured control.

*Stab lesions.* Animals were placed in a stereotaxic frame and a unilateral craniotomy (diameter of ~3 mm) was performed between cranial sutures bregma and lambda above the right cerebral cortex (adapted from ref .[19]). An incision wound was introduced into the right cortical parenchyma by inserting a sterile surgical scalpel blade (Kiato, #11) positioned 90 degrees perpendicular to the brain surface at 2.0 mm lateral and 1 mm anterior to bregma. To produce small stab wounds restricted to the cerebral cortex, the scalpel blade was lowered −0.5 mm from the brain surface in the dorsoventral axis to avoid direct damage to the deepest cortical layers and underneath white matter, minimizing neuroblast migration from the

subventricular zone of the lateral ventricles and subgranular zone in the dentate gyrus of the hippocampus into the injured cortex. For cortico-striatal stab wounds, the blade was lowered −3 mm from the brain surface in the dorsoventral axis. The lesions were created by moving the blade over 1.3 mm back and forth along the anterior–posterior axis from 1 mm anterior to 0.3 mm posterior to bregma. The craniotomy was then covered with bone wax (Ethicon) in order to preserve the meninges and the brain surface. Both male and female mice were used. GLAST-$CreER^{T2}$;R26R-EYFP animals were sacrificed at 5 days ($n = 3$), 14 days ($n = 4$), and 7 weeks ($n = 3$) after cortico-striatal stab lesions. The contralateral side to the lesion served as control ($n = 10$). A separate cohort of GLAST-$CreER^{T2}$;R26R-EYFP animals was sacrificed at 14 days after small stab lesions restricted to the cerebral cortex ($n = 3$). For cortico-striatal stab lesion experiments involving GLAST-Rasless-EYFP mice, animals were sacrificed at 14 days ($n = 6$ Cre$^+$; $n = 4$ Cre$^{WT}$) after injury. To label dividing cells following cortico-striatal stab lesions, a separate group of GLAST-$CreER^{T2}$;R26R-EYFP (also referred to as GLAST-EYFP) and GLAST-Rasless-EYFP mice was administered EdU (5-ethynyl-2'-deoxyuridine; Molecular Probes #A10044) twice by intraperitoneal injections (12 mg/ml, 100 μl per injection) at 6 h interval, followed by EdU administration in the drinking water for 5 days (0.2 mg/ml and 1% sucrose, exchanged every 2–3 days and kept in dark). Animals were sacrificed at 5 days ($n = 3$ GLAST-EYFP; $n = 4$ GLAST-Rasless-EYFP Cre$^+$) after injury. The contralateral side to the lesion served as control.

To generate large stab wounds restricted to the cerebral cortex, with similar depth and size as cortico-striatal stab lesions, after craniotomy, an incision wound was made to the right cortical parenchyma by inserting a sterile surgical scalpel blade (Kiato, #11) positioned parallel to the brain surface at 3 mm lateral and 1 mm anterior to bregma. The blade was slowly advanced from the brain surface 3 mm deep inside the cerebral cortex in the right–left lateral axis. The lesions were created by moving the blade over 1.3 mm back and forth along the anterior–posterior axis from 1 mm anterior to 0.3 mm posterior to bregma. The craniotomy was subsequently covered with bone wax (Ethicon) in order to preserve the meninges and the brain surface. Both male and female mice were used. GLAST-$CreER^{T2}$;R26R-tdTom animals were sacrificed at 5 days ($n = 4$) and 14 days ($n = 4$) after large stab lesions restricted to the cerebral cortex. The contralateral side to the lesion served as control ($n = 8$).

*Focal ischemic stroke.* Focal ischemic stroke was induced by two methods. First method is based on transient occlusion of the whole middle cerebral artery (MCA) by intraluminal filament method[21] leading to the selective lesion of the striatum or the striatum and cerebral cortex, depending on the duration of the occlusion. The second method induces the selective lesion of the cortex[22] and this is achieved through permanent occlusion of the distal cortical branch of the MCA.

Middle cerebral artery occlusion (MCAO)
Following a sagittal midline incision on the ventral surface of the neck, the common carotid artery (CCA) and its proximal branches were isolated. After the CCA and the external carotid artery (ECA) were ligated using a 4–0 silk suture (Ethicon), the internal carotid artery (ICA) was temporarily clipped with a metal microvessel clip. A small incision was made on the anterior surface of the ECA to insert a silicon-coated nylon microfilament (Doccol Corporation). The microfilament was advanced through the ICA until the branch point of the MCA and secured using the suture around the ECA. The filament was carefully removed 35 min after the occlusion to restore the blood flow in the MCA and the ECA was ligated permanently. Transiently occluding the MCA for 35 min led to ischemic lesions confined to the striatum (striatal stroke). Only male mice were used. GLAST-$CreER^{T2}$;R26R-EYFP animals were sacrificed at 5 days ($n = 3$), 14 days ($n = 3$), and 7 weeks ($n = 3$) after striatal stroke induction. The contralateral side to the lesion served as control ($n = 9$).

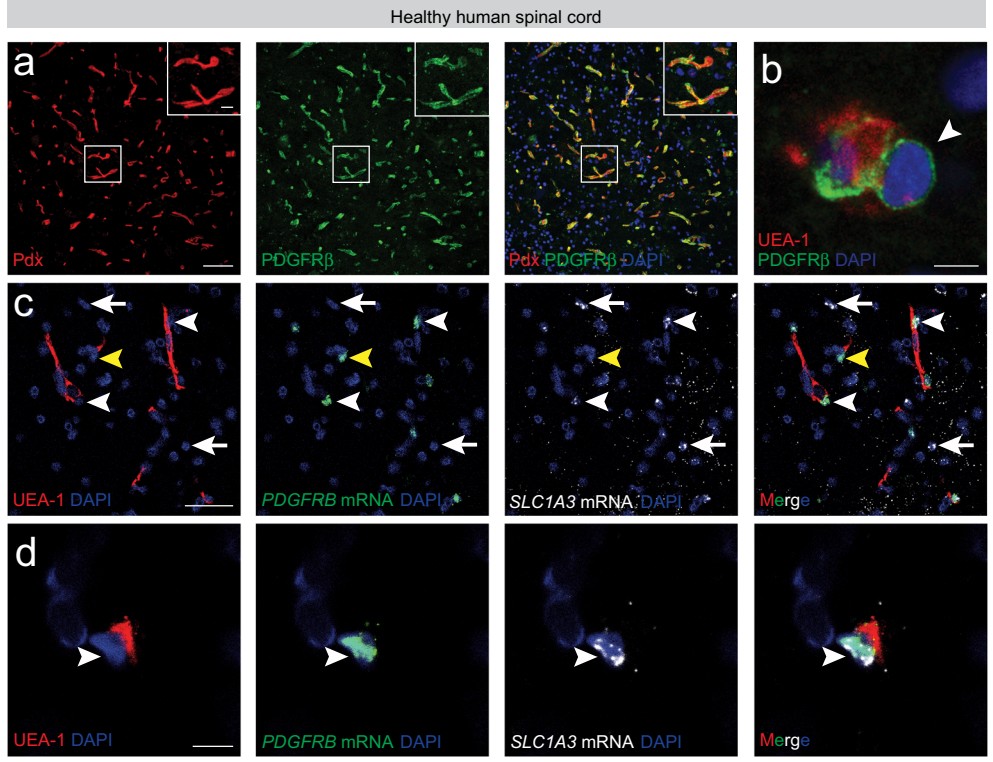

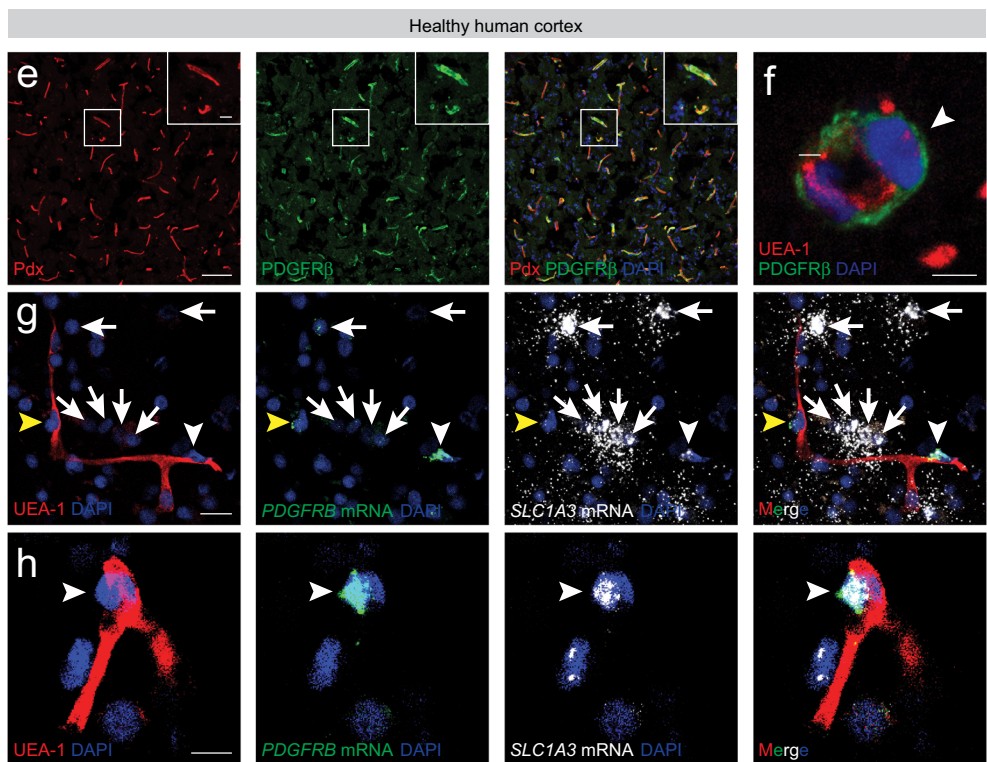

In a separate cohort of animals the MCA was occluded for 45 min, leading to ischemic lesions that, in addition to the striatum, extended into the cerebral cortex (cortico-striatal stroke). Only male mice were used. GLAST-*CreER^{T2};R26R-tdTom* animals were sacrificed at 5 days (*n* = 3) and 14 days (*n* = 3) after cortico-striatal stroke induction.

**Distal middle cerebral artery occlusion (dMCAO)**
Skin was cut between the right eye and the right ear and a small hole was then drilled in the cranium at the level of the MCA, the dura mater was removed, the distal portion of

the right MCA was exposed and occluded by cauterization. The artery was then cut off to be sure that there was no remaining blood flow to the corresponding cortical region. After the skin had been sutured, mice were injected with 0.5 ml Ringer's solution, returned to their home cages, and placed on a heating pad.

Permanent occlusion of the distal portion of the MCA led to ischemic lesions confined to the cerebral cortex (cortical stroke). Only male mice were used. GLAST-*CreER^{T2};R26R-tdTom* animals were sacrificed at 5 days (*n* = 4) and 14 days (*n* = 4) after cortical stroke induction. The contralateral side to the lesion served as control (*n* = 8).

**Fig. 9 GLAST-expressing perivascular cells reside in the human CNS vasculature.** Representative images of post-mortem human tissue from the healthy spinal cord (**a**) and occipital cortex (**e**) labeled with antibodies recognizing PDGFRβ and podocalyxin (Pdx). Insets show magnified boxed regions and reveal PDGFRβ-expressing perivascular cells enveloping the endothelial tube. Example of a pericyte (labeled with an antibody against PDGFRβ) with characteristic protruding ovoid cell body and processes encircling the endothelial wall of a small caliber blood vessel in the healthy human spinal cord (**b**). Example of a pericyte (PDGFRβ⁺) extending its processes around the endothelial wall of a small caliber blood vessel in the healthy human occipital cortex (**f**). Human endothelial cells were labeled with *Ulex Europaeus*-I (UEA-1) lectin. Arrowheads points at the pericyte soma. Detection of *SLC1A3* (GLAST) and *PDGFRB* mRNA in the healthy human spinal cord (**c, d**) and occipital cortex (**g, h**) by RNAscope in situ hybridization combined with immunofluorescence for the UEA-1. White arrowheads point at blood vessel-associated cells positive for *PDGFRB* and *SLC1A3* mRNA signals, representing GLAST⁺PDGFRβ⁺ perivascular cells; yellow arrowheads identify blood vessel-associated cells positive for *PDGFRB* mRNA signals with low/undetectable levels of *SLC1A3* mRNA, representing GLAST⁻ PDGFRβ⁺ perivascular cells; white arrows point at parenchymal (non-blood vessel-associated) cells positive for *SLC1A3* mRNA signals with low/undetectable levels of *PDGFRB* mRNA, likely representing GLAST⁺PDGFRβ⁺ astrocytes. Cell nuclei are labeled with DAPI. Scale bars: 100 μm (**a**, **e**), 50 μm (**c**), 20 μm (**g**, insets in **a**, **e**), 10 μm (**d**, **h**) and 5 μm (**b**, **f**). All images show coronal sections. Images are representative of four independent individuals.

*Experimental autoimmune encephalomyelitis (EAE).* To induce EAE, animals were anesthetized with isoflurane and immunized subcutaneously at the dorsal tail base with 300 μg of rodent MOG peptide (aminoacids 35–55) emulsified in CFA containing 5 mg/ml inactivated *Mycobacterium tuberculosis* (Difco). Pertussis toxin (75 ng/mouse in PBS) was injected intravenously on the day of immunization and 48 h later.

Animals allocated to control groups received pertussis toxin only or CFA containing 5 mg/ml inactivated *Mycobacterium tuberculosis* plus pertussis toxin, but no MOG peptide.

The animals were weighted and neurological deficits assessed daily until the end of the experiment according to previous scoring methods[54,55]: 0—no apparent symptoms, 1—flaccid tail or wobbling gait, 2—flaccid tail and wobbling gait, 2.5—single limb paresis and ataxia, 3—double limb paresis, 3.5—single limb paralysis and paresis of second limb, 4—complete paralysis of hind- and fore- limbs, 4.5—moribund, 5—dead.

Both male and female mice were used. GLAST-CreER^T2^;R26R-EYFP animals were sacrificed 30 days post-immunization with MOG peptide in CFA (EAE group, n = 4), or pertussis toxin alone and/or CFA (control group, n = 5). Only animals that have reached a clinical score of at least 3 by 30 days after EAE induction and presented chronic neurological deficits (and not relapse-remitting symptoms) were analyzed in the EAE group.

*Syngenic mouse model of glioblastoma.* GL261 mouse glioma cells (DSMZ, ACC 802) were maintained in DMEM 1X, high glucose, GlutaMAX (+pyruvate) medium (Life Technologies, 31966-021) supplemented with 10% heat inactivated Fetal Bovine Serum and 1x Pen-Strep at 37 °C in a humidified atmosphere containing 5% CO₂. Cells were cultured in the absence of antibiotics before transplantation.

Animals were head-fixed in a stereotaxic frame and a burr hole was drilled over the sensorimotor cortex to expose the brain. Fifty thousand GL261 cells were resuspended in 1 μl of sterile phosphate buffered saline (PBS) and injected in the striatum at 2.3 mm lateral to the midline, 0.1 mm posterior of the bregma and 2.3 mm ventral from dura, at a rate of 0.1 μl/min using a 10 μl syringe with a 26-gauge beveled needle tip (Hamilton, 901RN) coupled to a microinjector (UltraMicroPump III and Micro4 microsyringe pump controller, World Precision Instruments). After injection, the needle was kept in place for an additional 5 min to allow the cells to diffuse and prevent backflow of the cells to the surface, and then slowly withdrawn. Both male and female mice were used. GLAST-CreER^T2^;R26R-EYFP animals (n = 3–5) were sacrificed 3 weeks after inoculation of the tumor cells. The contralateral side to the tumor served as control.

**Human tissue collection and ethical compliance.** The study was conducted according to the Helsinki declaration and the protocols involving human tissue samples were approved by the Regional Ethics Committee of Sweden (2010/313-31/3). All human biospecimens were collected from consented participants or their next-of-kin for use in future research.

Supplementary Tables 1–4 show the clinical and neuropathological data of all subjects included in the study.

*Stroke and glioblastoma samples.* The institutional review boards and the local ethics committee of the Medical Faculty of the University of Erlangen-Nuremberg, Erlangen, Germany, approved the study and informed consent was obtained from the relatives of all analyzed patients. For stroke samples, tissue collection was performed in such cases that had a history of a supratentorial territorial or lacunar stroke involving cerebral cortical and subcortical areas and died from a non-neurological cause. The lesion was cut out according to the macroscopically visible infarct borders including a 0.5 cm rim of surrounding healthy tissue. Healthy occipital cortex and healthy tissue from the contralateral hemisphere in corresponding topography served as control tissue. Brain tissue was frozen and stored at −80 °C until further analysis. For glioblastoma samples, initial histological analysis for verification of tumor type and WHO classification[56] was performed by local experienced neuropathologists on formalin embedded tissue pieces (frozen tumor tissue samples, 20 μm microtome sectioning and fixed in 4%

formaldehyde buffered in PBS for 30 min). Standard hematoxylin and eosin staining, and immunohistochemical analyses for GFAP, MAP2 and pan cytokeratin-1 (KL-1) confirmed tumor types (glioblastoma, WHO grade IV). Brain tissue was frozen and stored at −80 °C until further analysis.

*Multiple sclerosis samples.* MS tissue samples and associated clinical and neuro-pathological data were supplied by the UK Multiple Sclerosis Tissue Bank, supported by the Multiple Sclerosis Society of Great Britain and Northern Ireland, in partnership with Imperial College London. The ethical permission was granted by the Regional Ethics Committee for UK (Research Ethics Committee for Wales). All samples have been donated with informed consent for use in future research. Spinal cord tissue was frozen and stored at −80 °C until analysis.

*Spinal cord injury samples.* Human spinal cord injury samples and related clinical and neuropathologial information were obtained from the International Spinal Cord Injury Biobank (ISCIB), which is housed in Vancouver, BC, Canada. Permission for post-mortem spinal cord acquisition and for sharing of biospecimens was granted by the Clinical Research Ethics Board (CREB) of the University of British Columbia, Vancouver, Canada. All biospecimens were collected from consented participants or their next-of-kin. Provided formalin-fixed and paraffin-embedded tissue sections were stored at 4 °C until analysis.

**Histology and immunohistochemistry.** Animals were euthanized by intraperitoneal injection of an overdose of sodium pentobarbital and transcardially perfused with cold PBS followed by 4% formaldehyde in PBS. Brains and spinal cords were dissected out and post-fixed in 4% formaldehyde in PBS overnight at 4 °C. Spinal cords were then cryoprotected in 30% sucrose and coronal or sagittal cryosections were collected on alternating slides and stored at −20 °C. Coronal and sagittal brain sections were obtained using a vibratome (Leica VT1000 S vibrating blade microtome) and stored free-floating in PBS supplemented with 0.04% sodium azide at 4 °C.

For human glioblastoma, stroke and MS tissue samples, fresh frozen sections were prepared with a cryostat (14–20 μm) and post-fixed in PBS-buffered 2% formaldehyde (wt/vol) for 10 min according to standard procedures. Human healthy spinal cord and occipital cortex tissues were processed in the same manner.

Formalin-fixed and paraffin-embedded coronal spinal cord tissue sections (5 μm thick) were obtained from spinal cord injury patients. For immunohistochemistry, spinal cord sections were deparaffinized in xylene and rehydrated in a descending ethanol series. Antigen retrieval was performed in citraconic acid solution (pH = 7.4; 0.05% citraconic acid) for 20 min in a domestic steamer[57]. The sections were allowed to cool down for 20 min before immunostaining was started.

Sections were incubated with blocking solution (10% normal donkey serum in PBS, with 0.3% Triton X-100) for 1 h at room temperature, and then incubated at room temperature overnight in a humidified chamber with primary antibodies diluted in 10% normal donkey serum. The following primary antibodies were used to immunostain mouse tissue: CD31 (1:100, rat, BD Biosciences), GFAP (1:200, guinea pig, Synaptic Systems; 1:1000, mouse directly conjugated to Cy3, Sigma-Aldrich; 1:1000, chicken, Millipore), GFP (1:2000, goat directly conjugated to FITC, Abcam; 1:10000, chicken, Aves Labs; 1:500, sheep, Bio-Rad), PDGFRβ (1:200, rabbit, Abcam; 1:100, rat, eBioscience; 1:200, goat, R&D Systems), podocalyxin (1:200, goat, R&D Systems), desmin (1:200, rabbit, Abcam), CD13 (1:500, rat, Abcam), αSMA (1:500, mouse directly conjugated to Cy3, Sigma-Aldrich; 1:100, rabbit, Abcam), vimentin (1:1000, chicken, Chemicon; 1:500, rabbit, Abcam), NG2 (1:200, rabbit, Millipore), Ki67 (1:1000, rat, eBioscience), CD45 (1:500, rabbit, Abcam), CD3 (1:200, rat, Biolegend), von Willebrand factor (1:200, rabbit, Dako), TFRC (1:200 mouse, Thermo Fisher Scientific; 1:200, rat, Novus Biologicals), fibronectin (1:500, sheep, Bio-Rad), collagen type I (1:100, goat, Southern Biotech; 1:200, rabbit, Abcam), Aldh1L1 (1:500, rabbit, Abcam), S100β (1:500, guinea pig, Synaptic Systems) and glutamine synthetase (1:2000, rabbit, Invitrogen). An antibody against NG2 (1:200, rabbit, Millipore) was used to detect GL261 tumor cells[58]. Myelin was labeled by incubating tissue sections for 30 min with FluoroMyelin Red

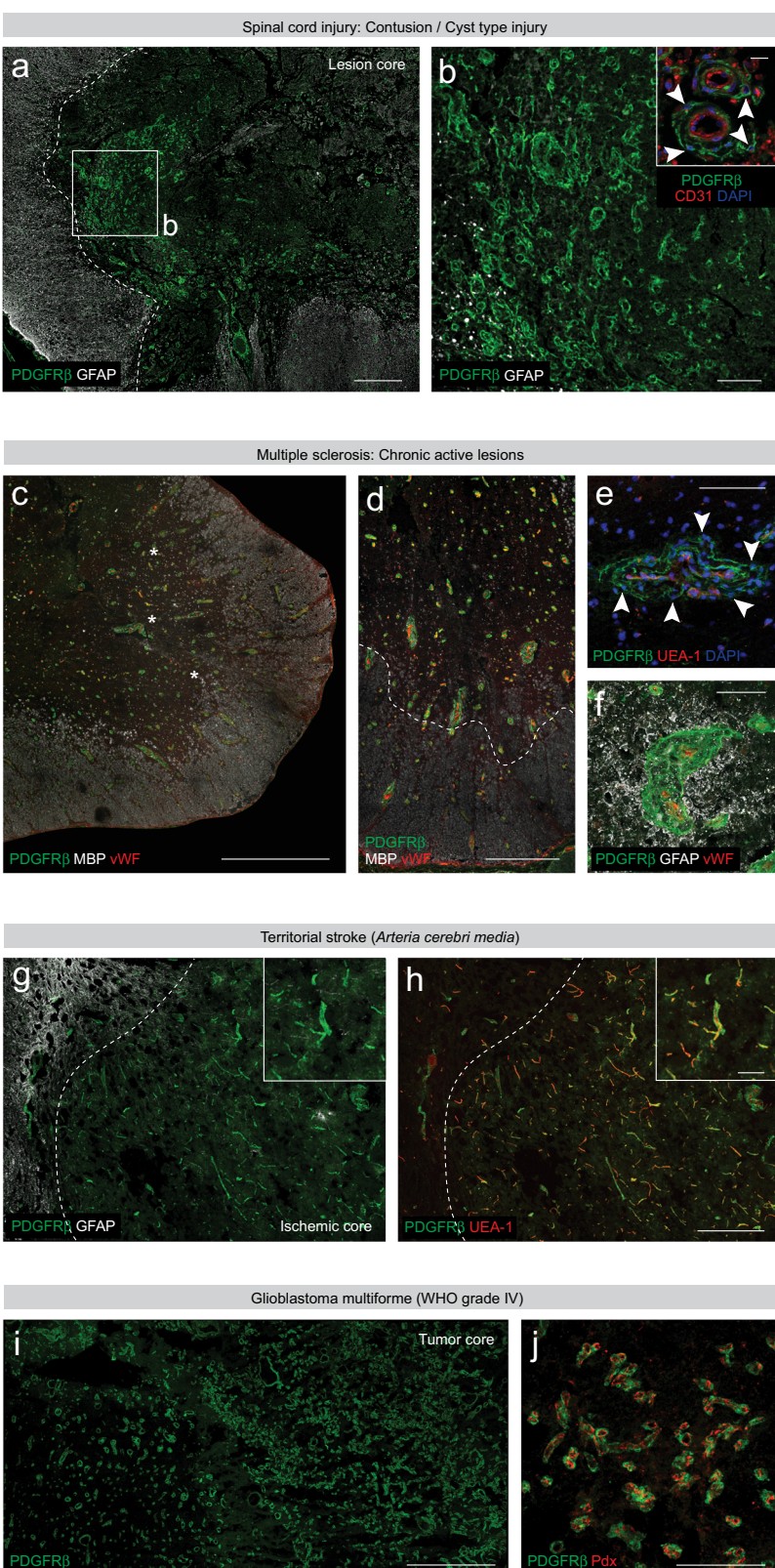

fluorescent myelin stain (1:500, Invitrogen, F34652). For collagen type I and fibronectin immunolabeling, heat-induced antigen retrieval was performed in citrate buffer (10 mM sodium citrate, 0.05% Tween 20, pH 6.0) for 5 min in a domestic steamer. The sections were allowed to cool down for 20 min before commencing the immunostaining. The following primary antibodies were used to immunostain human tissue: PDGFRβ (1:100, goat, R&D Systems), GFAP (1:200, rabbit, Dako; 1:1000, chicken, Millipore), MBP (1:500, rat, AbD Serotec; 1:1000, mouse, Covance), CD31 (1:100, mouse, Dako), podocalyxin (1:200, goat, R&D Systems) and von Willebrand factor (1:200, rabbit,

Dako). *Ulex Europaeus* agglutinin I (UEA-1; 1:200, directly conjugated to FITC or Rhodamine, Vector Labs) was used to label human endothelial cells. Following primary antibody incubation and washing, antibody staining was revealed using species-specific fluorophore-conjugated (DyLight 405, Alexa Fluor 488, Cy3, Alexa Fluor 647 from Jackson Immunoresearch) or biotin-conjugated secondary antibodies (1:500, Jackson Immunoresearch). Biotinylated secondary antibodies were revealed with fluorophore-conjugated streptavidin (1:500, Alexa Fluor 488 or Cy3 from Jackson Immunoresearch). EdU was detected with the Click-iT EdU Alexa Fluor 647 imaging kit (Molecular

**Fig. 10 Fibrotic scarring in human pathology.** PDGFRβ$^+$ stromal cells accumulate in the core of the lesion, 16 days after human traumatic spinal cord injury at C4 (**a**, **b**). As observed following experimental spinal cord injury in rodents, a fraction of PDGFRβ$^+$ stromal cells is located in distance to the vascular wall (CD31$^+$) following traumatic spinal cord injury in humans (inset in **b**). Dashed line in (**a**) demarcates the border between GFAP$^+$ reactive glia and non-neural scar tissue; (**b**) shows a higher magnification of the boxed region in (**a**). **c**–**f** Perivascular aggregates of PDGFRβ$^+$ stromal cells are detected in chronic active multiple sclerosis lesions (**c**, **d**), 21 years following the onset of the disease (secondary progressive multiple sclerosis). Gliosis (GFAP$^+$) is observed in regions of PDGFRβ$^+$ perivascular cell reactivity (**f**). Von Willebrand factor (vWF) marks endothelial cells in (**c**, **d**, **f**). Asterisks in (**c**) denote demyelinated regions (weak MBP signal) of the spinal cord. Arrowheads in (**e**) point at PDGFRβ$^+$ cells that do not contact the blood vessel wall (UEA-1 marks endothelial cells). Dashed line in (**d**) separates myelinated white matter tissue (MBP$^+$) from partially demyelinated lesion areas. **g**, **h** PDGFRβ$^+$ stromal cells remain attached to the blood vessel wall (UEA-1$^+$), 7 weeks following a territorial stroke located in the basal ganglia (**g**, **h** and insets). Dashed line marks the border between GFAP$^+$ reactive glia and the ischemic lesion core. PDGFRβ$^+$ stromal cells populate the stroma of a glioblastoma multiforme (grade IV tumor) that involves the corpus callosum and spreads bihemispherically (**i**). No substantial fraction of PDGFRβ$^+$ stromal cells (immunopositive for podocalyxin) is observed away from the vascular wall (**j**). MBP myelin basic protein; Cell nuclei are labeled with DAPI. Scale bars: 1 mm (**c**, **i**), 500 μm (**a**, **d**, **g**, **h**), 200 μm (**j**), 100 μm (**b**, **e**, **f**, insets in **g**, **h**) and 20 μm (inset in **b**). **g**, **h** show paired images. All images show coronal sections. Images are representative of six (spinal cord injury), six (multiple sclerosis), three (subcortical stroke) and three (glioblastoma) independent individuals.

Probes, #C10340) according to the manufacturer's instructions. Control tissue sections were stained with secondary antibody alone. Cell nuclei were stained with 4',6'-diamidino-2-phenylindole dihydrochloride (DAPI, 1 μg/ml, Sigma-Aldrich). Sections were coverslipped using VECTASHIELD antifade mounting medium (Vector Labs, H-1000).

For hematoxylin & eosin (H&E) staining, tissue sections were washed in ddH$_2$O for 5 min, followed by 1 min immersion in hematoxylin (Vector Labs, H-3401) and washed again under tap water until clear. Following the immersion in ddH$_2$O, sections were stained with eosin (Vector Labs, H-3502) for 3 min. Sections were finally washed in ddH$_2$O for 1 min, dehydrated and coverslipped with VectaMount permanent mounting media (Vector Labs, H-5000).

**RNAscope in situ hybridization.** For the detection of glutamate aspartate transporter 1 (SLC1A3) and platelet-derived growth factor receptor beta (PDGFRB) RNA molecules in the healthy human spinal cord and brain RNAscope in situ hybridization was performed on fresh frozen tissue sections (14–20 μm thick) stored at −80 °C. For the detection of T-box transcription factor 18 (Tbx18) and Pdgfrb RNA molecules in the mouse spinal cord and brain RNAscope in situ hybridization was performed on formaldehyde fixed tissue sections from the (1) uninjured spinal cord and (2) injured spinal cord, 14 days after complete spinal cord crush (20 μm thick sections); (3) injured brain, 5 days after a large cortical stab wound and (4) control brain, contralateral to the stab lesion (30 μm thick sections). Spinal cord cryosections were stored at −80 °C and free-floating brain sections obtained using a vibratome were stored in PBS at 4 °C.

In situ hybridization was performed following a modified version of the RNAscope Multiplex Fluorescent Reagent Kit v2 Assay (ACD Bio-Techne, 323100). Briefly, tissue sections were mounted onto SuperFrost Plus adhesion slides and baked for 30 min at 60 °C, prior to post-fixation in 4% formaldehyde in PBS for 15 min at 4 °C. Slides were subsequently dehydrated in 50, 70, and 100% ethanol for 5 min each at room temperature, followed by 10 min incubation with H$_2$O$_2$. After drying, slides were first boiled for 10 s in distilled water followed by 5 min in antigen retrieval solution, washed in distilled water and 100% ethanol at room temperature and allowed to dry again. For further antigen accessibility sections were incubated with protease III (ACD, 322337) at 40 °C for 30 min. For the RNAscope assay, the samples were washed in distilled water and incubated at 40 °C for 2 h with the following probes: Hs-SLC1A3-C2 (ACD Bio-Techne, 461081-C2) and Hs-PDGFRB (ACD Bio-Techne, 548991) for human spinal cord and brain samples, as well as Mm-Tbx18-C2 (ACD Bio-Techne, 515221-C2) and Mm-Pdgfrb-C3 (ACD Bio-Techne, 411381-C3) for mouse samples. Subsequent amplification and detection were performed following the assay protocol. Probes were detected with Opal 520 (Perkin Elmer, FP1487A) and Opal 650 (Perkin Elmer, FP1496A) fluorophores. Human blood vessels were labeled with fluorescein (FITC) conjugated-Ulex Europaeus Agglutinin 1 (UEA-1, Vector laboratories, FL-1061), which recognizes human endothelial cells. Blood vessels in the uninjured mouse spinal cord and control brain were labeled with biotinylated Lycopersicon Esculentum (Tomato) lectin (Vector Labs, B-1175-1). In mouse samples, tdTomato labeling was recovered by immunohistochemistry with an antibody against Red Fluorescent Protein (1:500, rabbit, Rockland; 1:500, chicken, Novus Biologicals). Cell nuclei were stained with DAPI (1 μg/mL, Sigma-Aldrich, D9542) and the sections were coverslipped with ProLong Gold antifade mountant (Invitrogen, P10144). The protocol was run in one day to preserve sample quality.

**Electron microscopy**

*Transcardial perfusion.* Uninjured GLAST-CreER$^{T2}$;R26R-tdTom animals were euthanized by intraperitoneal injection of an overdose of sodium pentobarbital and transcardially perfused with a mixture of 4 % formaldehyde and 0.1% glutaraldehyde in 0.1 M phosphate buffer for 15 min with an initial 20 s perfusion with ice cold 2% dextran in 0.1 M phosphate buffer. Spinal cords were removed, left in the fixation solution overnight and stored in a 1:10 dilution of the same solution in 0.1 M phosphate buffer.

*Post-embedding and immunogold electron microscopy.* Small blocks were dissected out from perfusion fixed spinal cord, and subjected to freeze substitution procedure[59]. Briefly, tissue blocks were cryoprotected in 4% glucose overnight, followed by suspending the tissues in graded glycerol solution (10, 20, and 30% glycerol in 0.1 M phosphate buffer for 30 min in each gradient). Following cryoprotection, the tissue blocks were rapidly frozen in propane cooled to −170 °C using liquid nitrogen before subjecting to freeze substitution. Samples were later embedded in methacrylate resin (Lowicryl HM20) and polymerized by UV irradiation below 0 °C. Sections were cut at 90–100 nm and placed on 300 mesh Ni-grids until further use.

For immunogold cytochemistry[60] the ultrathin sections were incubated with a primary antibody against Red Fluorescent Protein, which recognizes tdTomato (1:100, rabbit, Rockland, 600-401–379 s) overnight at room temperature in a humidified chamber. The following day, the sections were washed and incubated with secondary antibody (1:20, goat anti-rabbit IgG conjugated to 15 nm gold particles, Abcam, ab27236) for 2 h. The sections were counterstained using 2% uranyl acetate and 0.3% lead citrate for 90 s each. Images were acquired with a Tecnai 12 electron microscope (FEI) at 80 kV equipped with ITEM FEI version 5.1 software (Olympus Soft Imaging Solutions, Münster, Germany).

**Imaging and quantitative analysis.** Images were acquired with a Leica DM5500 B bright-field microscope coupled to the LAS X 3.7.2.22383 software, a Zeiss Axioplan 2 upright epifluorescent microscope equipped with the ZEN 2 software (version 2.0.14283.302) or a Leica TCS SP8X confocal microscope equipped with the LAS X 3.5.7.23225 software.

Image processing and assembly were performed with ImageJ/Fiji (version 2.0.0-rc-43/1.51j for Mac), Adobe Photoshop CC 2018 19.1.1 release and Illustrator CC 2018 22.0.1 release for Mac.

Coronal (dorsal funiculus incision spinal cord injury model) or sagittal (complete crush spinal cord injury model) spinal cord sections, spanning the injury site and 0.4 mm rostral and caudal to the injury site, were collected on 20 alternating slides at 20 μm thickness and used for quantifications. Matched segments of uninjured spinal cords were sectioned and collected in a similar fashion. For EAE spinal cord samples, 20 μm thick coronal sections covering cervical, thoracic and lumbar spinal segments were collected on 50 alternating slides and used for quantifications. Control spinal cord samples (pertussis toxin alone and/or CFA) were sectioned and collected in a similar fashion. Brains with stab lesions were sectioned at 30 μm thickness and collected in 20 alternating series. Coronal (large cortical stab wounds) or sagittal (cortico-striatal and small cortical stab wounds) sections covering the lesion were used for analyses. For focal ischemic stroke and mouse glioma samples, coronal brain sections covering the ischemic lesion and tumor, respectively, were collected in 20 alternating series at 30 μm thickness and used for analyses.

All quantifications were done in at least three alternate sections per animal covering the lesion epicenter and spaced 400 μm apart. The contralateral side to the lesion served as control and was used for mouse glioma, stroke and stab wound control analyses. Uninjured spinal cord samples were used as controls for spinal cord injury quantifications. Three alternate sections per animal, spaced 400 μm apart were sampled for control analyses. For EAE and respective control (pertussis toxin alone and/or CFA) samples, three cervical, three thoracic and three lumbar sections per animal, spaced 1000 μm apart were analysed.

To determine the lesion core area, sections were immunostained for PDGFRβ, GFAP, and DAPI. The lesions were imaged using a Zeiss Axioplan 2 epifluorescent microscope and the lesion core area, defined as the GFAP negative area within the GFAP-defined borders filled with PDGFRβ-expressing stromal cells, was manually outlined. For EAE samples, the area covered by the multiple scar-like clusters of PDGFRβ-expressing stromal cells and reactive astrocytes was manually delineated. Area measurements were carried out using the ImageJ/Fiji software.

The total number of type A-pericyte-derived stromal cells, defined as EYFP$^+$ or tdTom$^+$ cells expressing PDGFRβ within the lesion core, was assessed in spinal cord or brain sections immunolabeled for GFP (that cross-reacts and recognizes

EYFP), PDGFRβ, GFAP, and DAPI. TdTom signal is sufficiently bright and does not need antibody-based enhancement. Similarly, matched areas on the striata contralateral to the lesion side or uninjured/control spinal cord sections were used to quantify the total number of type A pericytes (EYFP+ or tdTom+ cells expressing PDGFRβ enwrapping the blood vessel wall) under control conditions. The data were averaged and presented as total number of recombined (EYFP+ or tdTom+) PDGFRβ-expressing cells per section.

The density of type A pericyte-derived stromal cells at the lesion epicenter, a measure that reflects cell clustering, was calculated by dividing the total number of recombined PDGFRβ-expressing cells per section by the corresponding lesion core area. For uninjured/control spinal cord quantifications, the total number of type A pericytes per spinal section was divided by the area of the corresponding spinal cord section. For control brain quantifications, the total number of type A pericytes counted in the striatum contralateral to the lesion side was divided by the corresponding striatal area. The data were pooled and presented as the mean number of recombined PDGFRβ-expressing cells per area.

To determine the proportion of recombined cells either associated with or dissociated from the blood vessel wall within the lesion core, tissue sections were immunostained for the endothelial markers CD31 or podocalyxin, GFP, PDGFRβ and DAPI. Four randomly selected fields ($0.83 \times 0.66$ mm) per section were photographed and used for analysis. A cell was considered OFF the vessel only when the cell soma or processes had no direct contacts with the nearest blood vessel surface and the center of the nucleus of the cell is at a distance >8 μm from the edge of the closest blood vessel surface (measured in 2D). Cells with any contact points (soma or processes) to the nearest blood vessel surface and with the center of the nucleus of the cell at distance ≤8 μm from the edge of the closest blood vessel surface were considered ON the vessel. The number of recombined PDGFRβ-positive cells in contact with the blood vessel wall (ON the vessel) or away from the blood vessel wall (OFF the vessel) was manually counted and the data expressed as a percentage of total recombined PDGFRβ-expressing cells. The percentage of type A pericytes ON or OFF the vessels under uninjured/control conditions was assessed in a similar fashion.

To measure the distance of recombined (EYFP+ or tdTom+) PDGFRβ-positive cells (type A pericytes and progeny) from the nearest vessel surface, tissue sections were immunostained for the endothelial cell marker podocalyxin, GFP, PDGFRβ, and DAPI. Three to four randomly selected fields ($0.83 \times 0.66$ mm) of the lesion epicenters or uninjured/control sections were captured and the images were analyzed using the ImageJ/Fiji software. Utilizing the angle tool, a straight line was drawn from the center of the DAPI+ nuclei of recombined PDGFRβ+cells to the nearest blood vessel surface. Analyses were focused on recombined cells in the center of the image; recombined cells situated close to the edge of the images were not included in the analysis, as they could have originated from blood vessels outside the field of view. Each dot in the graphs represents the measurement for one cell. The data are plotted as the distance of recombined PDGFRβ+cells ON the vessel (uninjured/control samples) or OFF the vessel (lesion samples) from the nearest vessel surface.

To assess the proportion of PDGFRβ-expressing stromal cells that originates from type A pericytes, the total number of recombined (EYFP+or tdTom+) and non-recombined (EYFP− or tdTom−) PDGFRβ-expressing stromal cells within the lesion core was manually counted and used to calculate the percentage of recombined PDGFRβ-expressing cells out of total PDGFRβ-expressing cells. Likewise, uninjured/control sections were used to calculate the percentage of type A pericytes out of all PDGFRβ-expressing pericytes.

For EAE samples, the number of lesions per section was quantified in three cervical, three thoracic and three lumbar sections per animal spaced 1000 μm apart. A lesion was identified as a large aggregate of PDGFRβ-expressing stromal cells and reactive astrocytes. The anatomical location of the lesion (dorsal, intermediate, ventrolateral, or ventral white matter) was noted and the results were presented as number of lesions per section at the cervical, thoracic or lumbar spinal levels.

To determine the lesion core area covered by fibronectin or collagen I 14 days after cortical and striatal ischemic strokes, five alternate coronal sections per animal containing the ischemic lesion core and spaced 120 μm apart were immunostained for fibronectin, collagen I, GFAP, and DAPI. The ischemic lesions were imaged at low magnification and the areas covered by fibronectin or collagen I signals were thresholded and measured using the ImageJ/Fiji software. The data were averaged and presented as a percentage of the total lesion core area. Likewise, matched areas on the striatum or cortex contralateral to the lesion side were used to quantify fibronectin or collagen I signals under control conditions.

To determine the lesion core volume 14 days after cortico-striatal stab lesions in GLAST-Rasless-EYFP animals, five alternate sagittal sections per animal spanning the lesion center and spaced 120 μm apart were immunostained for PDGFRβ, GFAP, and DAPI. The injury site was imaged at low magnification and the lesion core area, defined as the GFAP negative area within the GFAP-defined borders filled with PDGFRβ +fibrotic cells, was manually outlined using the ImageJ/Fiji software. Area measurements were retrieved and used to calculate the lesion core volume based on the formula: $V = \Sigma A \times T$, in which V = volume, A = lesion core area, T = distance between each sampled region. Similarly, to calculate the lesion core volume occupied by PDGFRβ+ cells, fibronectin or collagen I, the area covered by positive signal was thresholded using the ImageJ/Fiji software and used for volume estimations based on the same formula as above.

Proliferation of recombined (EYFP+) and non-recombined (EYFP-) type A pericytes and progeny (PDGFRβ+ cells) or GFAP+ astrocytes was assessed in

GLAST-Rasless-EYFP and GLAST-EYFP animals at 5 days after cortico-striatal stab lesions. EdU-, PDGFRβ-, GFAP-, and EYFP-expressing cells in the contralateral side to the lesion and within the lesion core were counted in 4–5 sections in 3–4 animals per group and expressed as number of cells per area or as a percentage. Similarly, the number of PDGFRβ-positive type A pericytes and progeny ON vessel or OFF vessel within the lesion core and in the contralateral side to the lesion was quantified in 4–5 sections in 3–4 animals per group and expressed as a percentage of total PDGFRβ-positive cells.

**Statistical analysis.** No statistical methods have been used to predetermine sample size (sample sizes were determined based on previous experience[3,6]). Data are presented as mean ± standard error of the mean (s.e.m.). Individual data points are plotted for most graphs. Sample sizes (n) of animals, number of biological repeats of experiments and statistical methods used are indicated in the corresponding figure legends. Datasets were tested for Gaussian distribution followed by appropriate statistical test. P values were calculated using two-tailed unpaired Student's t test, two-tailed Mann–Whitney test, one way- ANOVA and Kruskal–Wallis test. Post-hoc correction tests were employed following significance with an ANOVA or Kruskal–Wallis test and described in the respective figure legend. Differences were considered statistically significant at p values below 0.05. Source data, exact p values and statistical test results are provided as a Source Data file. All data were analyzed using GraphPad Prism version 6.0 g software for Mac.

**Reporting summary.** Further information on research design is available in the Nature Research Reporting Summary linked to this article.

## Data availability
The authors declare that all data supporting the findings of this study are included in this published article (and its supplementary information files). Source Data for Figs. 1–8 and Supplementary Figs. 4, 5, and 12 are provided with the paper.

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

## Acknowledgements

We thank Göritz lab members for valuable comments on the manuscript. D.O.D. was supported by the Foundation for Science and Technology from the Portuguese government (SFRH/BD/63164/2009). H.B.H. was supported by a grant from the German Research Foundation (Hu1961/2–1). C.G. is a Hållsten Academy and a Knut and Alice Wallenberg Academy Fellow. Research in the C.G. lab was supported by the European Union's Seventh Framework Programme (FP7)/ERC-2012-StG 310938 PERICYTESCAR, Swedish Research Council, Swedish Brain Foundation, Swedish Cancer Society, the Strategic Network for Stem Cells and Regenerative Medicine (STRATREGEN) at Karolinksa Institutet, Ming Wai Lau Centre for Reparative Medicine, JPND DACAPO-AD and Wings for Life Foundation. Multiple sclerosis tissue samples and associated patient clinical and neuropathological data were supplied by the UK Multiple Sclerosis Tissue Bank, supported by the Multiple Sclerosis Society of Great Britain and Northern Ireland, registered charity 207495. We also thank Dr. Brian Kwon and the International Spinal Cord Injury Biobank (ISCIB) located at the University of British Columbia in Vancouver, BC, Canada for providing tissue sections from spinal cord injured patients for histological analysis.

## Author contributions

D.O.D., J.K., Y.K., C.P.E., J.T., L.J., D.H., A.E., and S.B. performed experiments and/or analyses. H.B.H. and L.B. supplied human tissue. D.O.D. and C.G. designed experiments. L.B., Z.K., O.L., H.B.H., M.A.-M., J.F., and C.G. supervised experiments. C.G. coordinated the study. D.O.D. and C.G. wrote the manuscript.

## Funding

## Competing interests

The authors declare no competing interests.
