## [Peer Review File · Nature Communications]

REVIEWER COMMENTS

Reviewer #1 (Remarks to the Author):

This paper is of interest in documenting the similarities and differences by which a subset of pericytes contributes to forming “scar” tissue after many different kinds of lesion. This is a large and important amount of work, despite the results being primarily descriptive with no investigation of mechanisms. The paper’s broad focus establishes the generality of the results for a wide range of insults and for this reason I think it is potentially publishable in Nat Comms. However,

- (1) more quantification of data is needed, especially for cells moving away from the vessel wall;
- (2) controversies over the origins of stromal cells need addressing with a little more labeling work, which would significantly enhance the importance of the work;
- (3) the paper suffers from some problems of clarity of communication which need to be fixed before it could be accepted.

Specific points

Fig 1a: Show that the labeled cells are on vessels, by co-labeling with an endothelial cell antibody or isolectin B4 or imaging with a marker in the blood. Do the type A pericytes have a particular association with capillary branch points or with position in the capillary bed (e.g. near arterioles or near venules)? Quantify the mean distance of the cell soma from the nearest vessel surface, for comparison with the data shown later where they move off the vessel wall.

Fig 1h: Is PDGFRb expressed at all stages of the response to injury in the EYFP cells? Some EYFP+ cells do not appear to express PDGFRb, which implies they are not pericytes. Please show the red and green channels separately.

What are the red and green labels for the panels headed ‘lesion border’ in Fig 1? Also, make it clearer whether the colored text for labels (e.g. PDGFRb, EYFP) refers to the panel above or below the text.

In the text Fig 1j-r is said to quantify the number of EYFP+, PDGFRb+ cells, but on the graphs it says only No. (or %) Rec. PDGFRb+ cells. For the % graph this is ambiguous (not only because I initially assumed the Rec. might mean ‘recorded’ - not all of your readers will expect it to mean ‘recombined’!). It might mean the % of EYFP+ cells that are PDGFRb+ or the % of PDGFRb+ cells that are EYFP+; which is it? This really matters because if the cells are merely defined by PDGFRb expression they could also include type B pericytes. Plot a graph against time after the lesion of the % of EYFP cells which also express PDGFRb, and the % of PDGFRb+ cells which express EYFP (one of these is shown in panel r, but it is unclear which). Label the y axis unambiguously as “% of EYFP+ cells that label for PDGFRb” (or vice versa). Similarly, everywhere in the text that ambiguous phrases like “Percentage of recombined PDGFRb+ cells” (from Fig 5k legend) are used, they should be defined precisely as “Percentage of EYFP+ cells that express PDGFRb...” (or vice versa).

Write scale bar size on all figure panels, not in the legend, to make the reader's task easier.

Page 7: “We did not observe significant contribution from sparsely recombined ependymal cells and radial astrocytes to scar-forming cells”. Explain how this was ruled out.

For Fig 2o how were on and off the vessel wall defined - a rigorous criterion is needed.

In Fig 2h', I': It is not very convincing, especially for I', that the cells arrowed have moved away from the vessel wall, because the vessel wall is poorly labeled. Is a better picture available?

For all the different insult models in the paper it is unfortunate that we are merely invited to inspect images in which blood vessels are not always clearly labeled in order to see how far cells are from the vessels. Please, in addition, quantify the mean distance of the cells’ somata from the nearest vessel surface, so the reader can be more convinced by the shift away from the vessels.

For the spinal lesion experiments, cite Hesp..McTigue et al 2018, J Neurosci
<https://www.ncbi.nlm.nih.gov/pubmed/29279310>

Human data in Figure 6: It is excellent that the study was extended to PM human tissue, but labelling only for PDGFRb does not demonstrate that type A pericytes are involved. The authors should label for EAAT1 (GLAST) as well.

Fig 7: State that the darker regions of the schematics are the lesion areas.

Discussion - comparisons with earlier work.

Involvement of Col1a1 expressing cells. The RNAseq site mousebrain.org/genesearch reports that all 3 classes of pericyte reported there (and it is unclear whether this includes type A pericytes as none of those 3 classes expresses GLAST highly) do express Col1a1, even though much less than "fibroblasts". The authors should use an antibody for Col1a1, to determine whether type A pericytes express this protein.

Involvement of Tbx18-expressing pericytes. Similarly, it should be easy to determine whether type A pericytes in resting tissue and/or PDGFRb expressing cells in lesions express Tbx18.

Reviewer #2 (Remarks to the Author):

The study provided by Dias et al. aims to systematically compare fibrotic scarring in human pathological tissue and corresponding mouse models of penetrating and non-penetrating spinal cord injury, traumatic brain injury, ischemic stroke, multiple sclerosis and glioblastoma.

Using in vivo lineage tracing, the authors observed an accumulation of a subset of PDGFR β + stromal cells, which derived from Type A pericytes (GLAST+), in fibrotic scar tissues of human and mouse models of spinal cord injury, traumatic brain injury, ischemic stroke, multiple sclerosis and glioblastoma. The distribution of PDGFR β + stromal was varied depending on the type of lesion and, in most cases, similar between mice and humans. The authors have shown that pericyte-derived fibrotic scarring is conserved across diverse central nervous system lesions.

Overall, this paper is mainly descriptive, and the authors failed to describe a mechanistic overview for activation of the fibrotic scar. Generally speaking, the introduction is logical, easy to follow, and straightforward. Methods are explained very well and are comprehensive.

Main issue:

This paper does not suggest any mechanisms for the contribution of type A pericytes to fibrotic scar tissue formation. For example:

1. In mouse brain stab lesions the fibrotic scar tissue formed only after larger cortico-striatal injuries, but not after smaller injuries, restricted to the cortex.

Was the fibrotic scar tissue formation due to the larger size of the brain injury? Why didn't the authors compare similar injury sizes between the cortex and cortico-striatum?

2. In the mouse MCAO stroke model, when the ischemic lesion was restricted to the striatum PDGFR β + stromal cells were associated with the blood vessel wall, however, when the ischemic lesion extended from the striatum into the cortex a large number of PDGFR β + cells were away from the blood vessel wall, primarily in the cortex.

Why? Is it owing to the environmental difference between the striatum and cortex?

3. What is the mechanism underlying the specific distribution of PDGFR β + stromal cells in this type of lesion? Does the variable distribution of PDGFR β + stromal cells play different roles on the pathological impact on fibrotic scar tissues?

Minor Point:

For the statistical analysis, any examination for violations to the assumptions for t-tests and ANOVA (equal variances between treatments, homogeneity of variances and normality) should be tested/stated.

Reviewer #3 (Remarks to the Author):

A couple of years ago, the Goritz lab made the pioneering observation that a particular type of pericyte (they called it type A) may be the critical cell that is responsible for the production of fibrotic (fibroblastic-like) scarring with associated inhibitory ECM after spinal cord injury. Targeting this cell to reduce its numbers (but not too greatly) has been shown to promote some regeneration/sprouting and allow for some functional improvements. In this new work, a nicely described survey of the presence of type A pericyte associated fibrotic scarring in a variety of injury and disease states has been presented. They used GLAST-Cre ER/Rosa2EYFP transgenic mice to label this cell type (as well as astrocytes) after surgical as well as contusive spinal cord injury, TBI, ischemic stroke, MS (an EAE model) and glioblastoma. They show some similarities but also differences in the number and distribution of type A pericytes depending on the type of lesion. In addition, some human pathological tissue from individuals with equivalent types of injuries or diseases were analyzed for the possibility of type A pericytes also contributing to fibrosis. The authors conclude that pericyte-derived fibrosis is a conserved mechanism that could be explored in the future as a therapeutic target. Although this new work certainly warrants publication in Nature Communications, I have a few major concerns and questions that need to be addressed.

The author's comment and its implications about the validity of the Soderblom et al., paper (from the Jae Lee lab) about Col1a1+ perivascular fibroblasts perhaps possibly not being a constituent of fibrotic scar is confusing. Do the present authors agree or disagree with the colla1 story? Are these bonafide Col1a1 producing fibroblasts the same or a different cell type than type A pericytes identified in the present story? The authors of the Soderblom et al., papers suggest that NG2 + pericytes are not the source of colla1 fibroblasts. They suggest that cells that are already fibroblasts (not pericytes that differentiate into fibroblasts) are the source of fibrosis after SCI. Do Glast+ type A pericytes express the well-known pericyte NG2 proteoglycan? If they do or do not this would be important to know because there is evidence in the literature (Soderblom et al) that NG2 + pericytes do not contribute to fibrotic scar matrix after stab or contusive SCI.

An additional paper (Guimaraes-Camboa et al., Cell Stem Cell 2018 cited) also questions the role of pericytes in fibrosis, especially regarding the production of inhibitory ECM. Yet another paper by Roth et al., (J Neurosci Res 2019; not cited in the present paper) also questions the role of pericytes in collagen 1 and fibronectin production in a stroke model. Thus, since collagen 1 is a critical molecule in the formation of basal lamina and fibrotic tissue scarring the question remains whether type A pericytes contribute to collagen /ECM formation or not. Do type A pericytes make a variety of ECM molecules in the various models that have been studied here? Some immunohistological evidence in support of type A pericytes making abundant matrix is critical since ECM is a fundamental property of fibrotic scarring. If type A pericytes do not contribute to the production of inhibitory scar matrix then what is their role beyond their function in the support of blood vessels?

I also have a general question about GLAST as a marker of pericytes. Can you comment on how you discovered the presence of the well characterized astrocyte glutamate transporter GLAST in pericytes? Has GLAST been substantiated as a faithful pericyte marker by other labs that have done careful single cell analyses of blood vessel associated cells? In a recent paper by Vanlandewijck et al., (Nature ,2018) their exhaustive molecular atlas of cell types associated with brain vasculature did not find that GLAST is a pericyte marker nor did they document multiple subtypes of pericytes (a point they stressed), at least in the brain. They did, however, identify Col1a1 as a marker of large vessel associated perivascular fibroblasts, similar to those identified by the Lee lab.

In the EAE model of scar formation in regions of type A pericyte accumulation is there concomitant demyelination? The authors talk about demyelinated lesion but I couldn't readily find any evidence that particular lesions that were emphasized were actually demyelinated. While it is likely that such lesions were lacking myelin, staining sections for myelin to actually show this would be relatively easy.

Reviewer #4 (Remarks to the Author):

In this study, Dias et al. investigated fibrotic scarring in human pathological tissue and corresponding mouse models of penetrating and non-penetrating spinal cord injury, traumatic brain injury, ischemic stroke, multiple sclerosis and glioblastoma. They tried to determine the cellular origin of the fibrotic tissue scar in these spinal or brain lesions by using in vivo cell lineage tracing technique with immunofluorescence, and found that type A pericytes are the major source of scar-forming fibroblasts in many CNS lesions. They showed that the type A pericytes exhibited unique patterns and contributed to the scar formation after different CNS lesions. Overall, this is an interesting study. However, because current data are very preliminary, the major conclusions should be further validated in the pericyte-deficient mice (PDGFR β conditional knockout mice), and the molecular mechanisms underlying the recruitment of pericytes to the lesion sites and the differentiation of type A pericytes into fibroblasts to form the scars should be explored and defined. Current study is largely descriptive without mechanistic insights to support specific conclusions.

Major concerns:

1. To further confirm that pericytes are critical cells that contribute to the scar formation after brain lesions, brain pericyte-deficient mice (PDGFR β conditional knockout mice) should be used to examine whether pericyte deficiency attenuates the scar formation after brain lesions, because such mice are available for the study.
2. It is not clear how the type A pericytes are recruited to the lesion sites. What is the molecular signaling involved in this? The molecular mechanisms should be investigated.
3. Do these pericytes proliferate after they are recruited to the lesion sites or before the recruitment? If yes, how does the CNS lesion trigger pericyte proliferation? The molecular mechanisms underlying this should be determined.
4. Do the recruited type A pericytes differentiate into fibroblasts to form the scar after they are recruited to the lesion sites? If yes, what are the molecular mechanisms underlying the differentiation.
5. It is well recognized that there is no fibroblast in the brain. It is not clear how they define the fibroblasts in the scars after brain lesions. More fibroblast markers should be used to validate the cells in the scars.
6. It is not clear why only type A pericytes contribute to the scar formation. What is unique property of type A pericytes for this function? What are the major difference between type A pericytes and other pericytes in the brain? More pericyte markers should be used to define these pericytes.
7. What are other cellular components beside the pericyte-derived fibroblasts? Are the astrocytes involved in the scar formation after injury? What are the fractions of the pericyte-derived fibroblasts in the scars after different lesions?
8. The cell lineage tracing system (EYFP+) was used in this study, but why the EYFP+ cells were identified by immunostaining of GFP as indicated in the method section is not clear.
9. In figures 1, 2, and 4, it is necessary to show the representative images for the distribution and location of the type A pericytes 7 weeks following spinal cord injury beside the quantified analysis.
10. In most figures, the nuclei were not labeled with DAPI, which makes very hard to determine and quantify type A pericytes and other cells.
11. Most figure legends were poorly described and lack critical information.

Minor points:

1. In figures 3d, it is not clear what dWM, iWM, vIWM, and vWM stand for.
2. In figures 3 and 4, the damaged areas were not clearly indicated. It would be necessary to indicate the damage areas with some markers such as cd45 or cd3.
3. In figures 1g, 3g, 5d and 6, how did they define those GFAP+ astrocytes are the reactive astrocytes? GFAP+ is a general marker for most astrocytes.

Response to comments from the reviewers

Reviewer #1 (Remarks to the Author):

This paper is of interest in documenting the similarities and differences by which a subset of pericytes contributes to forming “scar” tissue after many different kinds of lesion. This is a large and important amount of work, despite the results being primarily descriptive with no investigation of mechanisms. The paper’s broad focus establishes the generality of the results for a wide range of insults and for this reason I think it is potentially publishable in Nat Comms. However,

(1) more quantification of data is needed, especially for cells moving away from the vessel wall;

(2) controversies over the origins of stromal cells need addressing with a little more labeling work, which would significantly enhance the importance of the work;

(3) the paper suffers from some problems of clarity of communication which need to be fixed before it could be accepted.

We were happy to read that the reviewer finds our study of interest and comments on the density and importance of the data. We also thank the reviewer for the positive feedback on the manuscript being potentially publishable in Nature Communications. We are grateful for the comments as they helped us to clarify the raised points, which improved the manuscript.

Specific points

Fig 1a: Show that the labeled cells are on vessels, by co-labeling with an endothelial cell antibody or isolectin B4 or imaging with a marker in the blood.

We have now added new images to figure 1 (figure 1b), as well as figures 2b, 5b and supplementary figure 4b, showing that type A pericytes are associated with the blood vessel wall in the uninjured spinal cord and brain, by co-labeling with the endothelial cell marker podocalyxin (Pdx).

In addition, we show that type A pericytes are on blood vessels marked by podocalyxin in supplementary figures 1 and 2.

Do the type A pericytes have a particular association with capillary branch points or with position in the capillary bed (e.g. near arterioles or near venules)?

We now supplement new images and description in the text to characterize the location of type A pericytes along the vascular tree. We show that in the uninjured spinal cord type A pericytes are associated with blood vessels throughout the grey and white matter and essentially distribute along the entire capillary bed, and upstream venous and arterial vasculature (supplementary figures 1b-d). Type A pericytes with similar distribution along the vasculature can also be observed in the mouse cerebral cortex and striatum (supplementary figures 2a-c).

Quantify the mean distance of the cell soma from the nearest vessel surface, for comparison with the data shown later where they move off the vessel wall.

As suggested by the reviewer, we have quantified the mean distance of type A pericytes' cell soma from the nearest vessel surface in the uninjured spinal cord. For comparison with uninjured data, we have also measured the distance of the cell soma of type A pericyte progeny not associated with the vessel wall from the nearest vessel surface for all time points investigated after spinal cord injury. The data is presented in figure 1v of the revised manuscript. In the graph, each single dot represents the measurement for every individual cell analyzed. The mean distance from all cells measured is also displayed.

We have extended these analyses to all main lesion models investigated in the manuscript and present the data in the revised main figures.

Fig 1h: Is PDGFRb expressed at all stages of the response to injury in the EYFP cells? Some EYFP+ cells do not appear to express PDGFRb, which implies they are not pericytes. Please show the red and green channels separately.

Nearly all EYFP⁺ type A pericyte-derived cells in the lesion core express PDGFR β at all stages of the injury response. We adjusted the contrast/brightness of former figure 1h (figure 1j of the revised manuscript) to more clearly demonstrate the overlap between the EYFP (green) and PDGFR β (red) channels. We also show the red and green channels separately in supplementary figure 5a.

As stated in the manuscript there is occasional recombination of radial astrocytes and ependymal cells in the spinal cord of GLAST-CreER^{T2};Rosa26-EYFP mice. However, astrocytes, ependymal cells and their respective progeny do not participate in fibrotic scar tissue formation (Barnabé-Heider et al. Cell Stem Cell 2010) and do not express the stromal marker PDGFR β (see Figure S10 of Göritz et al. Science 2011), which allows us to distinguish them from type A pericyte-derived scar-forming fibroblasts.

What are the red and green labels for the panels headed 'lesion border' in Fig 1? Also, make it clearer whether the colored text for labels (e.g. PDGFRb, EYFP) refers to the panel above or below the text.

We thank the reviewer for this comment. We have clearly labeled all the figure panels and increased the space between figure rows in the revised figures.

The panel headed 'lesion border' in former in figure 1 was moved to supplementary figure 6b,c in the revised manuscript. The labels for these panels are now clearly indicated: supplementary figure 6b shows GFAP in red and EYFP in green and supplementary figure 6c shows PDGFR β in red and EYFP in green.

In the text Fig 1j-r is said to quantify the number of EYFP+, PDGFRb+ cells, but on the graphs it says only No. (or %) Rec. PDGFRb+ cells. For the % graph this is ambiguous (not only because I initially assumed the Rec. might mean 'recorded' - not all of your readers will expect it to mean 'recombined'!). It might mean the % of EYFP+ cells that are PDGFRb+ or the % of PDGFRb+ cells that are EYFP+; which is it?

We thank the reviewer for this comment. We have replaced the label "Rec. cells" from the graphs' y-axis by "EYFP⁺ cells" to avoid ambiguity.

In former figure 1r (figure 1w of the revised manuscript) we show the percentage of PDGFR β ⁺ cells that are EYFP⁺. In other words, the graph shows the % of PDGFR β ⁺ cells that express EYFP (EYFP⁺PDGFR β ⁺ cells) out of total PDGFR β ⁺ cells in the lesion core. The y-axis of the graph was altered accordingly. We altered the y-axis labeling of all similar graphs throughout the figures of the manuscript.

This really matters because if the cells are merely defined by PDGFRb expression they could also include type B pericytes. Plot a graph against time after the lesion of the % of EYFP cells which also express PDGFRb, and the % of PDGFRb+ cells which express EYFP (one of these is shown in panel r, but it is unclear which).

As stated above, former panel 1r (figure 1w of the revised manuscript) shows the % of PDGFR β ⁺ cells which express EYFP in the lesion core.

Virtually all EYFP⁺ cells in the lesion core express PDGFR β at all time points investigated after the lesion. As stated in response to a reviewer's comment above, there is occasional recombination of radial astrocytes and ependymal cells in the spinal cord of GLAST-CreER^{T2};Rosa26-EYFP mice, but astrocyte- and ependymal-derived cells contribute to the glial scar that surrounds the lesion core (Barnabé-Heider et al. Cell Stem Cell 2010). Therefore, those occasional EYFP⁺ cells that do not express the stromal marker PDGFR β locate outside the lesion core and do not interfere with our quantifications, which focus on EYFP⁺ cells in the fibrotic lesion core.

Label the y axis unambiguously as “% of EYFP+ cells that label for PDGFRb” (or vice versa). Similarly, everywhere in the text that ambiguous phrases like “Percentage of recombined PDGFRb+ cells” (from Fig 5k legend) are used, they should be defined precisely as “Percentage of EYFP+ cells that express PDGFRb...” (or vice versa).

We have re-labeled all the y-axis throughout all the figures of the revised manuscript for clarity. We have also revised the text and figures legends for clarity.

Write scale bar size on all figure panels, not in the legend, to make the reader's task easier.

We have followed the guidelines of the journal when submitting the paper which are quoted below:

“Scale bars should be used rather than magnification factors, with the length of the bar defined in the legend rather than on the bar itself.”

Page 7: “We did not observe significant contribution from sparsely recombined ependymal cells and radial astrocytes to scar-forming cells”. Explain how this was ruled out.

We apologize for lack of clarity in the sentence. We tried to convey “We did not observe significant contribution from sparsely recombined ependymal cells and radial astrocytes to PDGFR β ⁺ scar-forming cells in the fibrotic compartment of the scar”.

As stated in response to a reviewer's comment above, ependymal cell-derived progeny and radial astrocyte-derived cells participate in glial scar formation and are not present at the lesion core (Barnabé-Heider et al. Cell Stem Cell 2010). Therefore, they do not contribute to the fibrotic scar. Additionally, they do not express the stromal marker PDGFR β (see Figure

S10 of Göritz et al. Science 2011), allowing us to differentiate them from type A pericyte-derived scar-forming cells.

In the revised manuscript text we have expanded on this topic as quoted below:

“Although astrocyte- and ependymal cell-derived progeny participate in glial scar formation, they do not contribute to the fibrotic compartment of the scar (Barnabé-Heider et al. Cell Stem Cell 2010) and do not express the stromal marker PDGFR β . Virtually all recombined cells found within the fibrotic scar expressed the stromal marker PDGFR β (Fig. 1j,o and Supplementary Fig. 4i,n), ruling out significant contribution from sparsely recombined ependymal cells and radial astrocytes to fibrotic scar-forming cells, as previously established (Göritz et al Science 2011).”

For Fig 2o how were on and off the vessel wall defined - a rigorous criterion is needed.

We have updated the methods section with a more detailed description of the criteria used to define a cell ON the vessel or OFF the vessel, as quoted below:

“To determine the proportion of recombined cells either associated with or dissociated from the blood vessel wall within the lesion core, tissue sections were immunostained for the endothelial markers CD31 or podocalyxin, GFP, PDGFR β and DAPI. Four randomly selected fields (0.83mm x 0.66 mm) per section were photographed and used for analysis. A cell was considered OFF the vessel only when the cell soma or processes had no direct contacts with the nearest blood vessel surface and the center of the nucleus of the cell is at a distance $>8\mu\text{m}$ from the edge of the closest blood vessel surface. Cells with any contact points (soma or processes) to the nearest blood vessel surface and with the center of the nucleus of the cell at distance $\leq 8\mu\text{m}$ from the edge of the closest blood vessel surface were considered ON the vessel. The number of recombined PDGFR β -positive cells in contact with blood vessel wall (ON the vessel) or away from the blood vessel wall (OFF the vessel) was manually counted and the data expressed as a percentage of total recombined PDGFR β -expressing cells. The percentage of type A pericytes ON or OFF the vessels under uninjured/control conditions was assessed in a similar fashion.”

Using spinal cord injury as an example, the data shows that under uninjured conditions, the center of the nucleus of type A pericytes sits ON the vessel at an average distance of 3-5 μm (with maximum data points up to 6-8 μm). After spinal cord injury, the center of the nucleus of type A pericyte-derived cells sits OFF the vessel at an average distance of 30 μm (with smaller data points as low as 9-10 μm). We have therefore chosen a distance of 8 μm as additional criteria to clearly distinguish between cells ON the vessel and OFF the vessel.

In Fig 2h', l': It is not very convincing, especially for l', that the cells arrowed have moved away from the vessel wall, because the vessel wall is poorly labeled. Is a better picture available?

We have replaced the image in figure 2i and 2i' (figures 2k and 2k' in the revised manuscript) with a new image where the blood vessel wall is clearly labeled. In figure 2h' (figure 2j' in the revised manuscript) we are not showing that the recombined cell has moved away from the vessel wall, but rather that develops protrusion-like structures.

For all the different insult models in the paper it is unfortunate that we are merely invited to inspect images in which blood vessels are not always clearly labeled in order to see how far cells are from the vessels. Please, in addition, quantify the mean distance of the cells' somata from the nearest vessel surface, so the reader can be more convinced by the shift away from the vessels.

We have quantified the distance of type A pericytes' cell soma from the nearest vessel surface in uninjured/control conditions and the distance of the cell soma of type A pericyte progeny that moved off the vessel wall from the nearest vessel surface at all time points investigated after the different lesions. The data is presented in figures 1v, 2v, 3w, 4x, 5w, 6w, 7q and supplementary figure 4u. In the graphs, each single dot represents the measurement for every individual cell analyzed. The mean distance from all cells measured is also displayed.

For the spinal lesion experiments, cite Hesp..McTigue et al 2018, J Neurosci <https://www.ncbi.nlm.nih.gov/pubmed/29279310>

As suggested by the reviewer, we have now cited the paper in conjunction with spinal lesion experiments.

Human data in Figure 6: It is excellent that the study was extended to PM human tissue, but labelling only for PDGFRb does not demonstrate that type A pericytes are involved. The authors should label for EAAT1 (GLAST) as well.

We thank the reviewer for raising this very important question.

We have used RNAscope *in situ* hybridization for detection of *SLC1A3* (GLAST/EAAT1) and *PDGFRB* mRNA in healthy human spinal cord and occipital cortex *post mortem* tissue, in combination with immunofluorescence for the UEA-1 lectin, which labels human endothelial cells. We found many GLAST⁺PDGFRβ⁺ perivascular cells, defined as blood vessel-associated cells positive for *PDGFRB* and *SLC1A3* mRNA signals, suggesting that the human vasculature may contain cells that are similar to type A pericytes in the mouse. These data were added to the text and presented in figure 9.

Unfortunately, GLAST is downregulated in type A pericyte-derived cells after injury, as we observed after spinal cord injury in the mouse. Therefore, labeling for GLAST cannot be used to faithfully identify PDGFRβ⁺ stromal cells equivalent to type A pericyte-derived cells in *post mortem* human pathological tissue.

Fig 7: State that the darker regions of the schematics are the lesion areas.

We have now indicated in the schematics and in the figure legend that the darker areas represent the lesion areas.

Discussion - comparisons with earlier work.

Involvement of Col1a1 expressing cells. The RNAseq site mousebrain.org/genesearch reports that all 3 classes of pericyte reported there (and it is unclear whether this includes type A pericytes as none of those 3 classes expresses GLAST highly) do express Col1a1, even though much less than “fibroblasts”. The authors should use an antibody for Col1a1, to determine whether type A pericytes express this protein.

Due to the difficulty in finding a reliable and specific antibody for Col1 α 1, we have chosen to use an antibody for collagen type I, which recognizes the α 1 and α 2 chains of collagen type I. We observed that type A pericytes and progeny were intimately surrounded by collagen type I in the lesion core and surrounding tissue after spinal cord injury and present this new data in supplementary figure 5c.

Involvement of Tbx18-expressing pericytes. Similarly, it should be easy to determine whether type A pericytes in resting tissue and/or PDGFRb expressing cells in lesions express Tbx18.

We show by RNA scope *in situ* hybridization that type A pericytes express *Tbx18* in the uninjured brain and spinal cord. Also, type A pericyte-derived cells kept expression of *Tbx18* after large cortical stab lesions and after spinal cord injury. These data were included in supplementary figures 1i, 2h, 5d and 9 of the revised manuscript. In contrast to small cortical stab lesions that do not generate extensive fibrotic scar tissue, we suggest in the discussion of the manuscript that *Tbx18*-expressing perivascular cells may contribute to CNS fibrosis in lesion models that generate substantial fibrotic scar tissue such as large cortical stab wounds or cortico-striatal stab wounds, as we have showed for type A pericytes (figures 2 and 3).

Reviewer #2 (Remarks to the Author):

The study provided by Dias et al. aims to systematically compare fibrotic scarring in human pathological tissue and corresponding mouse models of penetrating and non-penetrating spinal cord injury, traumatic brain injury, ischemic stroke, multiple sclerosis and glioblastoma.

Using in vivo lineage tracing, the authors observed an accumulation of a subset of PDGFR β + stromal cells, which derived from Type A pericytes (GLAST+), in fibrotic scar tissues of human and mouse models of spinal cord injury, traumatic brain injury, ischemic stroke, multiple sclerosis and glioblastoma. The distribution of PDGFR β + stromal was varied depending on the type of lesion and, in most cases, similar between mice and humans. The authors have shown that pericyte-derived fibrotic scarring is conserved across diverse central nervous system lesions.

Overall, this paper is mainly descriptive, and the authors failed to describe a mechanistic overview for activation of the fibrotic scar. Generally speaking, the introduction is logical, easy to follow, and straightforward. Methods are explained very well and are comprehensive.

We thank the reviewer for the positive feedback on the manuscript. We are grateful for the mechanistic questions as they helped us to investigate the raised points, which improved the manuscript.

Main issue:

This paper does not suggest any mechanisms for the contribution of type A pericytes to fibrotic scar tissue formation. For example:

1. In mouse brain stab lesions the fibrotic scar tissue formed only after larger cortico-striatal injuries, but not after smaller injuries, restricted to the cortex. Was the fibrotic scar tissue formation due to the larger size of the brain injury? Why didn't the authors compare similar injury sizes between the cortex and cortico-striatum?

We thank the reviewer for this question.

As suggested by the reviewer, we examined whether the lesion size influence fibrotic scar tissue generation. For that we fate mapped type A pericytes and compared fibrotic scar tissue formation between cortico-striatal stab lesions and stab lesions of the same size and depth restricted to the cerebral cortex, referred to as large cortical stab wound. In contrast to small cortical lesions (supplementary figure 8), large cortical stab wounds generated robust fibrotic scar tissue and type A pericyte-derived cells contributed to ~70% of all scar-forming PDGFR β + stromal cells in this injury paradigm. These results are presented in the new figure 3 of the revised manuscript. We conclude that larger mechanical injuries generate more robust fibrotic scarring when compared to smaller injuries and that fibrotic scar tissue formation enriched in PDGFR β + stromal cells is only triggered by large mechanical injuries to the cerebral cortex.

In addition, these experiments allowed us to investigate whether the anatomical location of the lesion (cortex versus cortico-striatum) influence fibrotic scar tissue generation.

Large cortical and cortical-striatal lesions resulted in a comparable stromal cell density after scar condensation at 14 days after injury. However, the dynamics of type A pericyte recruitment and resolution of fibrotic scar tissue differed between these 2 lesion models.

Despite the initial mechanical injury (lesion depth and size) being similar, large cortical stab lesions developed larger fibrotic lesion cores and triggered more widespread type A pericyte recruitment when compared to cortico-striatal stab wounds, as observed at 5 days after injury. This initial difference was compensated by a more efficient resolution of fibrotic cells during scar maturation after cortical wounds compared to cortico-striatal wounds. The results are presented in figure 2 and 3 of the revised manuscript.

2. In the mouse MCAO stroke model, when the ischemic lesion was restricted to the striatum PDGFR β ⁺ stromal cells were associated with the blood vessel wall, however, when the ischemic lesion extended from the striatum into the cortex a large number of PDGFR β ⁺ cells were away from the blood vessel wall, primarily in the cortex. Why? Is it owing to the environmental difference between the striatum and cortex?

To address this question, we included another ischemic lesion model, in which the primary damage is restricted to the cortex and compared the results to the already included striatal stroke (figure 5) and cortico-striatal stroke (supplementary figure 10) lesion models. The results for the newly added cortical lesion model are presented in figure 6 and described in the text under “Type A pericytes contribute to stromal cells following ischemic lesions to the brain” of the revised manuscript. For the comparison between ischemic lesions restricted to the striatum and cortex, we added a new section in the revised manuscript “Type A pericytes are required for extracellular matrix deposition after CNS lesions”. The corresponding data are presented in the new figure 8.

We found that in contrast to striatal stroke lesions, where most of the PDGFR β ⁺ stromal cells are associated with the vascular wall, a large number of PDGFR β ⁺ stromal cells located outside the blood vessel wall after cortical ischemic lesions, indicating that environmental differences between cortex and striatum influence the formation of fibrotic scar tissue formation. Importantly, in both lesion models, type A pericytes contributed substantially to PDGFR β ⁺ stromal populations.

Furthermore, we extended the analysis of cortico-striatal stroke lesions, including now a 14 days time point in addition to the 5 days time point that was included previously. The new results can be found in supplementary figure 10. In contrast to ischemic lesions restricted to the striatum, cortico-striatal lesions show many PDGFR β ⁺ stromal cells at a distance from the blood vessel wall in the striatum at 5 days and 14 days. The vast majority of these stromal cells were recombined and, therefore, originated from type A pericytes. These data suggest that environmental factors associated with the cerebral cortex and/or adjacent brain structures (*e.g. corpus callosum*) may influence type A pericyte recruitment from the blood vessel wall in the striatum.

Collectively, our results suggest that the location and magnitude of the ischemic insult dictate the distribution of type A pericyte progeny and generation of stromal fibroblasts.

3. What is the mechanism underlying the specific distribution of PDGFR β ⁺ stromal cells in this type of lesion? Does the variable distribution of PDGFR β ⁺ stromal cells play different roles on the pathological impact on fibrotic scar tissues?

We addressed whether the variable distribution of type A pericyte-derived stromal cells in relation to the blood vessel wall influences the pathological outcome of tissue fibrosis, such as deposition of the fibrotic ECM proteins collagen I and fibronectin. For that we compared ischemic stroke lesions confined to the striatum and stroke lesions restricted to the cerebral cortex, which exhibit clear differences in type A pericyte response to ischemia.

In striatal lesions, in which type A pericytes remained attached to the vascular wall, we observed a slight but not significant increase in the deposition of fibronectin and collagen I in the ischemic lesion core at 14 days after stroke, when compared to the contralateral stroke side. In sharp contrast, cortical ischemic lesions, with extensive, non-vessel associated type A pericyte-derived stromal fibroblasts, led to a vigorous increase in deposition of fibronectin and collagen I in comparison to striatal strokes and to the contralateral stroke side. These results, presented in the new figure 8a-e, indicate that detachment of type A pericyte-derived cells from the vascular wall, which leads to the generation of stromal fibroblasts, is required for the deposition of fibronectin and collagen I ECM proteins.

To validate that type A pericytes are involved in deposition of ECM in an independent lesion model in the brain, we employed a cell-specific and inducible genetic strategy to inhibit the generation of progeny by type A pericytes, via specifically blocking injury-induced proliferation. We previously established this strategy after spinal cord injury (Göritz et al., Science 2011 ; Dias et al., Cell 2018). We observed that control animals, where type A pericytes proliferate and leave the blood vessel wall extensively, developed a dense fibrotic lesion core enriched in PDGFR β -expressing stromal cells and widespread deposition of fibronectin and collagen I-rich ECM at 14 days after cortico-striatal stab lesions. On the contrary, animals in which type A pericyte proliferation was reduced and a higher percentage of cells remained attached to the blood vessel wall, presented significant attenuation of fibrotic scar tissue formation, revealed as smaller lesions with decreased PDGFR β ⁺ scarring compared to control animals. In these animals, we observed reduced deposition of fibronectin and collagen I compared to control animals, as observed previously after spinal cord injury (Göritz et al., Science 2011 ; Dias et al., Cell 2018). These new results are now described in the text under the section “Type A pericytes are required for extracellular matrix deposition after CNS lesions” and presented in the new figure 8g-l and supplementary figure 11.

Taken together, our results show that the number and location of type A pericytes and progeny in relation to the vascular wall influence the deposition of collagen I and fibronectin ECM proteins after CNS lesions.

Minor Point:

For the statistical analysis, any examination for violations to the assumptions for t-tests and ANOVA (equal variances between treatments, homogeneity of variances and normality) should be tested/stated.

Raw data were tested for Gaussian distribution and appropriate parametric or non-parametric statistical tests were employed for statistical analyses, as specified in the figure legends. Prior to ANOVA, to test for equal variances, we have verified that the standard deviations are not significantly different among the groups compared using the Brown-Forsythe test.

We have updated the methods section of the manuscript under “Statistical Analyses”.

Reviewer #3 (Remarks to the Author):

A couple of years ago, the Goritz lab made the pioneering observation that a particular type of pericyte (they called it type A) may be the critical cell that is responsible for the production of fibrotic (fibroblastic-like) scarring with associated inhibitory ECM after spinal cord injury. Targeting this cell to reduce its numbers (but not too greatly) has been shown to promote some regeneration/sprouting and allow for some functional improvements. In this new work, a nicely described survey of the presence of type A pericyte associated fibrotic scarring in a variety of injury and disease states has been presented. They used GLAST-Cre ER/Rosa2EYFP transgenic mice to label this cell type (as well as astrocytes) after surgical as well as contusive spinal cord injury, TBI, ischemic stroke, MS (an EAE model) and glioblastoma. They show some similarities but also differences in the number and distribution of type A pericytes depending on the type of lesion. In addition, some human pathological tissue from individuals with equivalent types of injuries or diseases were analyzed for the possibility of type A pericytes also contributing to fibrosis. The authors conclude that pericyte-derived fibrosis is a conserved mechanism that could be explored in the future as a therapeutic target. Although this new work certainly warrants publication in Nature Communications, I have a few major concerns and questions that need to be addressed.

We thank the reviewer for the positive feedback on the manuscript and the encouragement for publication in Nature Communications. We are grateful for the comments as they helped us to clarify the raised points, which improved the manuscript.

The author's comment and its implications about the validity of the Soderblom et al., paper (from the Jae Lee lab) about Col1a1+ perivascular fibroblasts perhaps possibly not being a constituent of fibrotic scar is confusing. Do the present authors agree or disagree with the colla1 story?

Are these bonafide Col1a1 producing fibroblasts the same or a different cell type than type A pericytes identified in the present story?

We thank the reviewer for raising this point for clarification.

The unclear statements in the Soderblom et al., J Neurosci 2013 publication, especially in the abstract, have unfortunately caused confusion in the field. Soderblom et al. did not lineage trace Col1 α 1⁺ cells but instead used a Col1 α 1-GFP reporter line, which reflects Col1 α 1 transcriptional activity. The authors show that a small population of perivascular cells located at larger vessels, which they classify as fibroblasts, expresses Col1 α 1. Furthermore, they also show that there is an increase in Col1 α 1-producing cells following contusive spinal cord injury. However, they do not establish a lineage relationship between the Col1 α 1 expressing perivascular cells prior to injury and Col1 α 1 expressing cells in the fibrotic scar. The paper only contains lineage tracing of NG2⁺ cells but unfortunately is written in a way that the two experiments can be confused.

We agree with the finding that stromal fibroblasts in the lesion core express Col1 α 1 after spinal cord injury. Indeed, we observed that type A pericytes and progeny were embedded in collagen I-positive ECM in the lesion core and surrounding tissue after spinal cord injury and present this new data in supplementary Fig. 5c.

We have rephrased the discussion in our revised manuscript for clarification.

The authors of the Soderblom et al., papers suggest that NG2 + pericytes are not the source of colla1 fibroblasts. They suggest that cells that are already fibroblasts (not pericytes that differentiate into fibroblasts) are the source of fibrosis after SCI. Do Glst+ type A pericytes express the well-known pericyte NG2 proteoglycan? If they do or do not this would be important to know because there is evidence in the literature (Soderblom et al) that NG2 + pericytes do not contribute to fibrotic scar matrix after stab or contusive SCI.

We have looked into NG2 expression by type A pericytes and progeny after injury and present the data in supplementary figure 5b. While we found type A pericytes and progeny co-expressing the NG2 proteoglycan, the majority of recombined cells do not share this marker after spinal cord injury.

An additional paper (Guimaraes-Camboa et al., Cell Stem Cell 2018 cited) also questions the role of pericytes in fibrosis, especially regarding the production of inhibitory ECM.

Guimaraes-Camboa et al., Cell Stem Cell 2018 used a Tbx18-CreER^{T2} line to fate map pericytes and smooth muscle cells after small stab wounds restricted to the cerebral cortex. The authors concluded that pericytes do not contribute to fibrotic scar tissue after brain injury. However, as pointed out by the authors, no substantial fibrotic scar tissue is generated in this lesion model and it is therefore expected to observe little/no recruitment of Tbx18-lineage cells after discrete cortical stab lesions. Our results employing a similar injury paradigm are presented in supplementary figure 8 and corroborate their findings. Indeed, we observed no substantial fibrotic tissue formation after small cortical stab wounds, hence no contribution of type A pericytes.

In contrast, larger cortico-striatal stab lesions (figure 2) and larger stab lesions restricted to the cerebral cortex (new figure 3 added to the revised manuscript) triggered robust fibrotic and glial scarring. We show that the majority of scar-forming PDGFR β ⁺ stromal cells generated in these lesion models originates from type A pericytes. Moreover, we show that type A pericyte-derived cells are intimately surrounded by dense deposits of collagen I and fibronectin after large brain lesions (figures 2q,r and 3p,q), suggesting that these cells may be involved in production of these ECM proteins.

In addition, we have used RNAscope *in situ hybridization* and show that type A pericytes in the uninjured brain and spinal cord, as well as their progeny, after spinal cord injury and large brain stab lesions express *Tbx18*. These data is now included in the revised manuscript and presented in supplementary figures 1i, 2h, 5d and 9.

Collectively, our results suggest that Tbx18-expressing perivascular cells contribute to CNS fibrosis in lesion models that generate substantial fibrotic scar tissue. We have rephrased the discussion in our revised manuscript for clarification.

Yet another paper by Roth et al., (J Neurosci Res 2019; not cited in the present paper) also questions the role of pericytes in collagen 1 and fibronectin production in a stroke model. Thus, since collagen 1 is a critical molecule in the formation of basal lamina and fibrotic tissue scarring the question remains whether type A pericytes contribute to collagen /ECM formation or not. Do type A pericytes make a variety of ECM molecules in the various models that have been studied here? Some immunohistological evidence in support of type A pericytes making abundant matrix is critical since ECM is a fundamental property of fibrotic scarring. If type A pericytes do not contribute to the production of inhibitory scar matrix then what is their role beyond their function in the support of blood vessels?

We have now added histological evidence across the different lesions models that type A pericyte-derived stromal cells are embedded in collagen I- and fibronectin- rich ECM. These new data was added to the manuscript text and can be found in figures 1-7 and supplementary figure 4 of the revised manuscript.

We also addressed whether the variable distribution of type A pericyte-derived stromal cells in relation to the blood vessel wall influences the pathological outcome of tissue fibrosis, such as deposition of fibrotic ECM proteins. For that we compared ischemic stroke lesions confined to the striatum and stroke lesions restricted to the cerebral cortex, which exhibit clear differences in type A pericyte response to ischemia. In striatal lesions, in which type A pericytes remained attached to the vascular wall, we observed a slight but not significant increase in the deposition of fibronectin and collagen I in the ischemic lesion core at 14 days after stroke, when compared to the contralateral stroke side. In sharp contrast, cortical ischemic lesions, with extensive, non-vessel associated type A pericyte-derived stromal fibroblasts, led to a vigorous increase in deposition of fibronectin and collagen I in comparison to striatal strokes and to the contralateral stroke side. These results, presented in the new figure 8a-e, indicate that detachment of type A pericyte-derived cells from the vascular wall, which leads to the generation of stromal fibroblasts, is required for the deposition of fibronectin and collagen I ECM proteins.

To validate that type A pericytes are involved in deposition of ECM in an independent lesion model in the brain, we have employed a genetic strategy to inhibit the generation of progeny by type A pericytes, via specifically blocking injury-induced proliferation, as previously established after spinal cord injury (Göritz et al., Science 2011 ; Dias et al., Cell 2018). We observed that control animals, where type A pericytes proliferate and leave the blood vessel wall extensively, developed a dense fibrotic lesion core enriched in PDGFR β -expressing stromal cells and widespread deposition of fibronectin and collagen I-rich ECM at 14 days after cortico-striatal stab lesions. On the contrary, animals in which type A pericyte proliferation was reduced and a higher percentage of cells remained attached to the blood vessel wall, presented significant attenuation of fibrotic scar tissue formation, revealed as smaller lesions with decreased PDGFR β ⁺ scarring compared to control animals. In these animals, we observed reduced deposition of fibronectin and collagen I compared to control animals, as observed previously after spinal cord injury (Göritz et al., Science 2011 ; Dias et al., Cell 2018). These new results are now described in the text under the section "Type A pericytes are required for extracellular matrix deposition after CNS lesions" and presented in the new figure 8g-l and supplementary figure 11.

Taken together, our results show that the number and location of type A pericytes and progeny in relation to the vascular wall influence the deposition of collagen I and fibronectin ECM proteins after CNS lesions.

The conclusions drawn by Roth et al., J Neurosci Res 2019 are difficult to compare to our experiments. I would like to point out that the authors use conventional, non-conditional, knock out mice for *Rgs5*, which seems to alter CNS scarring in several ways. Deletion of *Rgs5* impacts not only on pericytes, but also astrocytes, as indicated by the reduced thickening of

the vascular basal membrane and alterations in the number and density of GFAP⁺ cells and the thickness of the glial scar in RGS5-KO compared to WT mice in response to stroke. Since the glial scar intimately borders the ischemic lesion core and there is crosstalk between reactive astrocytes and fibrotic cells, as we show in supplementary figure 6, a less pronounced glial scar may impact on fibrotic scar tissue generation and ECM deposition.

I also have a general question about GLAST as a marker of pericytes. Can you comment on how you discovered the presence of the well characterized astrocyte glutamate transporter GLAST in pericytes?

The GLAST-CreER^{T2} transgenic line used in this study was created in the laboratory of Frank Pfrieger, where I performed my PhD. I was involved in the initial characterization of this line (Slezak et al., *Glia*. 2007 Nov 15;55(15):1565-76.). The line was intended to specifically target astrocytes. However, during the characterization I noticed that the line did not faithfully represent the GLAST expression in astrocytes in the spinal cord. Due to the near complete absence of recombined astrocytes in the spinal cord, the recombination in a small subset of perivascular cells became obvious. Later, I confirmed that this mouse line truthfully reflects GLAST expression in a subset of perivascular cells (type A pericytes) when comparing it to the knock-in GLAST-CreER line generated by Magdalena Götz (Mori et al., *Glia*. 2006 Jul;54(1):21-34.), which targets the same cells (in addition to astrocytes).

Has GLAST been substantiated as a faithful pericyte marker by other labs that have done careful single cell analyses of blood vessel associated cells? In a recent paper by Vanlandewijck et al., (Nature ,2018) their exhaustive molecular atlas of cell types associated with brain vasculature did not find that GLAST is a pericyte marker nor did they document multiple subtypes of pericytes (a point they stressed), at least in the brain. They did, however, identify Colla1 as a marker of large vessel associated perivascular fibroblasts, similar to those identified by the Lee lab.

We thank the reviewer for raising this matter.

We have carefully looked into the dataset generated by Vanlandewijck et al., *Nature*, 2018 using the online database accessible at

<<http://betsholtzlab.org/VascularSingleCells/database.html>>.

The expression of GLAST (gene name *Slc1a3*) can be clearly detected in a fraction of pericytes, confirming our observations that GLAST is only expressed in a fraction of perivascular cells (and not all perivascular cells) and GLAST-expressing type A pericytes are associated with the CNS microvasculature. For clarity, we attach a screenshot of *Slc1a3* (GLAST) expression in cell types associated with the brain vasculature obtained from <<http://betsholtzlab.org/VascularSingleCells/database.html>>. While the data are clearly in line with our observations, their description in the paper is an oversimplification based on average expression scores for each vascular and perivascular cell type.

Gene symbol: *Slc1a3*

Figure A: [Brain data] Detailed expression in each cell

Slc1a3

Each line represents expression of *Slc1a3* in a single cell. The total number of analyzed cells per cell type cluster is indicated under the name of the cluster in the x-axis (e.g. 1088 pericytes (PC), 37 fibroblasts subtype 1 (FB1) and 49 fibroblasts subtype 2 (FB2) were analyzed).

Abbreviations: PC, pericytes; vSMC, venous smooth muscle cells; aaSMC, arteriole smooth muscle cells; aSMC, arterial smooth muscle cell; MG, microglia; FB1, fibroblast subtype 1; FB2, fibroblast subtype 2; OL, oligodendrocytes; EC1-EC3, endothelial cell subtypes 1-3; vEC, venous endothelial cells; capEC, capillary endothelial cells

Source: <<http://betsholtzlab.org/VascularSingleCells/database.html>>

Vanlandewijck, M., He, L. *et al.* A molecular atlas of cell types and zonation in the brain vasculature. *Nature*, 554, 475-480 (2018).

He, L., Vanlandewijck, M. *et al.* Data Descriptor: Single cell RNAseq of mouse brain and lung vascular and vessel-associated cell types. *Scientific Data*, Volume 5, Article number: 180160 (2018)

As mentioned by the reviewer, the study did not uncover multiple subtypes of pericytes in the brain. Identification of pericyte / perivascular cell subpopulations may require the use of a different combination of molecular markers to identify perivascular cells and a larger number of sorted perivascular cells. In addition, limitations related to single cell profiling using SmartSeq2, sequencing depth and bioinformatics analyses among other factors may have hindered the identification of pericyte subpopulations.

The expression data also revealed *Slc1a3* expression in a subset of fibroblasts. There are ongoing efforts in the Betsholtz lab to determine the precise location of these fibroblast subtypes in the brain. At the moment it is unclear whether our GLAST line targets these fibroblasts as well. We added the following discussion to the revised manuscript to point this out:

“Single-cell sequencing analyses of vascular cells in the mouse brain revealed that in addition to pericytes, GLAST (*Slc1a3*) is expressed by a subset of perivascular fibroblasts (Vanlandewijck *et al.*, *Nature*, 2018), suggesting that type A pericytes could be more heterogeneous than previously anticipated. Future studies exploring type A pericyte heterogeneity at the single cell level will be required to clarify this matter.”

In the EAE model of scar formation in regions of type A pericyte accumulation is there concomitant demyelination? The authors talk about demyelinated lesion but I couldn't readily find any evidence that particular lesions that were emphasized were actually demyelinated. While it is likely that such lesions were lacking myelin, staining sections for myelin to actually show this would be relatively easy.

We thank the reviewer for raising this point.

We have now added new images to the revised manuscript (figure 4i) where we show that scar formation by type A pericyte-derived cells coincides with demyelinated regions of the spinal cord after EAE.

Reviewer #4 (Remarks to the Author):

In this study, Dias et al. investigated fibrotic scarring in human pathological tissue and corresponding mouse models of penetrating and non-penetrating spinal cord injury, traumatic brain injury, ischemic stroke, multiple sclerosis and glioblastoma. They tried to determine the cellular origin of the fibrotic tissue scar in these spinal or brain lesions by using in vivo cell lineage tracing technique with immunofluorescence, and found that type A pericytes are the major source of scar-forming fibroblasts in many CNS lesions. They showed that the type A pericytes exhibited unique patterns and contributed to the scar formation after different CNS lesions. Overall, this is an interesting study. However, because current data are very preliminary, the major conclusions should be further validated in the pericyte-deficient mice (PDGFR β conditional knockout mice), and the molecular mechanisms underlying the recruitment of pericytes to the lesion sites and the differentiation of type A pericytes into fibroblasts to form the scars should be explored and defined. Current study is largely descriptive without mechanistic insights to support specific conclusions.

We thank the reviewer for finding our study interesting. We have addressed the comments as described below.

Major concerns:

1. To further confirm that pericytes are critical cells that contribute to the scar formation after brain lesions, brain pericyte-deficient mice (PDGFR β conditional knockout mice) should be used to examine whether pericyte deficiency attenuates the scar formation after brain lesions, because such mice are available for the study.

As suggested by the reviewer, we created a new mouse line for conditional, cell specific, functional deletion of PDGFR β in type A pericytes. For this, we crossed the PDGFR β ^{tm11Sor} conditional knockout line (Schmahl et al. Genes Dev 3255–3267, 2008) with our GLAST-CreER^{T2}; R26R-tdTom line, to allow for functional gene deletion of PDGFR β in type A pericytes and heritable labeling of type A pericytes after tamoxifen-induced genetic recombination (see figure R1a below, under point 2). We confirmed that the line fatefully deleted Pdgfr β in recombined type A pericytes at the protein level (figure R1b-i). However, all type A pericytes remained associated with the vascular wall (figure R1f-i) and the number of recombined type A pericytes had not changed 2 months after recombination induction (figure R1j-l). Based on these results, we concluded that the approach did not allow for reduction of the number of pericytes prior to injury to validate their functional role in fibrotic tissue formation.

In the adult mouse brain, a reduction of pericyte – endothelial coverage has been, for example, described for PDGF-B retention motif knockout (*Pdgfb*^{ret/ret}) mice, in which PDGF-B binding to heparan sulphate proteoglycan is disrupted (Armulik A et al., Nature. 2010 Nov 25;468(7323):557-61). The reduced pericyte coverage phenotype at adult stages is most likely of developmental origin, as the number of pericytes in the forebrain of *Pdgfb*^{ret/ret} mice is reduced by about 50% of normal at embryonic day 15.5 (Lindblom et al Genes Dev. 17(15):1835-1840, 2003). Furthermore, it remains elusive whether conditional deletion of PDGFR- β in pericytes (e.g. using a *Pdgfrb*-CreER line) in adult mice leads to reduced pericyte coverage, as such results have not been published to our knowledge.

To address the question of the reviewer in an alternative way, we employed a cell-specific and inducible genetic strategy to inhibit the generation of progeny by type A pericytes, via specifically blocking injury-induced proliferation. This was achieved by generating new transgenic mice that, in addition to carrying the GLAST-CreER^{T2} and R26R-EYFP alleles, were homozygous for *H-Ras* and *N-Ras* null alleles and for floxed *K-Ras* alleles (Drosten et al. EMBO J 29, 1091-1104, 2010), referred to as GLAST-Rasless-EYFP mice (figure 8f). We observed that control animals, where type A pericytes proliferate and leave the blood vessel wall extensively, developed a dense fibrotic lesion core enriched in PDGFR β -expressing stromal cells and widespread deposition of fibronectin and collagen I-rich ECM at 14 days after cortico-striatal stab lesions. On the contrary, animals in which type A pericyte proliferation was reduced and a higher percentage of cells remained attached to the blood vessel wall, presented significant attenuation of fibrotic scar tissue formation, revealed as smaller lesions with decreased PDGFR β ⁺ scarring compared to control animals. In these animals, we observed reduced deposition of fibronectin and collagen I compared to control animals, as observed previously after spinal cord injury (Göritz et al., Science 2011 ; Dias et al., Cell 2018). These new results are now described in the text under the section “Type A pericytes are required for extracellular matrix deposition after CNS lesions” and presented in the new figure 8g-l and supplementary figure 11.

2. It is not clear how the type A pericytes are recruited to the lesion sites. What is the molecular signaling involved in this? The molecular mechanisms should be investigated.

We previously showed that type A pericytes enter the lesion associated with sprouting vessels after spinal cord injury (Göritz et al, Science 2011). The importance of angiogenesis for fibrotic tissue formation was confirmed by a recent study, from the McTigue’s lab, indicating that reduced angiogenesis leads to a reduction of fibrotic scarring (Hesp et al. J. Neurosci. 38, 1366-1382, 2018).

Furthermore, our newly added experiments (described above in point 1), in which we genetically inhibited injury-induced type A pericyte proliferation, revealed an association between proliferation and detachment from the vascular wall, showing that reduced proliferation of type A pericytes was paralleled by a higher proportion of type A pericyte-derived cells remaining attached to the vasculature (supplementary Fig. 11a-f). These data suggest that proliferation is required for pericyte dissociation from the vascular wall.

We also tested if PDGFR β -mediated signaling is important for the recruitment of type A pericytes to fibrotic scarring. For this, we performed cell-specific functional deletion of PDGFR β in type A pericytes in adult mice using the newly generated GLAST-CreER^{T2}; R26R-tdTom; PDGFR β ^{tm11Sor} line (figure R1a). We induced gene deletion of PDGFR β in tdTom⁺ type A pericytes prior to a spinal cord crush injury by tamoxifen-mediated genetic recombination. The results did not show significant differences between heterozygous and homozygote knock out mice, suggesting that PDGFR β -mediated signaling may not be crucial for type A pericyte-derived scarring after spinal cord injury. We included these data as full figure into the rebuttal letter, to demonstrate our efforts to address the reviewer’s comments (Figure R1m-t).

[REDACTED]

[REDACTED]

3. Do these pericytes proliferate after they are recruited to the lesion sites or before the recruitment? If yes, how does the CNS lesion trigger pericyte proliferation? The molecular mechanisms underlying this should be determined.

We have assessed proliferation of type A pericytes and progeny at 5 days after spinal cord injury, brain stab lesions and ischemic stroke lesions, a time point when there is active recruitment of these cells from the blood vessel wall. For EAE and tumor experiments, where one time point was investigated, we assessed proliferation at 30 days post-EAE induction and 3 weeks post-tumor cell inoculation. Across all lesion models, we found that type A pericytes and progeny express the mitotic marker Ki67 both while on the vessel wall and away from the vessel wall. These results indicate that cells proliferate irrespective of their location in relation to the blood vessel wall and not only after they are recruited from the blood vessel wall. These new data were added to the manuscript text and can be found across figures 1-7 and supplementary figure 4 of the revised manuscript. Nonetheless, when genetically inhibiting proliferation of type A pericytes, we noticed that more cells remained associated with the vascular wall (Supplementary figure 11a-f). This suggests that proliferation is required for pericyte dissociation from the vascular wall.

4. Do the recruited type A pericytes differentiate into fibroblasts to form the scar after they are recruited to the lesion sites? If yes, what are the molecular mechanisms underlying the differentiation.

We have previously shown (Göritz et al., Science 2011) that type A pericyte progeny transiently express the myofibroblast marker α SMA during the wound contraction phase

after spinal cord injury and are surrounded by dense extracellular matrix (ECM) deposits of fibronectin. These data support the notion that type A pericytes differentiate into a (myo)fibroblast stage after injury and are involved in fibrotic ECM production.

In the revised manuscript we have investigated whether type A pericytes (1) differentiate into myofibroblasts, also known as wound healing fibroblasts, by co-staining for α SMA; (2) express the fibroblast marker vimentin; (3) are surrounded by ECM proteins known to be upregulated after injury, such as fibronectin and collagen I. We show that type A pericytes in the uninjured spinal cord and brain do not express α SMA (Supplementary figures 1f and 2e). However, type A pericyte-derived cells acquire expression of the myofibroblast marker α SMA and express the fibroblast marker vimentin in all lesion models that generate stromal fibroblasts (figures 1,2,3,4,6,7 and supplementary figure 4). Interestingly, after ischemic striatal stroke (figure 5), type A pericytes remain on the blood vessel wall, and do not upregulate α SMA at 5 days post-lesion, indicating that these cells do not transition into myofibroblasts and do not differentiate into stromal fibroblasts in this lesion model. In a new set of experiments included in the revised manuscript we show that after ischemic stroke restricted to the cerebral cortex, type A pericyte-derived cells leave the vascular wall and differentiate into α SMA-expressing myofibroblasts (figure 6). Taken together, these data demonstrate that type A pericytes differentiate into (myo)fibroblasts after they dissociate from the vascular wall and are recruited to the lesion.

In an independent new set of experiments (figure 8a-e of the revised manuscript) we show that 14 days following striatal ischemic stroke, in which type A pericytes remained attached to the vascular wall, there is a slight but not significant increase in the deposition of fibronectin and collagen I in the ischemic lesion core, compared to the contralateral stroke side. On the contrary, 14 days after cortical ischemic stroke, with widespread, non-vessel associated type A pericyte-derived fibroblasts, we observe a robust increase in deposition of fibronectin and collagen I in comparison to striatal strokes and to the contralateral stroke side.

Taken together, these results indicate that detachment of type A pericyte-derived cells from the vascular wall, which leads to the generation of ECM-producing (myofibroblasts) / stromal fibroblasts, is required for deposition of fibronectin and collagen I ECM proteins.

5. It is well recognized that there is no fibroblast in the brain. It is not clear how they define the fibroblasts in the scars after brain lesions. More fibroblast markers should be used to validate the cells in the scars.

Stromal fibroblasts in the scars were defined in the text as PDGFR β ⁺ cells with no direct contact to the blood vessel wall (extravascular location), as quoted below:

“Fibrotic scar tissue contains PDGFR β ⁺ stromal cells associated with the blood vessel wall (ON the vessel), representing perivascular cells, and extravascular (OFF the vessel) PDGFR β ⁺ cells, representing stromal fibroblasts”.

In the revised manuscript we have performed new immunostainings across all lesion models and show that in addition to PDGFR β , type A pericyte-derived stromal fibroblasts express α SMA and vimentin, proteins that are present in wound healing fibroblast’s actin stress fibers and fibroblast’s intermediate filaments, respectively. As the main function of fibroblasts is production of ECM, we additionally show that type A pericyte-derived cells are encased by fibronectin and collagen type I ECM deposits in all CNS lesion models investigated, advocating for a role of these cells on production of ECM components. These

new data was added to the manuscript text and can be found across figures 1-7 and supplementary figure 4 of the revised manuscript.

6. It is not clear why only type A pericytes contribute to the scar formation. What is unique property of type A pericytes for this function? What are the major difference between type A pericytes and other pericytes in the brain? More pericyte markers should be used to define these pericytes.

Similar to the spinal cord, type A pericytes in the uninjured brain comprise about 10% of all PDGFR β -expressing perivascular cells on the microvasculature. We have now added more markers to define type A pericytes in the brain (and spinal cord) and to distinguish them from other pericytes. In addition to GLAST, we show that type A pericytes in the brain and spinal cord express PDGFR β , CD13 and *Tbx18*, but are not marked by desmin and alpha smooth muscle actin (α SMA), found in other pericytes and vascular smooth muscle cells. These new data was added to the manuscript text and can be found in supplementary figures 1 and 2 of the revised manuscript.

In our previous work (Dias *et al.* Cell, 2018) we have extensively showed that inhibiting the generation of progeny by type A pericytes results in attenuation of fibrotic scarring and efficiently reduces deposition of fibrotic extracellular matrix (ECM) in the injured spinal cord. In the revised manuscript we have conducted new experiments employing the same genetic strategy to inhibit type A pericyte proliferation and show that type A pericyte-derived cells are required for ECM deposition after stab lesions in the brain (figure 8f-h). These results suggest that a unique property of type A pericytes that underlie their specific contribution to scar formation is their involvement in ECM production.

Future studies exploring perivascular cell heterogeneity at the single cell level will likely be required to shed light on what are the unique molecular properties of type A pericytes that trigger their specific involvement in scar formation in comparison to other perivascular cells.

7. What are other cellular components beside the pericyte-derived fibroblasts? Are the astrocytes involved in the scar formation after injury? What are the fractions of the pericyte-derived fibroblasts in the scars after different lesions?

The mature scar that forms after CNS injury is a compartmentalized, multicellular structure, where a reactive glial scar surrounds a central lesion core filled with non-neural fibrotic tissue, referred to as fibrotic scar. In addition to pericyte-derived fibroblasts, the fibrotic scar is mainly constituted by peripherally-derived macrophages, whereas the glial component of the scar is primarily made of reactive astrocytes and NG2-expressing glia. So, astrocytes are greatly involved in scar formation after CNS injury, particularly as the main component of the glial scar, as we have shown across all lesion models (see GFAP⁺ reactive astrocytes participating in glial scarring in figures 1-7 and supplementary figures 4, 6, 8 and 10).

We also already present data on the contribution of type A pericyte-derived cells to all PDGFR β ⁺ stromal fibroblasts in the scars across all lesion models (see last graph of figures 1-7 and supplementary figure 4, in which the % recombined PDGFR β ⁺ cells / total PDGFR β ⁺ cells is presented). In summary, we show that type A pericytes are the main source of PDGFR β -expressing stromal cells after penetrating and non-penetrating spinal cord injuries, traumatic brain injury, ischemic stroke and EAE, but contribute less extensively to tumor stroma in the GL261 orthotopic glioma model.

8. The cell lineage tracing system (EYFP+) was used in this study, but why the EYFP+ cells were identified by immunostaining of GFP as indicated in the method section is not clear.

EYFP is a genetic mutation of GFP that shares large sequence homology and is therefore recognized by anti GFP antibodies. The GFP antibody is used to enhance the EYFP signal. Specific antibodies for YFP do not exist to our knowledge.

9. In figures 1, 2, and 4, it is necessary to show the representative images for the distribution and location of the type A pericytes 7 weeks following spinal cord injury beside the quantified analysis.

The manuscript already contained representative overview images of the distribution and location of type A pericyte-derived cells at 6-7 weeks after injury (Figures 1e, 2e, 4e and supplementary figure 3e of former manuscript; Figures 1f, 2f, 5f and supplementary figure 4f of the revised manuscript).

Nonetheless, we have created a new supplementary figure (Supplementary Fig. 7) where we show higher magnification images of the lesion core and the distribution and location of type A pericytes and progeny 6-7 weeks after complete spinal cord crush, dorsal *funiculus* incision, cortico-striatal stab wound and striatal ischemic stroke. As observed at 5 days and 14 days after injury, type A pericyte-derived cells reside in the lesion core as part of the fibrotic scar and a fraction has dissociated from the blood vessel wall in all models, with the exception of striatal ischemic stroke.

10. In most figures, the nuclei were not labeled with DAPI, which makes very hard to determine and quantify type A pericytes and other cells.

Quantitative data on type A pericyte-derived cells is provided for every figure across all CNS lesions investigated. We opted to not show the labeled nuclei with DAPI in most of the figures because it becomes too crowded and it is more difficult for the reader to perceive co-localization of 2 or 3 markers in the same cells. If the reviewer considers this information critical for certain panels, we can provide the specified images as supplementary figures.

11. Most figure legends were poorly described and lack critical information.

The figure legends were updated and now include additional information.

Minor points:

1. In figures 3d, it is not clear what dWM, iWM, vLWM, and vWM stand for.

We have now indicated in the figure legend that dWM, iWM, vLWM and vWM stand for dorsal, intermediate, ventro-lateral and ventral white matter, respectively.

2. In figures 3 and 4, the damaged areas were not clearly indicated. It would be necessary to indicate the damage areas with some markers such as cd45 or cd3.

We have added additional images to former figures 3 and 4 (Figs. 4i-k and Fig. 5k,l,o,p in the current revised form of the manuscript, respectively) indicating that the damaged areas are filled with CD45⁺ immune cells, including CD3⁺ T-cells.

3. In figures 1g, 3g, 5d and 6, how did they define those GFAP+ astrocytes are the reactive astrocytes? GFAP+ is a general marker for most astrocytes.

In the adult mouse spinal cord, cerebral cortex and striatum, GFAP is only expressed by a subset of parenchymal astrocytes under homeostatic conditions. Pan astrocytic markers under uninjured conditions are S100 β , Aldh1L1, glutamine synthetase and aldolase C, among others (see supplementary figure 3). In response to CNS lesions, astrocytes become activated, undergo cellular hypertrophy and exhibit a striking increase in GFAP immunoreactivity, a process collectively referred to as reactive astrogliosis. GFAP is therefore commonly used as a reactive astrocyte marker.

REVIEWER COMMENTS

Reviewer #1 (Remarks to the Author):

The authors have responded well to all of my criticisms, and so I believe this paper should be published. This is a remarkable amount of work documenting how type A pericytes respond to brain injury or pathology. It remains largely descriptive but will significantly inform future studies. The following minor points should be addressed:

(1) Define briefly what is meant by "away from the vessel wall" in the main text where such data are first presented (i.e. no contact of processes or soma with the vessel, and soma > 8um from vessel wall).

(2) On page 44-45 state whether the distance was measured in 2D images or in 3D.

Reviewer #2 (Remarks to the Author):

The authors have carefully addressed my previous comments.

Reviewer #3 (Remarks to the Author):

The authors have responded to the criticisms and suggestions for additional work that I had suggested. In addition, they have provided extensive rebuttal of several papers that have concluded that the fibrotic component of scarring in the CNS is not due to pericyte involvement but, rather, the proliferation, migration and matrix production by adventitial fibroblasts. Thus, other labs suggest simply that fibroblasts generate more fibroblasts and produce fibrotic matrix in response to injury. The Goritz lab has championed the idea in this and previous manuscripts that after certain types of extensive injury to the CNS, a subtype of pericyte, called type A, breaks free of the basal lamina, proliferates, completely changes its phenotype and becomes fibroblastic and produces all of the cadre of fibrotic matrix components of scar. This manuscript contains an enormous amount of data which represents a great deal of work in support of pericyte origin of fibrotic scar. None-the-less, this reviewer remains concerned that the GLAST-CreER transgenic mouse line with EYFP or tdTomato reporter alleles may not be labeling solely so-called type A pericytes and a sub-set of easily distinguished astrocytes but also a population of bonafide fibroblasts. True pericytes should be very tightly juxtaposed to the outer plasma membrane of endothelial cells and embedded within the thin basal lamina that surrounds them and the endothelial cell. The authors of the present paper do not demonstrate this for so-called type A pericytes. Indeed, in suppl fig 2F there is a clear and distinct gap between the EYFP labeled cell and the endothelium. The presence of this gap in this and other images suggests the possibility that these cells are abluminal to pericytes which, of course, would disqualify them as being pericytes, even though they express PDGFRb. In addition, there is essentially no evidence presented that type A pericytes make the NG2 proteoglycan which is a well-known marker of pericytes. In figure 8 the accumulation of fibronectin and collagen attributed to type A pericytes appears to this reviewer to be of leptomeningeal fibroblast origin. To be more convincing the authors should perform immunoEM to show that the gap between their EYFP labeled or PDGFRb labeled cells and the endothelial cell membrane is not filled with collagen or NG2+ pericytes helping to rule out that they are abluminal to pericytes or perhaps smooth muscle cells and are, thusly, fibroblasts. In addition, they should demonstrate that prior to or just after injury their type A pericytes are, in fact, tightly opposed to endothelial cells and embedded within a basal lamina.

Reviewer #4 (Remarks to the Author):

The authors have done additional experiments to address the major concerns and validate the results. The main conclusions have been strengthened. The revised manuscript has been significantly improved.

Response to the reviewer's comments

Reviewer #1 (Remarks to the Author):

The authors have responded well to all of my criticisms, and so I believe this paper should be published. This is a remarkable amount of work documenting how type A pericytes respond to brain injury or pathology. It remains largely descriptive but will significantly inform future studies.

We thank the reviewer for the positive feedback on our manuscript. We have addressed the additional comments as specified below:

The following minor points should be addressed:

(1) Define briefly what is meant by “away from the vessel wall” in the main text where such data are first presented (i.e. no contact of processes or soma with the vessel, and soma > 8um from vessel wall).

We have changed the main text according to the reviewer's instructions and further defined the terms ON and OFF the vessel.

(2) On page 44-45 state whether the distance was measured in 2D images or in 3D.

We have specified that the distance was measured in 2D images.

Reviewer #2 (Remarks to the Author):

The authors have carefully addressed my previous comments.

We are happy that we could address the comments of this reviewer.

Reviewer #3 (Remarks to the Author):

The authors have responded to the criticisms and suggestions for additional work that I had suggested. In addition, they have provided extensive rebuttal of several papers that have concluded that the fibrotic component of scarring in the CNS is not due to pericyte involvement but, rather, the proliferation, migration and matrix production by adventitial fibroblasts. Thus, other labs suggest simply that fibroblasts generate more fibroblasts and produce fibrotic matrix in response to injury. The Goritz lab has championed the idea in this

and previous manuscripts that after certain types of extensive injury to the CNS, a subtype of pericyte, called type A, breaks free of the basal lamina, proliferates, completely changes its phenotype and becomes fibroblastic and produces all of the cadre of fibrotic matrix components of scar. This manuscript contains an enormous amount of data which represents a great deal of work in support of pericyte origin of fibrotic scar.

We are happy that we could address the comments of this reviewer.

None-the-less, this reviewer remains concerned that the GLAST-CreER transgenic mouse line with EYFP or tdTomato reporter alleles may not be labeling solely so-called type A pericytes and a sub-set of easily distinguished astrocytes but also a population of bonafide fibroblasts. True pericytes should be very tightly juxtaposed to the outer plasma membrane of endothelial cells and embedded within the thin basal lamina that surrounds them and the endothelial cell. The authors of the present paper do not demonstrate this for so-called type A pericytes. Indeed, in suppl fig 2F there is a clear and distinct gap between the EYFP labeled cell and the endothelium. The presence of this gap in this and other images suggests the possibility that these cells are abluminal to pericytes which, of course, would disqualify them as being pericytes, even though they express PDGFRb. In addition, there is essentially no evidence presented that type A pericytes make the NG2 proteoglycan which is a well-known marker of pericytes.

Type A pericytes have a perivascular position, are embedded within the basal lamina that surrounds them and the endothelial cells, express the pericytic markers PDGFR β and CD13, are surrounded by astrocytic endfeet and can be found in the microvasculature, at the capillary level. Their characteristics in terms of location in the vascular wall and marker expression fall in the definition of pericytes. Perivascular fibroblasts have been described to locate outside the vascular basal membrane along larger penetrating vessels. We do not find that these criteria match the characteristics of type A pericytes. The function to generate fibrotic tissue in response to injury clearly distinguishes type A pericytes from the majority of pericytes. By naming this subset of pericytes as type A, we emphasize their (functional) distinction towards other pericytes. Throughout the manuscript, we use the term type A pericytes to specify mural cells that generate fibrotic tissue and are, under the currently accepted definitions, best described as a specific subset of pericytes.

In figure 8 the accumulation of fibronectin and collagen attributed to type A pericytes appears to this reviewer to be of leptomeningeal fibroblast origin.

We have a different interpretation of the results regarding this point. In our view, what can be appreciated in Figures 8g and h is that in the Cre WT group fibronectin and collagen I signals localize to the brain surface (meningeal space) and within the brain parenchyma. In the Cre+ group, in which fibrotic scarring by type A pericytes is reduced, fibronectin and collagen I signals remain predominantly at the brain surface, while their deposition is significantly reduced within the brain parenchyma. Thus, type A pericytes contribute to fibrotic ECM deposition within the brain parenchyma and reduction of type A pericyte-derived scarring does not substantially impact on meningeal-derived ECM deposition. We have clarified this point in the main text.

We understand that the continuous appearance of fibrous ECM may give the impression that the source may be the same. However, this experiment demonstrates that fibrotic ECM within the brain parenchyma is derived from type A pericytes, while it has a different origin at the brain surface. Furthermore, studying early time points after injury, one can appreciate the appearance of type A pericyte-derived cells from deep within the brain/spinal cord parenchyma (Fig. 1d,i and j; Fig. 2d; Fig. 3b; Fig. 5d; Fig. 6b,d,f and g; Fig. S5d; Fig. S6c and Fig. S11b).

To be more convincing the authors should perform immunoEM to show that the gap between their EYFP labeled or PDGFRb labeled cells and the endothelial cell membrane is not filled with collagen or NG2+ pericytes helping to rule out that they are abluminal to pericytes or perhaps smooth muscle cells and are, thusly, fibroblasts. In addition, they should demonstrate that prior to or just after injury their type A pericytes are, in fact, tightly opposed to endothelial cells and embedded within a basal lamina.

We have now included immunoEM characterization of type A pericytes. For this, we immuno-gold labelled type A pericytes with antibodies against red fluorescent protein (RFP), which recognizes tdTomato, in uninjured GLAST-CreER^{T2};R26R-tdTom mice. Our results show that immuno-gold labelled type A pericytes are in direct contact to the endothelial basal membrane and embedded within the thin basal lamina that surrounds them and the endothelial cells. Only at positions in which type A pericytes overlap with other pericytes, type A pericytes are found in an abluminal position to the underlying pericyte. However, also in these cases, type A pericytes are surrounded by a basal lamina. The results are shown in the new supplemental Figure S2 and described in the text.

Reviewer #4 (Remarks to the Author):

The authors have done additional experiments to address the major concerns and validate the results. The main conclusions have been strengthened. The revised manuscript has been significantly improved.

We thank the reviewer for the positive feedback on our manuscript.

REVIEWERS' COMMENTS

Reviewer #3 (Remarks to the Author):

The authors have adequately addressed my concerns. The addition of the immuno EM is quite convincing. The paper is ready for publication.

Response to the reviewer's comments

Reviewer #3 (Remarks to the Author):

The authors have adequately addressed my concerns. The addition of the immuno EM is quite convincing. The paper is ready for publication.

We are glad to read we have adequately addressed the concerns from this reviewer.